# A Piece-wise Polynomial Filtering Approach for Graph Neural Networks

## Abstract

Graph Neural Networks (GNNs) exploit signals from node features and the input graph topology to improve node classification task performance. However, these models tend to perform poorly on heterophilic graphs, where connected nodes have different labels. Recently proposed GNNs work across graphs having varying levels of homophily. Among these, models relying on polynomial graph filters have shown promise. We observe that solutions to these polynomial graph filter models are also solutions to an overdetermined system of equations. It suggests that in some instances, the model needs to learn a reasonably high order polynomial. On investigation, we find the proposed models ineffective at learning such polynomials due to their designs. To mitigate this issue, we perform an eigendecomposition of the graph and propose to learn multiple adaptive polynomial filters acting on different subsets of the spectrum. We theoretically and empirically show that our proposed model learns a better filter, thereby improving classification accuracy. We study various aspects of our proposed model including, dependency on the number of eigencomponents utilized, latent polynomial filters learned, and performance of the individual polynomials on the node classification task. We further show that our model is scalable by evaluating over large graphs. Our model achieves performance gains of up to 10% over the state-of-the-art models and outperforms existing polynomial filter-based approaches in general. Our anonymized code is available at https://tinyurl.com/PPGNN.

## 1 Introduction

In this work, we are interested in the problem of classifying nodes in a graph. In this problem, a graph and features for all the nodes in the graph are made available. We are also given labels for a subset of the nodes in the graph. The learning model utilizes this information to predict labels for the remaining nodes. Graph Neural Networks (GNNs) perform well on such problems (Kipf & Welling, 2017). GNNs predict a node's label by aggregating information from its neighbours. Thus, the performance of GNN is dependent on how well the given graph correlates with the node labels. Characterizing the correlation between the graph and node labels is an active area of research. Several metrics have been proposed including edge homophily (Abu-El-Haija et al., 2019a; Zhu et al., 2020), node homophily (Pei et al., 2020), class homophily (Lim et al., 2021). All these metrics show that GNNs perform well when the graphs and node labels are positively correlated. For example, in the simplest case, GNNs work well when the node and its neighbours share similar labels. However, the performance can be poor if this criterion is not satisfied.

Several proposed GNN models attempt to be robust to the underlying correlation between graphs and labels. Some approaches modify the aggregation mechanism (Pei et al., 2020; Zhu et al., 2020; Kim & Oh, 2021), while other approaches propose to estimate and leverage the label-label compatibility matrix as a prior (Zhu et al., 2021). More recent approaches have tackled this problem from a filter learning perspective (Bo et al., 2021; Chien et al., 2021). These methods learn a filter function that operates on the eigenvalues of the graph. With eigenvalues having frequency interpretations (Shuman et al., 2013), the filter function selectively accentuates and suppresses various frequencies as required by the task. This process enables the model to learn better embeddings, which translates to improved performance. Our interest lies in the area of modelling GNNs that can learn an effective filter.

Chien et al. (2021) proposed to learn a polynomial filter. This model achieves good performance gains on several datasets with varying correlations between graphs and labels. However, we observe that learning a polynomial of a degree lower than the number of nodes can be seen as solving an over-determined system of equations. Our observation suggests needing a higher-order polynomial to model richer or more complex frequency responses for certain datasets. The design of the model proposed in Chien et al. (2021) makes it ineffective in learning a higher-order polynomial. It happens because the model attempts to overcome over-smoothing by giving smaller coefficient values to larger powers. In this paper, we address this issue and make the following contributions:

- Inspired by Chien et al. (2021), we propose to learn a polynomial filter; however, we do so by learning a sum of polynomials over different subsets of eigenvalues. Such modelling enables learning higher-order polynomials with several low order polynomials acting over the different subsets.

- Such modelling, however, requires an eigendecomposition of the graph, which can be expensive. We leverage the model in Chien et al. (2021) and the fact that there exist efficient algorithms for computing top and bottom eigen components to propose a practically efficient model.

- We present theoretical analysis that suggests that our sum of polynomials approach can approximate a latent optimal filter better than a single polynomial. We also show that the space of learnable filters and graphs using our approach is larger than what is possible with Chien et al. (2021).

- Our experimental results show that our model performs better than state-of-the-art methods on several datasets, giving up to 10% gains. These gains indicate that modelling the filter as a sum of polynomials is needed to approximate the latent filter better. We also answer other research questions through ablative studies, including the importance of the top and bottom eigenvalues, the number of top and bottom eigenvalues needed, the number of polynomials needed, and the trade-offs involved.

The rest of the paper is organized as follows: We present related work in Section 2, problem setup and motivation for the approach in Section 3, details of the proposed approach with complexity and theoretical analysis in Section 4 and finally experimental results and empirical analysis in Section 5.

## 2 RELATED WORKS

In the recent times, Graph Neural Networks (GNNs) have become an increasingly popular method for semi-supervised classification with graphs. Bruna et al. (2014) set the stage for early GNN models, which was then followed by various modifications (Defferrard et al., 2016; Kipf & Welling, 2017; Hamilton et al., 2017; Veličković et al., 2018). Incorporating random walk information (Abu-El-Haija et al., 2019b;a; Li et al., 2018) gave further improvements in these models, but they still suffer from over smoothing. Several proposed models attempt to overcome this problem (Klicpera et al., 2019; Lukovnikov & Fischer, 2021; Chamberlain et al., 2021; Yang et al., 2021).

Another line of research explored the question of which graphs worked well with GNNs. The critical understanding was that the performance of GNN was dependent on the correlation of the graphs with the node labels. Several approaches (Abu-El-Haija et al., 2019a; Zhu et al., 2020; Wang & Derr, 2021) considered edge homophily and proposed a robust GNN model by aggregating information from several higher-order hops. Kim & Oh (2021) also considered edge homophily and mitigated the issue by learning robust attention models. Pei et al. (2020) talks about node homophily and proposes to aggregate information from neighbours in the graph and neighbours inferred from the latent space. Zhu et al. (2021) proposes to estimate label-label compatibility matrix and uses it as a prior to update posterior belief on the labels. However, these approaches still rely on the given graphs for information like the hop neighbours etc., which might be poor approximations of the desired neighbours with similar labels.

Some of the recent approaches focused on learning filter functions that operate on the eigenvalues of the graph. Learning these filter functions can be viewed as directly adapting the graph for the desired task. Bo et al. (2021) models the filter function as an attention mechanism on the edges, which learns the difference in the proportion of low-pass and high-pass frequency signals. Chien et al.

(2021) proposes a polynomial filter on the eigenvalues that directly adapts the graph for the desired task. Zheng et al. (2021) decompose the graph into low-pass and high-pass frequencies, and define a framelet based convolutional model. Our work is closely related to these lines of exploration. In this work, we propose to learn a filter function as a sum of polynomials over different subsets of the eigenvalues, enabling design of effective filters to model task-specific complex frequency responses with compute trade-offs.

## 3 PROBLEM SETUP AND MOTIVATION

We focus on the problem of semi-supervised node classification on a simple graph $\mathcal{G} = (\mathcal{V}, \mathcal{E})$, where $\mathcal{V}$ is the set of vertices and $\mathcal{E}$ is the set of edges. Let $\mathbf{A} \in \{0, 1\}^{n \times n}$ be the adjacency matrix associated with $\mathcal{G}$, where $n = |\mathcal{V}|$ is the number of nodes. Let $\mathcal{Y}$ be the set of all possible class labels. Let $\mathbf{X} \in \mathbb{R}^{n \times d}$ be the $d$-dimensional feature matrix for all the nodes in the graph. Given a training set of nodes $\mathcal{D} \subset \mathcal{V}$ whose labels are known, along with $\mathbf{A}$ and $\mathbf{X}$, our goal is to predict the labels of the remaining nodes. Let $\mathbf{A_I} = \mathbf{A} + \mathbf{I}$ where $\mathbf{I}$ is the identity matrix. Let $\mathbf{D_{A_I}}$ be the degree matrix of $\mathbf{A_I}$ and $\widetilde{\mathbf{A}} = \mathbf{D_{A_I}}^{-1/2} \mathbf{A_I} \mathbf{D_{A_I}}^{-1/2}$. Let $\widetilde{\mathbf{A}} = \mathbf{U \Lambda U}^T$ be the eigendecomposition of $\widetilde{\mathbf{A}}$. Then, the spectral convolution of $\mathbf{X}$ on the graph $\mathbf{A}$ can be defined via the reference operator $\widetilde{\mathbf{A}}$ and a filter function $h$ operating on the eigenvalues, in the Fourier domain (Tremblay et al., 2017; Chien et al., 2021) as,

$$Z = \mathbf{U}H(\mathbf{\Lambda})\mathbf{U}^T X = \mathbf{U}\text{diag}(V\boldsymbol{\alpha})\mathbf{U}^T X = \sum_{j=1}^{k} \alpha_j \widetilde{\mathbf{A}}^j X \tag{1}$$

where $H(\mathbf{\Lambda}) = \text{diag}(h(\mathbf{\Lambda}))$. In this process, the filter function is essentially adapting the graph for the desired task at hand. The second equality follows from using a polynomial filter, $H(\lambda) = \sum_{i=1}^{k} \alpha_i \lambda^i$ where $\alpha_i$'s are coefficients of the polynomial, $k$ is the order of the polynomial and $\lambda$ is any eigenvalue from $\mathbf{\Lambda}$. Note that $H(\mathbf{\Lambda}) = \text{diag}(V\boldsymbol{\alpha})$ where $V \in \mathbb{R}^{n \times k}$ is a Vandermonde matrix constructed using eigenvalues from $\mathbf{\Lambda}$ and $\boldsymbol{\alpha} \in \mathbb{R}^{k \times 1}$ is the vector consisting of $\alpha_j$'s. The last equality follows from the eigendecomposition properties, enabling avoidance of eigendecomposition. It is well-known that polynomial filters can approximate any graph filter (Shuman et al., 2013; Tremblay et al., 2017). However, several practical challenges arise in learning a good polynomial filter. We briefly discuss them below.

Let $h^*$ be the optimal filter for some given task. Learning a polynomial filter involves the approximation: $h^* \approx V\boldsymbol{\alpha}$. Notice that this system is over-determined since the number of unknowns ($k$, the size of $\boldsymbol{\alpha}$) is less than the number of equations ($n$, the number of nodes). To obtain a consistent solution, we may have to work with higher-order polynomials. Working with higher powers poses several practical challenges, and the following observations are in order. First, node features computed via $\widetilde{\mathbf{A}}^j X$ becomes indistinguishable (see Figure 1a and A.2.1). Since the node features are indistinguishable, the importance of the coefficient associated with $\widetilde{\mathbf{A}}^j X$ for the task at hand reduces significantly. These observations are in line with Theorem 4.2 in Chien et al. (2021), where it states that as over-smoothing happens, the corresponding coefficients of the polynomial go to zero. We observe that with insignificant contributions from higher-order terms, the performance (aka test accuracy) does not improve with the increase in the order of the polynomial (See Figure 1b, 1c and A.2.2). Next, one can think of addressing this issue with $\widetilde{\mathbf{A}}^j X$ by directly working with full eigendecomposition. However, this approach is prohibitively expensive for large graphs. Even if we were to do this, $\lambda_i$'s are in the range of $[-1, 1]$ and thus $\lambda_i^j$ will diminish with higher powers. Therefore, it will continue to be less effective in learning coefficients of a higher power. Motivated by the above observations, we propose a novel piece-wise polynomial filter learning approach.

## 4 PROPOSED APPROACH

The central goal of our approach is to get improved task-specific performance by learning effective task-specific graph filters. We propose to learn the polynomial filter as a sum of polynomials operating on different eigenvalue intervals, taking task-specific requirements and practical considerations into account. We show that our proposed filter design can approximate the latent optimal graph

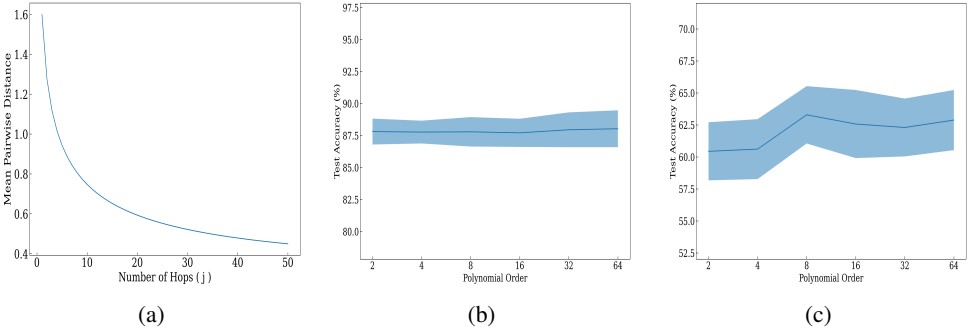

Figure 1: In (a), we plot the average of pairwise distances between node features for the Cora dataset, after computing $\widetilde{\mathbf{A}}^j X$ for increasing $j$ values. X-axis represents the various powers $j$ and the Y-axis represents the average of pairwise distances between node features. In (b) and (c), we plot the test accuracies of the model in Chien et al. (2021) for increasing order of polynomials for Cora and Chameleon dataset respectively. X-axis represents the order of the polynomial and Y-axis represents the test accuracy achieved for that order.

filter better than a single polynomial, and the resultant class of learnable filters/graphs is richer. We briefly discuss the training and computational complexity of the proposed model.

## 4.1 PIECE-WISE POLYNOMIAL/SPLINE GRAPH FILTER FUNCTION

We model the filtering function $h(\lambda)$ as a piece-wise polynomial or spline function where each polynomial is of a lower degree (e.g., a cubic polynomial). We partition the eigen spectrum in $[-1, 1]$ (or $[0, 2]$ as needed) into contiguous intervals and approximate the desired frequency response by fitting a low degree polynomial in each interval. This process helps us to learn a more complex shaped frequency response as needed for the task. Let $\mathcal{S} = \{\sigma_1, \sigma_2, \ldots, \sigma_m\}$ denote a partition set with $m$ contiguous intervals and $h_{i,k_i}(\lambda; \gamma_i)$ denote a $k_i$-degree polynomial filter function of the interval $\sigma_i$ with polynomial coefficients $\gamma_i$. We define PP-GNN filter function as:

$$h(\lambda) = \sum_{\sigma_i \in \mathcal{S}} h_{i,k_i}(\lambda; \gamma_i) \tag{2}$$

and learn a smooth filter function by imposing additional constraints to maintain continuity between polynomials of contiguous intervals at different endpoints (*aka* knots). This class of filter functions is rich, and its complexity is controlled by choosing intervals (i.e., endpoints and number of partitions) and polynomial degrees. Given the filter function, we compute the node embedding matrix as:

$$\mathbf{Z} = \sum_{i=1}^{m} \mathbf{U}_i H_i(\gamma_\mathbf{i}) \mathbf{U}_i^T \mathbf{Z}_0(\mathbf{X}; \mathbf{\Theta}) \tag{3}$$

where $\mathbf{U}_i$ is a matrix with eigenvectors corresponding to eigenvalues that lie in $\sigma_i$, $H_i(\gamma_i)$ is the diagonal matrix with diagonals containing the $h_i$ evaluated at the eigenvalues and $\mathbf{Z}_0(\mathbf{X}; \mathbf{\Theta})$ is an MLP network with parameters $\mathbf{\Theta}$.

## 4.2 PRACTICAL AND IMPLEMENTATION CONSIDERATIONS

The filter function (3) requires computing eigendecomposition of $\widetilde{\mathbf{A}}$ and is expensive, therefore, not scalable for very large graphs. Thus, having finer control to learn complex frequency responses comes with a high computational cost. We address this problem by devising a graph filter function that enables a trade-off between computational cost and frequency response control. First, we make the following observations:

1. Top-k and bottom-k eigenvalues represent $k$ low and high frequencies. There are efficient algorithms and off-the-shelf library packages available to get top-k and bottom-k eigenvalues and eigenvectors of sparse matrices.

2. GPR-GNN method also learns a graph filter but operates on the entire eigen spectrum by sharing the filter coefficients across the spectrum. Therefore, it is a special case of our proposed model (3). Since GPR-GNN learns a global polynomial by having shared coefficients across the spectrum (like learning a global polynomial to fit a function), it is not flexible enough to learn complex frequency responses, as needed in several practical applications. Note that increasing the degree of the global polynomial does not help beyond some high degree specific to tasks. We demonstrate this through extensive experiments on benchmark datasets in the experiment and appendix sections. One important advantage of GPR-GNN is that it does not require computation of eigendecomposition; therefore, efficient.

3. Many recent works, including GPR-GNN, investigated the problem of designing robust graph neural networks that work well across homophilic and heterophilic graphs. They found that graph filters that amplify or attenuate low and high-frequency components of signals (i.e., low-pass and high-pass filters) are critical to improving performance on several benchmark datasets.

**Efficient Variant.** Using the above observations, we propose an efficient variant of (2) as:

$$\tilde{h}(\lambda) = \eta_l \sum_{\sigma_i \in \mathcal{S}^l} h_i^{(l)}(\lambda; \gamma_i^{(l)}) + \eta_h \sum_{\sigma_i \in \mathcal{S}^h} h_i^{(h)}(\lambda; \gamma_i^{(h)}) + \eta_{gpr} h_{gpr}(\lambda; \gamma) \tag{4}$$

where $\mathcal{S}^l$ consists of partitions over low frequency components, $\mathcal{S}^h$ consists of partitions over high frequency components, the first and second terms fit piece-wise polynomials[1] in low and high frequency regions, as indicated through superscripts. However, we do not want to lose any useful information from other frequencies in the central region, yet maintain efficiency. We achieve this by adding the GPR-GNN filter function, $h_{gpr}(\lambda; \gamma)$, which is computationally efficient. Since $h_{gpr}(\lambda; \gamma)$ is a special case of (2) and the terms in (4) are additive, it is easy to see that (4) is same as (2) with a modified set of polynomial coefficients. In practice, the required low and high-frequency components are computed based on affordable computational cost. Furthermore, we can also control the contributions from each term by setting or optimizing over hyperparameters, $\eta_l$, $\eta_h$ and $\eta_{gpr}$. Thus, the proposed model offers richer capability and flexibility to learn complex frequency response and balance computation costs over GPR-GNN.

**Model Training.** Like GPR-GNN, we apply SOFTMAX activation function on (3) and use the standard cross-entropy loss function to learn the sets of polynomial coefficients ($\gamma$) and classifier model parameters ($\Theta$) using labeled data. To ensure smoothness of the learned filter functions, we add a regularization term that penalizes squared differences between the function values of polynomials of contiguous intervals at each other's interval end-points. More details can be found in the appendix (A.4.1).

**Computational Cost.** There is some pre-training cost of computing the eigendecomposition for top and bottom k eigenvalues. Most algorithms for this task utilize Lanczos' iteration, convergence bounds of which depends on the input matrix' spectrum (Saad, 1980; LI, 2010), which although have superlinear convergence, but are observed to be efficient in practice. We compute node embeddings afresh whenever the model parameters are updated. This computation involves matrix multiplication with eigenmatrices incurring an additional cost (over GPR-GNN) of $O(nkL)$ where $k$ and $L$ denote the number of selected low/high eigenvalues and classes, respectively.

### 4.3 ANALYSIS

We first give a simple theoretical justification for using multiple polynomials. We upper-bound the approximation error achieved by using multiple polynomials by the error that the single polynomial parameterization achieves. We relegate the proofs of all the theorems of this section to the appendix.

**Theorem 4.1.** For any frequency response $h^*$, and an integer $K \in \mathbb{N}$, let $\tilde{h} := h + h_f$, with $h_f$ having a continuous support over a subset of the spectrum, $\sigma_f$. Assume that $h$ and $h_f$ are parameterized by independent $K$ and $K'$-order polynomials, $p$ and $p_f$, respectively, with $K' \leq K$. Then there exists $\tilde{h}$, such that $\min \|\tilde{h} - h^*\|_2 \leq \min \|h - h^*\|_2$, where the minimum is taken over the polynomial parameterizations. Moreover, for multiple polynomial adaptive filters $h_{f_1}, h_{f_2}, ..., h_{f_m}$

---

[1]For brevity, we dropped the polynomial degree dependency.

parameterized by independent $K'$-degree polynomials with $K' \leq K$ but having disjoint, contiguous supports, the same inequality holds for $\tilde{h} = h + \sum_{i=1}^{m} h_{f_i}$.

For a detailed proof please refer to A.4.3. We constructed waveforms of arbitrary complexity, approximated by first fitting a single globally-defined polynomial and then adding locally defined polynomials on top of it. We assess the performance by varying the degrees of the local polynomials. The corresponding plots (See Figure 8) indicate that use of lower-order filters achieves a fit comparable to one by a relatively higher-degree polynomial. Since an actual waveform is not observed in practice and instead, we estimate it by optimizing over the observed labels via learning a graph filter, we theoretically show that the family of filters that we learn is a strict superset of the polynomial filter family. The same result holds for the families of the resulting adapted graphs.

**Theorem 4.2.** Define $\mathbb{H} := \{h(\cdot) \mid \forall \text{ possible K-degree polynomial parameterizations of } h\}$ to be the set of all K-degree polynomial filters, whose arguments are $n \times n$ diagonal matrices, such that a filter response over some $\mathbf{\Lambda}$ is given by $h(\mathbf{\Lambda})$ for $h(\cdot) \in \mathbb{H}$. Similarly $\mathbb{H}' := \{\tilde{h}(\cdot) \mid \forall \text{ possible polynomial parameterizations of } \tilde{h}\}$ is set of all filters learn-able via PP-GNN , with $\tilde{h} = h + h_{f_1} + h_{f_2}$, where $h$ is parameterized by a K-degree polynomial supported over entire spectrum, $h_{f_1}$ and $h_{f_2}$ are adaptive filters parameterized by independent $K'$-degree polynomials which only act on top and bottom $t$ diagonal elements respectively, with $t < n/2$ and $K' \leq K$; then $\mathbb{H}$ and $\mathbb{H}'$ form a vector space, with $\mathbb{H} \subset \mathbb{H}'$. Also, $\frac{\dim(\mathbb{H}')}{\dim(\mathbb{H})} = \frac{K+2K'+3}{K+1}$.

**Corollary 4.2.1.** The corresponding adapted graph families $\mathbb{G} := \{\mathbf{U}^T h(\cdot) \mathbf{U}^T \mid \forall h(\cdot) \in \mathbb{H}\}$ and $\mathbb{G}' := \{\mathbf{U}^T \tilde{h}(\cdot) \mathbf{U}^T \mid \forall \tilde{h}(\cdot) \in \mathbb{H}'\}$ for any unitary matrix $\mathbf{U}$ form a vector space, with $\mathbb{G} \subset \mathbb{G}'$ and $\frac{\dim(\mathbb{G}')}{\dim(\mathbb{G})} = \frac{K+2K'+3}{K+1}$.

This implies that our model is learning from a more diverse space of filters and the corresponding adapted graphs. Moreover, the dimension of the space increases significantly by using just two adaptive filters. Note that learning from such a diverse region is feasible. This observation is possible from the proofs of Theorem 4.2 (A.4.4) and Corollary 4.2.1 (A.4.5). Using the adaptive filters without any filter with the entire spectrum as support results in learning a set of adapted graphs, $\hat{\mathbb{G}}$. This set is disjoint from $\mathbb{G}$, with $\mathbb{G}' = \mathbb{G} \oplus \hat{\mathbb{G}}$. We conduct various ablative studies where we demonstrate the effectiveness of learning from $\hat{\mathbb{G}}$ and $\mathbb{G}'$.

## 5 EXPERIMENTS

We conduct comprehensive experiments to demonstrate the effectiveness and competitiveness of the proposed PP-GNN model over state-of-the-art (SOTA) models/methods and answer the following research questions.

1. **[RQ1]** Does the PP-GNN model outperform SOTA models/methods over a broad range of datasets (homophilic/heterophilic)?
2. **[RQ2]** Does the proposed model learn better and complex frequency responses that improve performance [Frequency response plots]?
3. **[RQ3]** With the PP-GNN model comprising of multiple filters, how does each filter contribute to achieving improved performance?
4. **[RQ4 and RQ5]** Does the model learn better embedding, and do we need a large number of eigenvalues/vectors and polynomials to get improved performance?
5. **[RQ6]** How does the training time of the PP-GNN and GPR-GNN models compare?

**Datasets and Setup.** We evaluate our model on several real-world heterophilic and homophilic datasets including a few large graphs for the node classification task. Detailed statistics of the benchmark datasets are provided in the appendix (A.5.1). The heterophilic datasets include **Texas**[2], **Cornell**[2], **Wisconsin**[2], **Chameleon**, **Squirrel**, (Rozemberczki et al., 2021) and **Flickr**. We use the 10 random splits (48%/32%/20% of nodes for train/validation/test set) from Pei et al. (2020). For **Flickr**, **Ogbn-Arxiv**, and **Wiki-CS**, we use the data splits from Kim & Oh (2021). We also

---

[2]http://www.cs.cmu.edu/afs/cs.cmu.edu/project/theo-11/www/wwkb

evaluate on 8 homophilic datasets: **Cora-Full**, **Ogbn-Arxiv**, **Wiki-CS**, **Citeseer**, **Pubmed**, **Cora**, **Computer**, and **Photos** borrowed from Kim & Oh (2021). For the remaining homophilic datasets, we create 10 random splits for each dataset following Kim & Oh (2021). We use PCA node features (Kim & Oh (2021)) for the **Chameleon** and **Cora-Full** datasets. We report mean and standard deviation of test accuracy over splits to compare model performance.

**Baselines.** We compare our model against (a) filtering based approaches, GPR-GNN, FAGCN, APPNP, and LGC, (b) other state-of-the-art models include SUPERGAT, TDGNN, $H_2$GCN, and GEOM-GCN, and (c) standard baselines, namely, LR (Logistic Regression), MLP (Multi Layer Perceptron), GCN, and SGCN. Detailed descriptions for all the baselines, hardware and software specifications are in the appendix (A.5.2 and A.5.5)

**Hyperparameter Tuning.** For the PP-GNN model, we separately partition the low-end and high-end eigenvalues into several contiguous partitions and use shared filter parameters for frequencies of each partition. The number of partitions, which can be interpreted as the number of filters, is swept in the range [2,3,4,5,10,20]. The polynomial filter order is swept in the range [1,10] in steps of size 1. The number of eigenvalues/vectors are swept in the range [32, 64, 128, 256, 512, 1024]. In our experiments, we set $\eta_l = \eta_h$ and we vary them in range (0, 1) and set $\eta_{gpr} = 1 - \eta_l$. We did hyperparameter tuning for other models as suggested in respective references and public repositories. More details are in the appendix (A.5.5).

| Test Acc | Texas | Wisconsin | Squirrel | Chameleon | Cornell | Flickr |
|---|---|---|---|---|---|---|
| **LR** | 81.35 (6.33) | 84.12 (4.25) | 34.73 (1.39) | 45.68 (2.52) | 83.24 (5.64) | 46.51 |
| **MLP** | 81.24 (6.35) | 84.43 (5.36) | 35.38 (1.38) | 51.64 (1.89) | 83.78 (5.80) | 46.93 |
| **SGCN** | 62.43 (4.43) | 55.69 (3.53) | 45.72 (1.55) | 60.77 (2.11) | 62.43 (4.90) | 50.75 |
| **GCN** | 61.62 (6.14) | 58.82 (4.89) | 47.78 (2.13) | 62.83 (1.52) | 62.97 (5.41) | 53.4 |
| **SuperGAT** | 61.08 (4.97) | 56.47 (3.90) | 31.84 (1.26) | 43.22 (1.71) | 57.30 (8.53) | 53.47 |
| **Geom-GCN** | 67.57* | 64.12* | 38.14* | 60.90* | 60.81* | NA |
| **H2GCN** | 84.86 (6.77)* | 86.67 (4.69)* | 37.90 (2.02)* | 58.40 (2.77) | 82.16 (4.80)* | OOM |
| **FAGCN** | 82.43 (6.89) | 82.94 (7.95) | 42.59 (0.79) | 55.22 (3.19) | 79.19 (9.79) | OOM |
| **APPNP** | 81.89 (5.85) | 85.49 (4.45) | 39.15 (1.88) | 47.79 (2.35) | 81.89 (6.25) | 50.33 |
| **LGC** | 80.20 (4.28) | 81.89 (5.98) | 44.26 (1.49) | 61.14 (2.07) | 74.59 (3.42) | 51.67 |
| **GPR-GNN** | 81.35 (5.32) | 82.55 (6.23) | 46.31 (2.46) | 62.59 (2.04) | 78.11 (6.55) | 52.74 |
| **TDGNN** | 83.00 (4.50)* | 85.57 (3.78)* | 43.84 (2.16) | 55.20 (2.30) | 82.92 (6.61)* | OOM |
| **PP-GNN** | **89.73 (4.90)** | **88.24 (3.33)** | **56.86 (1.20)** | **67.74 (2.31)** | **82.43 (4.27)** | **55.17** |

Table 1: Results on Heterophilic Datasets. We underline the results for the best performing baseline model. '*' indicates that the results were borrowed from the corresponding papers.

| Test Acc | Cora-Full | OGBN-ArXiv | Wiki-CS | Citeseer | Pubmed | Cora | Computer | Photos |
|---|---|---|---|---|---|---|---|---|
| **LR** | 39.10 (0.43) | 52.53 | 72.28 (0.59) | 72.22 (1.54) | 87.00 (0.40) | 73.94 (2.47) | 64.92 (2.59) | 77.57 (2.29) |
| **MLP** | 43.03 (0.82) | 54.96 | 73.74 (0.71) | 73.83 (1.73) | 87.77 (0.27) | 77.06 (2.16) | 64.96 (3.57) | 76.96 (2.46) |
| **SGCN** | 61.31 (0.78) | 68.51 | 78.30 (0.75 | 76.77 (1.52) | 88.48 (0.45) | 86.96 (0.78) | 80.65 (2.78) | 89.99 (0.69) |
| **GCN** | 59.63 (0.86) | 69.37 | 77.64 (0.49) | 76.47 (1.33) | 88.41 (0.46) | 87.36 (0.91) | 82.50 (1.23) | 90.67 (0.68) |
| **SuperGAT** | 57.75 (0.97) | 55.1* | 77.92 (0.82) | 76.58 (1.59) | 87.19 (0.50) | 86.75 (1.24) | 83.04 (1.02) | 90.31 (1.22) |
| **Geom-GCN** | NA | NA | NA | 77.99* | 90.05* | 85.27* | NA | NA |
| **H2GCN** | 57.83 (1.47) | OOM | OOM | 77.07 (1.64)* | 89.59 (0.33)* | 87.81 (1.35)* | OOM | 91.17 (0.89) |
| **FAGCN** | 60.07 (1.43) | OOM | 79.23 (0.66) | 76.80 (1.63) | 89.04 (0.50) | 88.21 (1.37) | 82.16 (1.48) | 90.91 (1.11) |
| **APPNP** | 60.83 (0.55) | 69.2 | 79.13 (0.50) | 76.86 (1.51) | 89.57 (0.53) | 88.13 (1.53) | 82.03 (2.04) | 91.68 (0.62) |
| **LGC** | 61.84 (0.90) | 69.64 | 79.82 (0.49) | 76.96 (1.73) | 88.78 (0.51) | 88.02 (1.44) | 83.44 (1.77) | 91.56 (0.74) |
| **GPR-GNN** | 61.37 (0.96) | 68.44 | 79.68 (0.50) | 76.84 (1.69) | 89.08 (0.39) | 87.77 (1.31) | 82.38 (1.60) | 91.43 (0.89) |
| **TDGNN** | OOM | OOM | 79.58 (0.51) | 76.64 (1.54)* | 89.22 (0.41)* | 88.26 (1.32)* | 84.52 (0.92) | 92.54 (0.28) |
| **PP-GNN** | **61.42 (0.79)** | **69.28** | **80.04 (0.43)** | **78.25 (1.76)** | **89.71 (0.32)** | **89.52 (0.85)** | **85.23 (1.36)** | **92.89 (0.37)** |

Table 2: Results on Homophilic Datasets.

## 5.1 RQ1: PP-GNN VERSUS SOTA MODELS

We present several important observations from Tables 1 and 2. We observe that the PP-GNN model consistently outperforms all models, including recent filtering approach based models on all datasets (both heterophilic and homophilic) except Pubmed and Cornell, where it achieves similar performance. This result demonstrates the effectiveness and robustness of our model across a wide variety of datasets. Furthermore, compared with the GPR-GNN model, learning piece-wise polynomial filters improves performance significantly over learning a single polynomial filter. In particular, PP-GNN achieves performance improvements of around 10% and 5% on the Chameleon and Squirrel datasets.

## 5.2 RQ2: ADAPTABLE FREQUENCY RESPONSES

We computed the frequency responses (i.e., $h(\lambda)$) of learned polynomials of the PP-GNN and GPR-GNN models on several datasets, including Squirrel and Citeseer datasets shown in Figure 2. We observe from Figure 2a, though the GPR-GNN model can learn some variations at low/high frequencies, it is insufficient to achieve higher classification accuracy. On the other hand, the PP-GNN model can capture complex shapes at the low and high ends of the spectrum, enabling it to achieve significantly improved test accuracy. We observed a similar phenomenon for the chameleon dataset as well. To illustrate another behaviour, we present the frequency responses for the Citeseer dataset in Figure 2b, and the responses are similar except for some variations at the low-end of the spectrum. Note that the GPR-GNN model does well on several homophilic datasets. These observations suggest that the PP-GNN model adapts very well in learning desired frequency responses, as dictated by the task at hand. We can observe such a behaviour for two other datasets in the appendix (A.5.6).

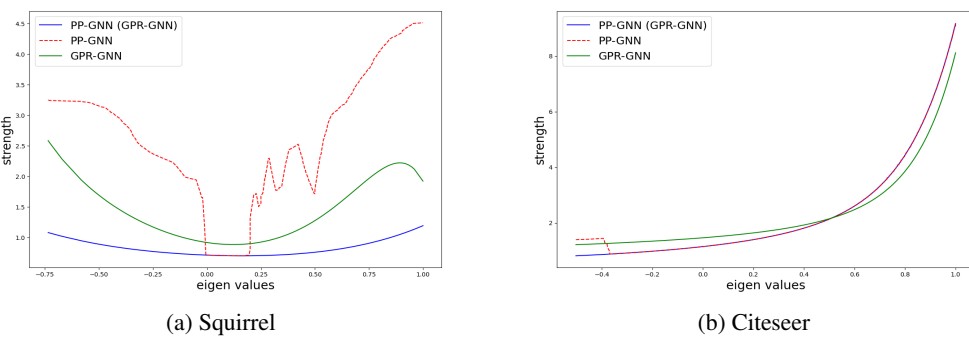

|     |     |
| :-: | :-: |
| (a) Squirrel | (b) Citeseer |

Figure 2: Visualization of adapted eigenspectrum by our proposed model and GPR-GNN

## 5.3 RQ3: PERFORMANCE COMPARISON OF DIFFERENT FILTERS

Recall that the PP-GNN model is a sum of polynomials model comprising of polynomial filters operating at different parts of the spectrum. Here, we study the importance and effect of using a combination of filters operating on different regions. For reference, we also compare the performance of GPR-GNN model that operates on the entire spectrum using a single filter. Several interesting observations are in order. From Table 3, we see that the best performance is achievable using high-frequency signals alone for the heterophilic datasets (Squirrel and Chameleon), suggesting that significant discriminatory information is available at high frequencies compared to low-frequency signals. In contrast, homophilic datasets (Cora and Citeseer) exhibit a reverse trend. On comparing the first (second) and third (fourth) row results, we see that having the GPR-GNN filter as part of the PP-GNN filter helps to get improved performance over individual filters (PP-GNN(Low) or PP-GNN (High)). Finally, the PP-GNN model can adapt well, capture contrasting information bands across datasets, and outperform the GPR-GNN model.

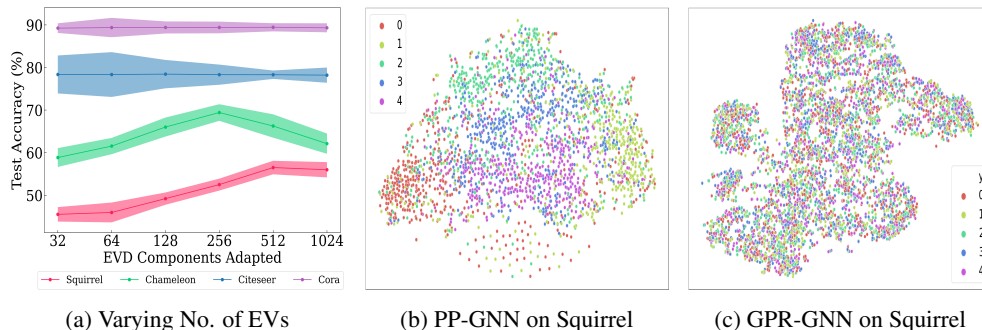

| (a) Varying No. of EVs | (b) PP-GNN on Squirrel | (c) GPR-GNN on Squirrel |

Figure 3: Analyzing varying number of eigenvalues and the learned embeddings

| Test Acc | Squirrel | Chameleon | Citeseer | Cora |
|---|---|---|---|---|
| **PP-GNN (Low)** | 45.75 (1.69) | 56.73 (4.03) | 76.23 (1.54) | 88.03 (0.79) |
| **PP-GNN (High)** | 58.70 (1.60) | **69.19 (1.88)** | 55.50 (6.38) | 73.76 (2.03) |
| **PP-GNN (GPR-GNN+Low)** | 50.96 (1.26) | 63.71 (2.69) | 78.07 (1.71) | **89.56 (0.93)** |
| **PP-GNN (GPR-GNN + High)** | 60.39 (0.91) | 67.83 (2.30) | **78.30 (1.60)** | 89.42 (0.97) |
| **GPR-GNN** | 42.06 (1.55) | 56.29 (1.58) | 76.74 (1.33) | 87.93 (1.52) |
| **PP-GNN** | **56.21 (1.79)** | 68.93 (1.95) | 78.25 (1.76) | 89.52 (0.85) |

Table 3: Performance of different filters

## 5.4 RQ4 - RQ6: VARYING NO. OF EVS, LEARNED EMBEDDING AND TIMING COMPARISON

We conducted ablative studies to see the effect of varying different hyperparameters of the PP-GNN model. From Figure 3a, we see that as small as 32 eigen components are sufficient to achieve near-best performance on the homophilic datasets (Cora, Citeseer). On the other hand, peak performance is achieved for some intermediate ($\approx$250-500) number of eigen components for the Squirrel and Chameleon datasets. Also, we note that we can achieve comparable performance to state of the art models on large datasets (Flickr and OGBN-Arxiv) using only 1024 components (See Tables 1 and 2). This study suggests that the PP-GNN model is scalable at least for medium sized graphs ( 100K nodes). More evaluation and investigation are needed to study performance versus computation cost trade-off for very large graphs (millions of nodes).

To assess the quality of learned embedding, we also created t-SNE plots and made a visual inspection. From Figure 3b and 3c, we see that the clusters belonging to different classes of the Squirrel dataset are better separated for the PP-GNN  model compared to the GPR-GNN  model. We also measured the time taken by the GPR-GNN and PP-GNN models. We observed that the GPR-GNN model is only $(1.2 - 2)\times$ faster than the PP-GNN model (using 1024 eigencomponents) on many datasets. Thus, the extra training cost incurred by the PP-GNN model is not significantly different for practical purposes. More details are in the appendix (A.6).

## 6 CONCLUSION

Several proposed models attempt to be robust to the correlations between graph and node labels. We build on the filter-based approach of GPR-GNN (Chien et al., 2021). This work proposed an effective polynomial filter design. We combine GPR-GNN with additional polynomials that adapt specifically to low and high-frequency components. Our experiments demonstrate that such an approach can learn filter functions that improve performance on the node classification task. Our plots of these filter functions suggest that they are of high order on several datasets. It would be interesting to analyze these filter plots and identify some common characteristics. These will enable us to a] characterize the correlation of graphs and labels, b] further improve the performance and c] build robust graph privacy models. We plan to do this as our future work.

## 7 REPRODUCIBILITY STATEMENT

We have taken several steps to ensure the reproducibility of our work. First, we provide a detailed description of our model in Section 4. Section 5 provides a brief description of the datasets utilized in our experiments. A detailed description of the datasets, including their sources, statistics are given in the appendix (A.5.1). We also provide details on the number of splits and the method used to generate them. We also list out all the baselines utilized in the experiments in Section 5. A detailed description of these baselines and the parameter sweep ranges are given in the appendix (A.5.2). We also provide implementation detail of our model in the appendix (A.5.5). Additionally, we share an anonymized URL of the code for our model, and the data splits in the appendix (A.1).

## 8 ETHICS STATEMENT

Social network graphs form the most exciting application for GNNs. GNNs can be used to reveal user information that the user otherwise would have preferred to keep private. Privacy-preserving methods might rely on obfuscating the graph by adding spurious edges, and these edges will, in turn, reduce the graph's correlation with the node labels. Earlier GNNs would likely not have predicted the target label with reasonable accuracy under this setting. Thereby, such obfuscation methods would give some level of privacy for the users. However, our model exhibits good performance even in the presence of a lower correlation. Such methods can make simple obfuscation based privacy approaches obsolete while still revealing important information about the users. On the other hand, our model can also give insights into what makes graphs reveal certain information and develop a more robust privacy model for graphs.

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

## A  APPENDIX

The appendix is structured as follows. In Section A.1, we provide a URL with our code and data, along with some information about the contents in the URL. In Section A.2, we present additional evidence of the limitations of GPR-GNN. In Section A.3, we show a representative experiment that motivates Section 4. In Section A.4, we provide proofs for theorems and corollaries defined in Section 4.3. In Section A.5, we provide more details regarding the baselines, datasets and their respective splits. We also provide implementation details and rank PP-GNN against current SoTA. In Section A.6, we perform a timing analysis where we compare PP-GNN to GPR-GNN.

### A.1  CODE AND DATASET

Our code, along with the datasets and their respective splits, is available at https://tinyurl.com/PPGNN. A README file is available explaining how the code can be executed on various datasets. The requirements.txt file lists all the necessary python packages required to execute the code. The src folder also contains bash scripts to run the code on various datasets.

### A.2  MOTIVATION

#### A.2.1  NODE FEATURE INDISTINGUISHABLY PLOTS

In the main paper (Figure 1a), we plot the average of pairwise distances between node features for the Cora dataset, after computing $\widetilde{\mathbf{A}}^j X$ for increasing $j$ values, and showed that the mean pairwise node feature distance decreases as $j$ increases. We observe that this is consistent across three more datasets: Citeseer, Chameleon and Squirrel. This is observed in Figure 4.

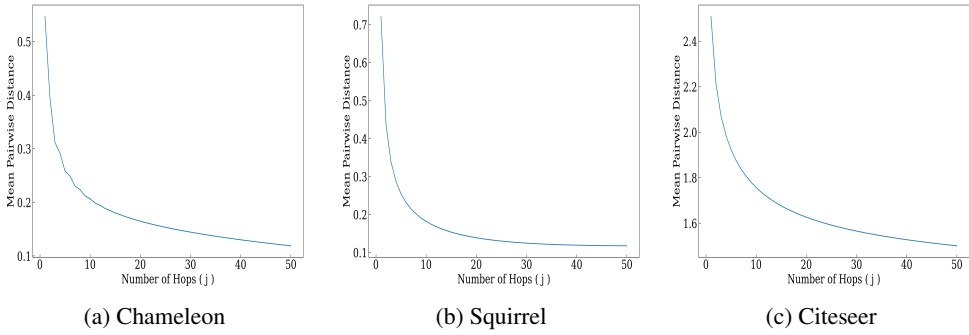

(a) Chameleon               (b) Squirrel               (c) Citeseer

Figure 4: Average of pairwise distances between node features, after computing $\widetilde{\mathbf{A}}^j X$, for increasing $j$ values

We also observed the mean of the variance of each dimension of node features, after computing $\widetilde{\mathbf{A}}^j X$, for increasing $j$ values. We observe that this mean does indeed reduce as the number of hops increase. We also observe that the variance of each dimension of node features reduces for Cora, Squirrel and Chameleon as the number of hops increase; however, we don't observe such an explicit phenomenon for Citeseer. See Figure 5.

#### A.2.2  EFFECT OF VARYING THE ORDER OF THE GPR-GNN POLYNOMIAL

In the main paper (Figure 1a), we plot the test accuracies of the GPR-GNN model while increasing the order of the polynomials for the Cora and Chameleon dataset, respectively. We observe that on increasing the polynomial order, the accuracies do not increase any further. We can show a similar phenomenon on two other datasets, Squirrel and Citeseer, in Figure 6.

In Section 3 of the main paper, we claim that due to the over-smoothing effect, even on increasing the order of the polynomial, there is no improvement in the test accuracy. Moreover, in Figure 2 we can see that our model can learn a complicated filter polynomial while GPR-GNN cannot. This section shows that even on increasing the order of the GPR-GNNpolynomial, neither does the test accuracy increase nor does the waveform become as complicated as PP-GNN. See Figure 7.

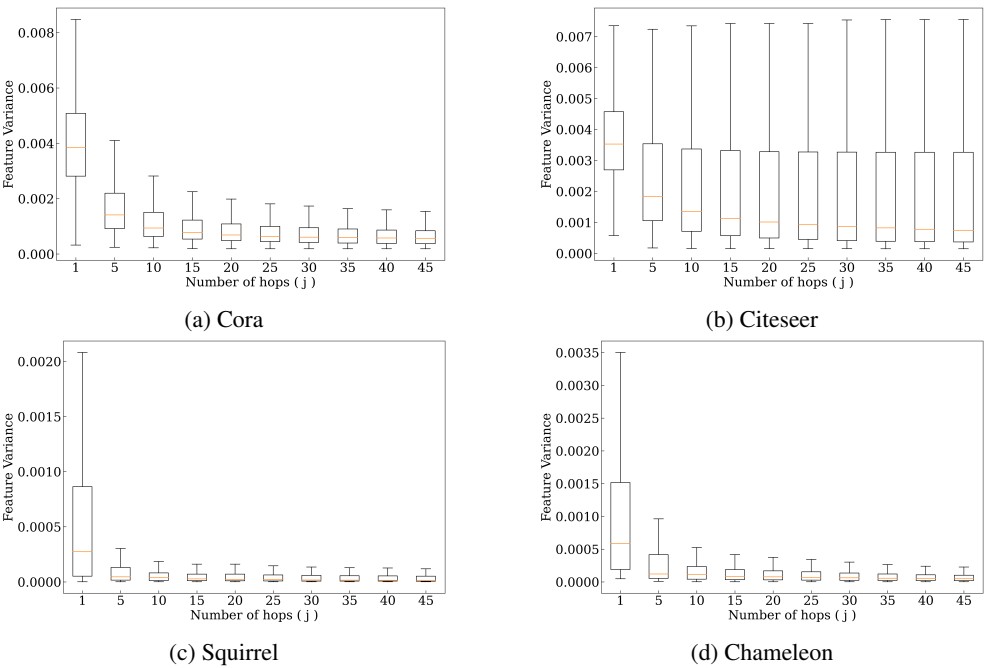

Figure 5: Variance of each dimension of node features

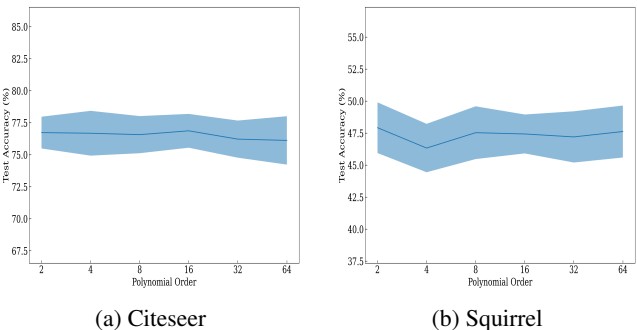

Figure 6: Accuracy of the GPR-GNN model on inceasing the order of the polynomial

## A.3 FICTITIOUS POLYNOMIAL

In Section 4, we claim that having multiple disjoint low order polynomials can approximate a complicated waveform better than a higher-order polynomial. We create a representative experiment that shows this is indeed true by creating a fictitious complicated polynomial and trying to fit it using both a single polynomial and the disjoint multi polynomial. We observe in Figure 8 that the lower order disjoint polynomial fits the complicated waveform better than a higher-order single polynomial.

## A.4 PROPOSED APPROACH

### A.4.1 DETAILS REGARDING BOUNDARY REGULARIZATION

To induce smoothness in the learned filters, we add a regularization term that penalizes squared differences between function values of polynomials at knots (endpoints of contiguous bins). Our regularization term looks as follows:

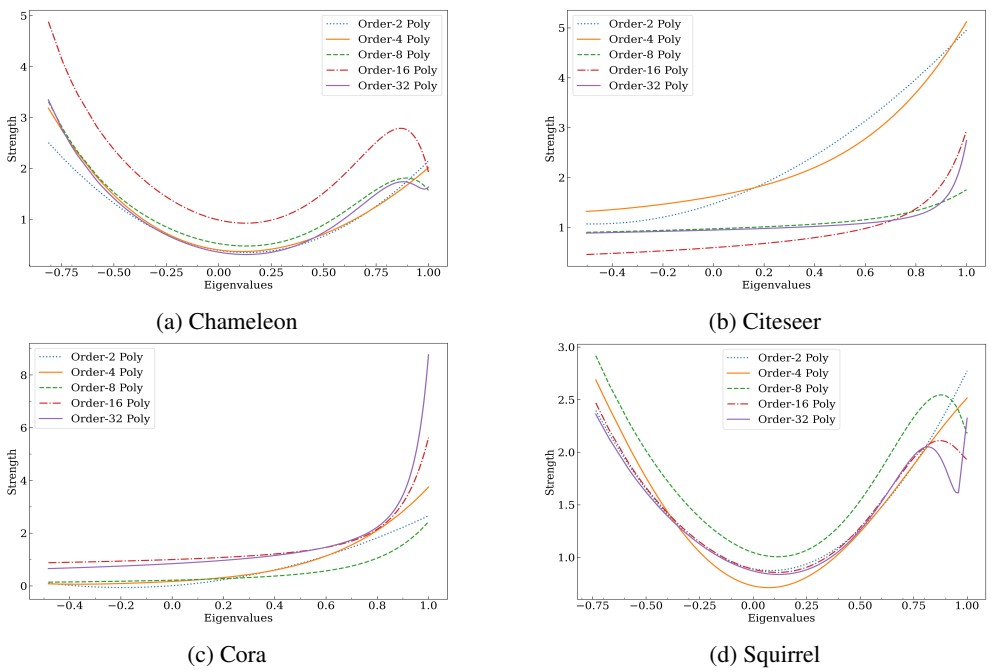

(a) Chameleon        (b) Citeseer

(c) Cora        (d) Squirrel

Figure 7: Varying Polynomial Order in GPR-GNN

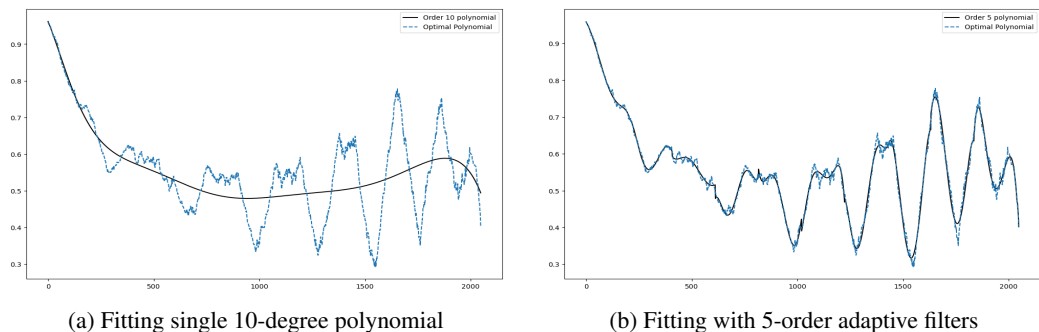

(a) Fitting single 10-degree polynomial        (b) Fitting with 5-order adaptive filters

Figure 8: To demonstrate the adaptability of adaptive polynomial filter, we try to approximate a complex waveform (blue dashed line) via (a) a single 10 degree polynomial, and (b) a 10-degree polynomial with 10 adaptive filters of order 5. The waveform learnt via the filter is shown in a solid dashed line. The corresponding RMSE are: (a) 3.9011, (b) 0.2800

$$\sum_{i=1}^{m-1} \exp^{-(\sigma_i^{max} - \sigma_{i+1}^{min})^2} (h_i(\sigma_i^{max}) - h_{i+1}(\sigma_{i+1}^{min}))^2 \tag{5}$$

In equation 5, $\sigma_i^{max}$ and $\sigma_i^{min}$ refer to the maximum and minimum eigenvalues in $\sigma_i$ (Refer to Section 4). This regularization term is added to the Cross-Entropy loss. We perform experiments with this model and report the performance in Table 4. We observe that we are able to reach similar performance even without the presence of this regularization term. Therefore, majority of the results reported in our Main paper are without this regularization term.

### A.4.2 NOTATION USED

Vectors are denoted by lower case bold Roman letters such as $\mathbf{x}$, and all vectors are assumed to be column vectors. In the paper, $h$ with any sub/super-script refers to a frequency response, which is

| Test Acc | Computer | Chameleon | Citeseer | Cora | Squirrel |
|---|---|---|---|---|---|
| PPGNN (With Reg) | 83.53 (1.67) | 67.92 (2.05) | 76.85 (2) | 88.19 (1.19) | 55.42 (2.1) |
| PPGNN | 85.23 (1.36) | 67.74 (2.31) | 76.74 (1.33) | 89.52 (0.85) | 56.86 (1.20) |

Table 4: Results with and without boundary regularization

also considered to be a vector. A superscript $T$ denotes the transpose of a matrix or vector; Matrices are denoted by bold Roman upper case letters, such as $\mathbf{M}$. A field is represented by $\mathbb{K}$; sets of real and complex numbers are denoted by $\mathbb{R}$ and $\mathbb{C}$ respectively. $\mathbb{K}[x_1, \ldots, x_n]$ denotes a multivariate polynomial ring over the field $\mathbb{K}$, in indeterminates $x_1, \ldots, x_n$. Set of $n \times n$ square matrices with entries from some set $\mathbb{S}$ are denoted by $\mathbb{M}_n(\mathbb{S})$. Moore-Penrose pseudoinverse of a matrix $\mathbf{A}$ is denoted by $\mathbf{A}^\dagger$. Eigenvalues of a matrix are denoted by $\lambda$, with $\lambda_1, \lambda_2, \ldots$ denoting a decreasing order when the eigenvalues are real. A matrix $\mathbf{\Lambda}$ denotes a diagonal matrix of eigenvalues. Set of all eigenvalues, i.e., spectrum, of a matrix is denoted by $\sigma_\mathbf{A}$ or simply $\sigma$ when the context is clear. $L_p$ norms are denoted by $\|\cdot\|_p$. Frobenius norm over matrices is denoted by $\|\cdot\|_F$. Norms without a subscript default to $L_2$ norms for vector arguments and Frobenius norm for matrices. $\oplus$ denotes a direct sum. For maps $f_i$ defined from the vector spaces $V_1, \cdots, V_m$, with a map of the form $f : V \mapsto W$, with $V = V_1 \oplus V_2 \oplus \cdots \oplus V_m$, the the phrase "$f : V \mapsto W$ by mapping $f(\mathbf{v_i})$ to $f_i(g(\mathbf{v_i}))$" means that $f$ maps a vector $\mathbf{v} = \mathbf{v_1} + \ldots + \mathbf{v_m}$ with $\mathbf{v_i} \in V_i$ to $f_1(g(\mathbf{v_1})) + \ldots + f_m(g(\mathbf{v_m}))$

### A.4.3   PROOF OF THEOREM 4.1

**Theorem.** For any desired frequency response $h^*$, and an integer $K \in \mathbb{N}$, let $\tilde{h} := h + h_f$, with $h_f$ having a continuous support over a subset of the spectrum, $\sigma_f$. Assuming $h$ and $h_f$ to be parameterized by independent $K$ and $K'$-order polynomials $p$ and $p_f$ respectively, with $K' \leq K$, then there exists $\tilde{h}$, such that $\min \|\tilde{h} - h^*\|_2 \leq \min \|h - h^*\|_2$, where the minimum is taken over the polynomial parameterizations. Moreover, for multiple polynomial adaptive filters $h_{f_1}, h_{f_2}, ..., h_{f_m}$ parameterized by independent $K'$-degree polynomials with $K' \leq K$ but having disjoint, contiguous supports, the same inequality holds for $\tilde{h} = h + \sum_{i=1}^m h_{f_i}$.

*Proof.* We make the following simplifying assumptions:

1. $|\sigma_{f_i}| > K, \quad \forall i \in [m]$, i.e., that is all support sizes are lower bounded by $K$ (and hence $K'$)

2. All eigenvalues of the reference matrix are distinct

For methods that use a single polynomial filter, the polynomial graph filter, $h_K(\mathbf{\Lambda}) = diag(\mathbf{V}\gamma)$ where $\gamma$ is a vector of coefficients (i.e, $\gamma$ parameterizes $h$), with eigenvalues sorted in descending order in components, and $\mathbf{V}$ is a Vandermonde matrix:

$$\mathbf{V} = \begin{bmatrix} 1 & \lambda_1 & \lambda_1^2 & \cdots & \lambda_1^K \\ 1 & \lambda_2 & \lambda_2^2 & \cdots & \lambda_2^K \\ \vdots & \vdots & \vdots & \ddots & \vdots \\ 1 & \lambda_n & \lambda_n^2 & \cdots & \lambda_n^K \end{bmatrix}$$

And to approximate a frequency response $h^*$, we have the following objective:

$$\min \|h - h^*\|_2^2 := \min_\gamma \|diag(h^*) - diag(\mathbf{V}\gamma)\|_F^2 = \min_\gamma \|h^* - \mathbf{V}\gamma\|_2^2 = \min_\gamma \|\mathbf{e}_p(\gamma)\|_2^2$$

Where $\|\|_F$ and $\|\|_2$ are the Frobenius and $L_2$ norms respectively. Due to the assumptions, the system of equations $h^* = \mathbf{V}\gamma$ is well-defined and has a unique minimizer, $\gamma^* = \mathbf{V}^\dagger h^*$, and thus

$\|\mathbf{e}_p(\gamma^*)\| = \min_\gamma \|\mathbf{e}_p(\gamma)\|$. Next we break this error vector as:

$$\mathbf{e}_p(\gamma^*) := h^* - \mathbf{V}\gamma^*$$
$$= \sum_{i=1}^m (h_i^* - \mathbf{V}_i\gamma^*) + (h_L^* - \mathbf{V}_L\gamma^*)$$
$$:= \sum_{i=1}^m \mathbf{e}_{p_i}^* + \mathbf{e}_{p_L}^*$$

Where $\mathbf{e}_{p_i}^* := (h_i^* - \mathbf{V}_i\gamma^*)$ with similar definition for $\mathbf{e}_{p_L}$; $h_i^*$ is a vector whose value at components corresponding to the set $\sigma(h_{f_i})$ is same as that of $h^*$ and rest are zero. Similarly, $\mathbf{V}_i^*$ is a matrix whose rows corresponding to the set $\sigma(h_{f_i})$ are same as that of $\mathbf{V}$ with other rows being zero. Also, $\mathbf{V}_L = \mathbf{V} - \sum_{i=0}^m \mathbf{V}_i$ and $h_L^* = h^* - \sum_{i=0}^m h_i^*$. Note that as a result of this construction, $[\mathbf{e}_{p_i}^*] \cup \mathbf{e}_{p_L}^*$ is a linearly independent set since the supports $[\sigma(h_{f_i})]$ form a disjoint set (note the theorem statement). We split the proof in two cases:

**Case 1:** $K' = K$. We now analyze the case where we have $m$ polynomial adaptive filters added, all having an order of $K$, where the objective is $\min \|\tilde{h} - h^*\|$, which can be written as:

$$\min_{\gamma, [\gamma_i]} \left\| diag(h^*) - diag\left(\mathbf{V}\gamma + \sum_{i=0}^m \mathbf{V}_i\gamma_i\right) \right\|_F^2 = \min_{\gamma, [\gamma_i]} \left\| h^* - \mathbf{V}\gamma - \sum_{i=0}^m \mathbf{V}_i\gamma_i \right\|_2^2 = \min_{\gamma, [\gamma_i]} \|\mathbf{e}_g(\gamma, [\gamma_i])\|_2^2$$

Before characterizing the above system, we break a general error vector as:

$$\mathbf{e}_g(\gamma, [\gamma_i]) := h^* - \mathbf{V}\gamma - \sum_{i=0}^m \mathbf{V}_i\gamma_i$$
$$= \sum_{i=1}^m (h_i^* - \mathbf{V}_i(\gamma + \gamma_i)) + (h_L^* - \mathbf{V}_L\gamma)$$
$$:= \sum_{i=1}^m \mathbf{e}_{g_i} + \mathbf{e}_{g_L}$$

Where $\mathbf{e}_{g_i} := (h_i^* - \mathbf{V}_i(\gamma + \gamma_i))$ with similar definition for $\mathbf{e}_{g_L}$. Clearly, the systems of equations, $\mathbf{e}_{g_i} = 0, \ \forall i$ and $\mathbf{e}_{g_L} = 0$ are well-defined due to the assumptions 1 and 2. Since all the systems of equations have independent argument, unlike in the polynomial filter case where the optimization is constrained over a single variable; one can now resort to individual minimization of squared norms of $\mathbf{e}_{g_i}$ which results in a minimum squared norm of $\mathbf{e}_g$. Thus, we can set:

$$\gamma = \mathbf{V}_L^\dagger h_L^* = \gamma_g^* \qquad \gamma_i = \mathbf{V}_i^\dagger h_i^* - \mathbf{V}_L^\dagger h_L^* = \gamma_i^*, \ \forall i \in [m]$$

to minimize squared norms of $\mathbf{e}_{g_i}$ and $\mathbf{e}_{g_L}$. Note that $[\mathbf{e}_{g_i}] \cup \mathbf{e}_{g_L}$ is a linearly independent set since the supports $[\sigma(h_{f_i})]$ form a disjoint set and by the above construction, this is also an orthogonal set, and hence we have $\|\mathbf{e}_g\|^2 = \sum_{i=1}^m \|\mathbf{e}_{g_i}\|^2 + \|\mathbf{e}_{g_L}\|^2$, and hence the above assignment implies:

$$\|\mathbf{e}_g(\gamma_g^*, [\gamma_i^*])\| = \min_{\gamma, [\gamma_i]} \|\mathbf{e}_g(\gamma_g, [\gamma_i])\| := \min \|\tilde{h} - h^*\|_2$$

Hence, it follows that, $\min_x \|h_i^* - \mathbf{V}_i x\|^2 = \left\|\mathbf{e}_{g_i}^*\right\|^2 \le \left\|\mathbf{e}_{p_i}^*\right\|^2 = \|h_i^* - \mathbf{V}_i\gamma^*\|^2$ and $\min_x \|h_L^* - \mathbf{V}_L x\|^2 = \left\|\mathbf{e}_{g_L}^*\right\|^2 \le \left\|\mathbf{e}_{p_L}^*\right\|^2 = \|h_L^* - \mathbf{V}_L\gamma^*\|^2$. Hence,

$$\sum_{i=1}^m \left\|\mathbf{e}_{g_i}^*\right\|^2 + \left\|\mathbf{e}_{g_L}^*\right\|^2 \le \sum_{i=1}^m \left\|\mathbf{e}_{p_i}^*\right\|^2 + \left\|\mathbf{e}_{p_L}^*\right\|^2$$
$$\min \|\tilde{h} - h^*\| \le \min \|h - h^*\|$$

**Case 2:** $K' < K$. We demonstrate the inequality showing the existence of an $\tilde{h}$ that achieves a better approximation error. By definition, the minimum error too will be bounded above by this

error. For this, we fix $\gamma$, the parameterization of $h$ as $\gamma = \mathbf{V}^\dagger h^* = \gamma_p^*$ $(say)$. Note that $\gamma_{p*} = \arg\min_\gamma \|\mathbf{e}_p(\gamma)\|$. Now our objective function becomes

$$\mathbf{e}_g(\gamma_p^*, [\gamma_i]) := h^* - \mathbf{V}\gamma_p^* - \sum_{i=0}^m \mathbf{V}_i'\gamma_i$$

$$= \sum_{i=1}^m (h_i^* - \mathbf{V}_i\gamma_p^* + \mathbf{V}_i'\gamma_i) + (h_L^* - \mathbf{V}_L\gamma_p^*)$$

$$= \sum_{i=1}^m \mathbf{e}_{g_i}' + \mathbf{e}_{g_L}'$$

Where $h_i^*, h_L^*, \mathbf{V}_i, \mathbf{V}_L$ have same definitions as that in case 1 and $\mathbf{V}_i'$ is a matrix containing first $K'+1$ columns of $\mathbf{V}_i$ as its columns (and hence has full column rank), and, $\gamma_i \in \mathbb{R}^{K'+1}$. By this construction, we have

$$\|\mathbf{e}_g(\gamma_p^*, \mathbf{0})\| = \min_\gamma \|\mathbf{e}_p(\gamma)\| = \|\mathbf{e}_p(\gamma_p^*)\|$$

Our optimization objective becomes $\min_{[\gamma_i]} \|\mathbf{e}_g(\gamma_p^*, [\gamma_i])\|$, which is easy since the problem is well-posed by assumption 1 and 2. The unique minimizer of this is obtained by setting

$$\gamma_i = \mathbf{V}_i'^\dagger(h_i^* - \mathbf{V}_i\gamma_p^*) = \gamma_i^* \ (say) \ \forall i \in [m]$$

Now,

$$\|\mathbf{e}_g(\gamma_p^*, [\gamma_i^*])\| = \min_{[\gamma_i]} \|\mathbf{e}_g(\gamma_p^*, [\gamma_i])\| \le \|\mathbf{e}_g(\gamma_p^*, \mathbf{0})\| = \min_\gamma \|\mathbf{e}_p(\gamma)\| = \min \|h - h^*\|$$

By the definition of minima, $\min_{\gamma, [\gamma_i]} \|\mathbf{e}_g(\gamma, [\gamma_i])\| \le \min_{[\gamma_i]} \|\mathbf{e}_g(\gamma_p^*, [\gamma_i])\|$, and by the definition, $\min \|\tilde{h} - h^*\| = \min_{\gamma, [\gamma_i]} \|\mathbf{e}_g(\gamma, [\gamma_i])\|$, we have:

$$\min \|\tilde{h} - h^*\| \le \min \|h - h^*\|$$

$\square$

### A.4.4 Proof of Theorem 4.2

**Theorem.** Define $\mathbb{H} := \{h(\cdot) \mid \forall \text{ possible K-degree polynomial parameterizations of } h\}$ to be the set of all K-degree polynomial filters, whose arguments are $n \times n$ diagonal matrices, such that a filter response over some $\mathbf{\Lambda}$ is given by $h(\mathbf{\Lambda})$ for $h(\cdot) \in \mathbb{H}$. Similarly $\mathbb{H}' := \{\tilde{h}(\cdot) \mid \forall \text{ possible polynomial parameterizations of } \tilde{h}\}$ is set of all filters learn-able via PP-GNN , with $\tilde{h} = h + h_{f_1} + h_{f_2}$, where $h$ is parameterized by a K-degree polynomial supported over entire spectrum, $h_{f_1}$ and $h_{f_2}$ are adaptive filters parameterized by independent $K'$-degree polynomials which only act on top and bottom $t$ diagonal elements respectively, with $t < n/2$ and $K' \le K$; then $\mathbb{H}$ and $\mathbb{H}'$ form a vector space, with $\mathbb{H} \subset \mathbb{H}'$. Also, $\frac{\dim(\mathbb{H}')}{\dim(\mathbb{H})} = \frac{K+2K'+3}{K+1}$.

*Proof.* We start by constructing the abstract spaces on top of the polynomial vector space. Consider the set of all the univariate polynomials having degree at most $K$ in the vector space over the ring $\mathbb{K}_n^x := \mathbb{K}[x_1, \ldots, x_n]$ where $\mathbb{K}$ is the field of real numbers. Partition this set into $n$ subsets, say $V_1, \ldots, V_n$, such that for $i \in [n]$, $V_i$ contains all polynomials of degree up to $K$ in $x_i$. It is easy to see that $V_1, \ldots, V_n$ are subspaces of $\mathbb{K}[x_1, \ldots, x_n]$. Define $V = V_1 \oplus V_2 \oplus \cdots \oplus V_n$ where $\oplus$ denotes a direct sum. Define the matrix $D_i[c]$ whose $(i,i)^{\text{th}}$ entry is $c$ and all the other entries are zero. For $i \in [n]$, define linear maps $\phi_i : V_i \to \mathbb{M}_n(\mathbb{K}_n^x)$ by $f(x_i) \mapsto D_i[f(x_i)]$. $\text{Im}(\phi_i)$ forms a vector space of all diagonal matrices, whose $(i,i)$ entry is the an element of $V_i$. Generate a linear map $\phi : V \to \mathbb{M}_n(\mathbb{K}_n^x)$ by mapping $\phi(f(x_i))$ to $\phi_i(f(x_i))$ for all $i \in [n]$ as the components of the direct sum present in its argument. Note that $\phi_i$ for $i \in [n]$ are injective maps, making $\phi$ an injective map. This implies that $\mathbb{H} \subset \text{Im}(\phi)$ is a subspace with basis $\mathcal{B}_h := \{\phi(x_1^0 + \cdots + x_n^0), \phi(x_1 + \cdots + x_n), \ldots, \phi(x_1^K + \cdots + x_n^K)\}$, making $\dim(\mathbb{H}) = K+1$. Similarly we have, $\mathbb{H}' \subset \text{Im}(\phi)$, a subspace with basis $\mathcal{B}_{h'} := \mathcal{B}_h \bigcup \{\phi(x_1^0 + \cdots + x_t^0 + 0 + \cdots + 0), \phi(x_1 + \cdots + x_t + 0 + \cdots + 0), \ldots, \phi(x_1^{K'} + \cdots + x_t^{K'} + 0 + \cdots + 0)\} \bigcup \{\phi(0 + \cdots + 0 + x_{n-t+1}^0 + \cdots + x_n^0), \phi(0 + \cdots + 0 + x_{n-t+1} + \cdots + x_n), \ldots, \phi(0 + \cdots + 0 + x_{n-t+1}^{K'} + \cdots + x_n^{K'})\}$ where $x_i^0$ and $0$ are the corresponding multiplicative and additive identities of $\mathbb{K}_n^x$, implying $\mathbb{H} \subset \mathbb{H}'$ and $\dim(\mathbb{H}') = K + 2K' + 1$. $\square$

### A.4.5 PROOF OF COROLLARY 4.2.1

**Corollary.** The corresponding adapted graph families $\mathbb{G} := \{\mathbf{U}^T h(\cdot)\mathbf{U}^T \mid \forall h(\cdot) \in \mathbb{H}\}$ and $\mathbb{G}' := \{\mathbf{U}^T \tilde{h}(\cdot)\mathbf{U}^T \mid \forall \tilde{h}(\cdot) \in \mathbb{H}'\}$ for any unitary matrix $\mathbf{U}$ form a vector space, with $\mathbb{G} \subset \mathbb{G}'$ and $\frac{\dim(\mathbb{G}')}{\dim(\mathbb{G})} = \frac{K + 2K' + 3}{K + 1}$.

*Proof.* Consider the injective linear maps $f_1, f_2 : \mathbb{M}_n(\mathbb{K}_n^x) \to \mathbb{M}_n(\mathbb{K}_n^x)$ as $f_1(\mathbf{A}) = \mathbf{U}^T \mathbf{A}$ and $f_2(\mathbf{A}) = \mathbf{A}\mathbf{U}$. Define $f_3 : \mathbb{H} \to \mathbb{M}_n(\mathbb{K}_n^x)$ and $f_4 : \mathbb{H}' \to \mathbb{M}_n(\mathbb{K}_n^x)$ as $f_3(\mathbf{A}) = (f_1 \circ f_2)(\mathbf{A})$ for $\mathbf{A} \in \mathbb{H}$ and $f_4(\mathbf{A}) = (f_1 \circ f_2)(\mathbf{A})$ for $\mathbf{A} \in \mathbb{H}'$. Since $\mathbf{U}$ is given to be a unitary matrix, $f_3$ and $f_4$ are monomorphisms. Using this with the result from Theorem 4.2, $\mathbb{H} \subset \mathbb{H}'$, we have $\mathbb{G} \subset \mathbb{G}'$. $\square$

## A.5 EXPERIMENTS

### A.5.1 DATASETS

We evaluate on multiple benchmark datasets to show the effectiveness of our approach. Detailed statistics of the datasets used are provided in Table 5. We borrowed **Texas**, **Cornell**, **Wisconsin** from WebKB[3], where nodes represent web pages and edges denote hyperlinks between them. **Actor** is a co-occurrence network borrowed from Tang et al. (2009), where nodes correspond to an actor, and and edge represents the co-occurrence on the same Wikipedia page. **Chameleon**, **Squirrel** are borrowed from Rozemberczki et al. (2021). Nodes correspond to web pages and edges capture mutual links between pages. For all benchmark datasets, we use feature vectors, class labels from Kim & Oh (2021). For datasets in (Texas, Wisconsin, Cornell, Chameleon, Squirrel, Actor), we use 10 random splits (48%/32%/20% of nodes for train/validation/test set) from Pei et al. (2020). We borrowed **Cora**, **Citeseer**, and **Pubmed** datasets and the corresponding train/val/test set splits from Pei et al. (2020). The remaining datasets were borrowed from Kim & Oh (2021). We follow the same dataset setup mentioned in Kim & Oh (2021) to create 10 random splits for each of these datasets. We also experiment with two slightly larger datasets Flickr Chua et al. (July 8-10, 2009) and OGBN-arXiv Hu et al. (2020). We use the publicly available splits for these datasets.

| Properties | Texas | Wisconsin | Actor | Squirrel | Chameleon | Cornell | Flickr | Cora-Full | OGBN-arXiv | Wiki-CS | Citeseer | Pubmed | Cora | Computer | Photos |
|---|---|---|---|---|---|---|---|---|---|---|---|---|---|---|---|
| **Homophily Level** | 0.11 | 0.21 | 0.22 | 0.22 | 0.23 | 0.30 | 0.32 | 0.59 | 0.63 | 0.68 | 0.74 | 0.80 | 0.81 | 0.81 | 0.85 |
| **#Nodes** | 183 | 251 | 7600 | 5201 | 2277 | 183 | 89250 | 19793 | 169343 | 11701 | 3327 | 19717 | 2708 | 13752 | 7650 |
| **#Edges** | 492 | 750 | 37256 | 222134 | 38328 | 478 | 989006 | 83214 | 1335586 | 302220 | 12431 | 108365 | 13264 | 259613 | 126731 |
| **#Features** | 1703 | 1703 | 932 | 2089 | 500 | 1703 | 500 | 500 | 128 | 300 | 3703 | 500 | 1433 | 767 | 745 |
| **#Classes** | 5 | 5 | 5 | 5 | 5 | 5 | 7 | 70 | 40 | 10 | 6 | 3 | 7 | 10 | 8 |
| **#Train** | 87 | 120 | 3648 | 2496 | 1092 | 87 | 446625 | 1395 | 90941 | 580 | 1596 | 9463 | 1192 | 200 | 160 |
| **#Val** | 59 | 80 | 2432 | 1664 | 729 | 59 | 22312 | 2049 | 29799 | 1769 | 1065 | 6310 | 796 | 300 | 240 |
| **#Test** | 37 | 51 | 1520 | 1041 | 456 | 37 | 22313 | 16349 | 48603 | 5487 | 666 | 3944 | 497 | 13252 | 7250 |

Table 5: Dataset Statistics.

### A.5.2 BASELINES

We provide the methods in comparison along with the hyper-parameters ranges for each model. For all the models, we sweep the common hyper-parameters in the same ranges. Learning rate is swept over $[0.001, 0.003, 0.005, 0.008, 0.01]$, dropout over $[0.2, 0.3, 0.4, 0.5, 0.6, 0.7, 0.8]$, weight decay over $[1e-4, 5e-4, 1e-3, 5e-3, 1e-2, 5e-2, 1e-1]$, and hidden dimensions over $[16, 32, 64]$. For model-specific hyper-parameters, we tune over author prescribed ranges. We use undirected graphs with symmetric normalization for all graph networks in comparison. For all models, we report test accuracy for the configuration that achieves the highest validation accuracy. We report standard deviation wherever applicable.

**LR and MLP:** We trained Logistic Regression classifier and Multi Layer Perceptron on the given node features. For MLP, we limit the number of hidden layers to one.

**GCN:** We use the GCN implementation provided by the authors of Chien et al. (2021). Note: We observe discrepancies in performance with various implementations of GCN. We make a note of this in Section A.7.

---

[3]http://www.cs.cmu.edu/afs/cs.cmu.edu/project/theo-11/www/wwkb

**SGCN:** SGCN (Wu et al., 2019) is a spectral method that models a low pass filter and uses a linear classifier. The number of layers in SGCN is treated as a hyper-parameter and swept over $[1, 2]$.

**SUPERGAT:** SUPERGAT (Kim & Oh, 2021) is an improved graph attention model designed to also work with noisy graphs. SUPERGAT employs a link-prediction based self-supervised task to learn attention on edges. As suggested by the authors, on datasets with homophily levels lower than 0.2 we use SUPERGAT$_{SD}$. For other datasets, we use SUPERGAT$_{MX}$. We rely on authors code[4] for our experiments.

**GEOM-GCN:** GEOM-GCN (Pei et al., 2020) proposes a geometric aggregation scheme that can capture structural information of nodes in neighborhoods and also capture long range dependencies. We quote author reported numbers for Geom-GCN. We could not run Geom-GCN on other benchmark datasets because of the unavailability of a pre-processing function that is not publicly available.

**H$_2$GCN:** H$_2$GCN (Zhu et al., 2020) proposes an architecture, specially for heterophilic settings, that incorporates three design choices: i) ego and neighbor-embedding separation, higher-order neighborhoods, and combining intermediate representations. We quote author reported numbers where available, and sweep over author prescribed hyper-parameters for reporting results on the rest datasets. We rely on author's code[5] for our experiments.

**FAGCN:** FAGCN (Bo et al., 2021) adaptively aggregates different low-frequency and high-frequency signals from neighbors belonging to same and different classes to learn better node representations. We rely on author's code[6] for our experiments.

**APPNP:** APPNP (Klicpera et al., 2019) is an improved message propagation scheme derived from personalized PageRank. APPNP's addition of probability of teleporting back to root node permits it to use more propagation steps without oversmoothing. We use GPR-GNN's implementation of APPNP for our experiments.

**LightGCN:** LightGCN (Navarin et al., 2020) is a spectrally grounded GCN that adapts the entire eigen spectrum of the graph to obtain better node feature representations.

**GPR-GNN:** GPR-GNN (Chien et al., 2021) adaptively learns weights to jointly optimize node representations and the level of information to be extracted from graph topology. We rely on author's code[7] for our experiments.

**TDGNN:** TDGNN (Wang & Derr, 2021) is a tree decomposition method which mitigates feature smoothening and disentangles neighbourhoods in different layers. We rely on author's code[8] for our experiments.

**ARMA:** ARMA (Bianchi et al., 2021) is a spectral method that uses $K$ stacks of $ARMA_1$ filters in order to create an $ARMA_K$ filter (an ARMA filter of order $K$). Since (Bianchi et al., 2021) do not specify a hyperparameter range in their work, following are the ranges we have followed: GCS stacks ($S$): $[1, 2, 3, 4, 5, 6, 7, 8, 9, 10]$, stacks' depth($T$): $[1, 2, 3, 4, 5, 6, 7, 8, 9, 10]$. However we only select configurations such that the number of learnable parameters are less than or equal to those in PP-GNN. The input to the ARMAConv layer are the node features and the output is the number of classes. This output is then passed through a softmax layer. We use the implementation from the official PyTorch Geometric Library [9]

**BernNet:** BernNet (He et al., 2021) is a method that approximates any filter over the normalised Laplacian spectrum of a graph, by a $K^{th}$ Order Bernstein Polynomial Approximation. We use the model specific hyper-parameters prescribed by the authors of the paper. We vary the Propagation Layer Learning Rate as follows: $[0.001, 0.002, 0.01, 0.05]$. We also vary the Propagation Layer

---

[4] https://github.com/dongkwan-kim/SuperGAT
[5] https://github.com/GemsLab/H2GCN
[6] https://github.com/bdy9527/FAGCN
[7] https://github.com/jianhao2016/GPRGNN
[8] https://github.com/YuWVandy/TDGNN
[9] https://pytorch-geometric.readthedocs.io/en/latest/_modules/torch_geometric/nn/conv/arma_conv.html#ARMAConv

Dropout as follows: $[0.2, 0.3, 0.4, 0.5, 0.6, 0.7, 0.8]$. We rely on the authors code [10] for our experiments.

**AdaGNN:** AdaGNN (Dong et al., 2021) is a method that captures the different importance's for varying frequency components for node representation learning. We use the model specific hyperparameters prescribed by the authors of the paper. The No. of Layers hyper-parameter is varied as follows: $[2, 4, 8, 16, 32, 128]$. We rely on the authors code[11] for our experiments.

### A.5.3 COMPARISON AGAINST ADDITIONAL BASELINES

We performed experiments over the same datasets using three new baselines: ARMA (Bianchi et al. (2021)), BernNet (He et al. (2021)) and AdaGNN (Dong et al. (2021)). Details about these baselines are given in Section A.5.2. The results are reported in Tables 6 and 7. The results are obtained after thorough optimization over the respective hyperparameters, the details of which are listed in A.5.2:

| Test Acc | Texas | Wisconsin | Squirrel | Chameleon | Cornell | Flickr |
|---|---|---|---|---|---|---|
| BernNET | 83.24 (6.47) | 84.90 (4.53) | 52.56 (1.69) | 62.02 (2.28) | 80.27 (5.41) | 52.35 |
| ARMA | 79.46 (3.65) | 82.75 (3.56) | 47.37 (1.63) | 60.24 (2.19) | 80.27 (7.76) | 53.79 |
| AdaGNN | 71.08 (8.55) | 77.70 (4.91) | 53.50 (0.96) | 65.45 (1.17) | 71.08 (8.36) | 52.30 |
| GPR-GNN | 81.35 (5.32) | 82.55 (6.23) | 46.31 (2.46) | 62.59 (2.04) | 78.11 (6.55) | 52.74 |
| PP-GNN | **89.73** (4.90) | **88.24** (3.33) | **56.86** (1.20) | **67.74** (2.31) | **82.43** (4.27) | **54.44** |

Table 6: Results on Heterophilic Datasets.

| Test Acc | Cora-Full (PCA) | OGBN-ArXiv | Wiki-CS | Citeseer | Pubmed | Cora | Computer | Photos |
|---|---|---|---|---|---|---|---|---|
| BernNET | 60.77 (0.92) | 67.32 | 79.75 (0.52) | 77.01 (1.43) | 89.03 (0.55) | 88.13 (1.41) | 83.69 (1.99) | 91.61 (0.51) |
| ARMA | 60.23 (1.21) | 69.49 | 78.94 (0.32) | 78.15 (0.74) | 88.73 (0.52) | 87.37 (1.14) | 78.55 (2.62) | 90.26 (0.48) |
| AdaGNN | 59.57 (1.18) | 69.44 | 77.87 (4.95) | 74.94 (0.91) | 89.33 (0..57) | 86.72 (1.29) | 81.27 (2.10) | 89.93 (1.22) |
| GPR-GNN | 61.37 (0.96) | 68.44 | 79.68 (0.50) | 76.84 (1.69) | 89.08 (0.39) | 87.77 (1.31) | 82.38 (1.60) | 91.43 (0.89) |
| PP-GNN | **61.42** (0.79) | **69.28** | **80.04** (0.43) | **78.25** (1.76) | **89.71** (0.32) | **89.52** (0.85) | **85.23** (1.36) | **92.89** (0.37) |

Table 7: Results on Homophilic Datasets.

For AdaGNN and BernNet, we use the authors code and tune it over the hyperparameters as provided in the paper. For ARMA, we use the official PyTorch Geometric implementation (see A.5.2). As a sanity check, we also tested ARMA on the node classification datasets described in the paper and were able to reproduce similar numbers.

### A.5.4 COMPARISON AGAINST GENERAL FIR FILTERS

Instead of using a polynomial filter, we can use a general FIR filter (GFIR) which is described by the following equation:

$$Z = \sum_{k=0}^{K} S^k X H_k$$

where $S$ is the graph shift operator (which in our case is $\widetilde{A}$), $X$ is the node feature matrix and $H_k$'s are learnable filter matrices. One can see GCN, SGCN, GPR-GNN as special cases of this GFIR filter, which constrain the $H_k$ in different ways.

We first demonstrate that constraint on the GFIR filter is necessary for getting improvement in performance, particularly on heterophilic datasets. Towards this, we build two versions of GFIR: one with regularization (constrained), and the other without regularization (unconstrained). We ensure that that the number of trainable parameters in these models are comparable to those used in PP-GNN. We provide further details of the versions of the GFIR models below and report the results in Table 8 below:

---

[10] https://github.com/ivam-he/BernNet
[11] https://github.com/yushundong/AdaGNN

- **Unconstrained Setting**: In this setting, we do not impose any regularization constraints such as dropout and L2 regularization.

- **Constrained Setting**: In this setting, we impose dropouts as well as L2 regularization on the GFIR model. Both dropouts and L2 regularization were applied on the $H_k$'s (the learnable filter matrices from the above equation).

We also compare PP-GNN (the proposed model) as well as GPR-GNN to the General FIR filter model (GFIR).

| Train Acc /Test Acc | Computers | Chameleon | Citeseer | Cora | Squirrel | Texas | Wisconsin |
|---|---|---|---|---|---|---|---|
| GFIR (Unconstrained) | 78.39 (1.09) | 51.71 (3.11) | 75.83 (1.94) | 87.93 (0.90) | 36.50 (1.12) | 73.24 (6.91) | 77.84 (3.21) |
| GFIR (Constrained) | 79.57 (2.12) | 61.27 (2.42) | 76.24 (1.43) | 87.46 (1.26) | 41.12 (1.17) | 74.59 (4.45) | 79.41 (3.10) |
| GPR-GNN | 82.38 (1.60) | 62.59 (2.04) | 76.84 (1.69) | 87.77 (1.31) | 46.31 (2.46) | 81.35 (5.32) | 82.55 (6.23) |
| PP-GNN | **85.23** (1.36) | **67.74** (2.31) | **78.25** (1.76) | **89.52** (0.85) | **56.86** (1.20) | **89.73** (4.90) | **88.24** (3.33) |

Table 8: Comparing PP-GNN and GPR-GNN against the GFIR filter models.

We can make the following observation from the results reported in Table 8:

- Firstly, constrained GFIR performs better than the unconstrained version, with performance lifts of up to $\sim$10%. This suggests that regularization is important for GFIR models.

- GPR-GNN outperforms the constrained GFIR version. It is to be noted that GPR-GNN further restricts the space of graphs explored as compared to GFIR. This suggests that regularization beyond simple L2/dropout kind of regularization (polynomial filter) is beneficial.

- PP-GNN performs better than GPR-GNN. Our model slightly expands the space of graphs explored (as compared to GPR-GNN, but lesser than GFIR), while retaining good performance. This suggests that there is still room for improvement on how regularization is done.

PP-GNN has shown one possible way to constrain the space of graphs while improving performance on several datasets, however, it remains to be seen whether there are alternative methods that can do even better. We hope to study and analyze this aspect in the future.

### A.5.5 Implementation Details

In this subsection, we present several important points that are useful for practical implementation of our proposed method and other experiments related details. Our approach is based on adaptation of eigen graphs constructed using eigen components. Following Kipf & Welling (2017), we use a symmetric normalized version ($\tilde{\mathbf{A}}$) of adjacency matrix $\mathbf{A}$ with self-loops: $\tilde{\mathbf{A}} = \tilde{D}^{-\frac{1}{2}}(\mathbf{A} + \mathbf{I})\tilde{D}^{-\frac{1}{2}}$ where $\tilde{D}_{ii} = 1 + D_{ii}$, $D_{ii} = \sum_j A_{ij}$ and $\tilde{D}_{ij} = 0, i \neq j$. We work with eigen matrix and eigen values of $\tilde{\mathbf{A}}$.

To reduce the learnable hyper-parameters, we separately partition the low-end and high-end eigen values into several contiguous bins and use shared filter parameters for each of these bins. The number of bins, which can be interpreted as number of filters, is swept in the range $[2, 3, 4, 5, 10, 20]$. The orders of the polynomial filters are swept in the range $[1, 10]$ in steps of 1. The number of EVD components are swept in the range $[32, 64, 128, 256, 512, 1024]$. In our experiments, we set $\eta_l = \eta_h$ and we vary the $\eta_l$ parameter in range $(0, 1)$ and $\eta_{gpr} = 1 - \eta_l$.

For optimization, we use the Adam optimizer (Kingma & Ba, 2015). We set early stopping to 200 and the maximum number of epochs to 1000. We utilize learning rate with decay, with decay factor set to 0.99 and decay frequency set to 50. All our experiments were performed on a machine with Intel Xeon 2.60GHz processor, 112GB Ram, Nvidia Tesla P-100 GPU with 16GB of memory, Python 3.6, and PyTorch 1.9.0 (Paszke et al., 2019). We used Optuna (Akiba et al., 2019) to optimize the hyperparameter search.

### A.5.6 ADAPTABLE FREQUENCY RESPONSES

In Figure 2 of the main paper, we observe that PP-GNN learns a complicated frequency response for a heterophilic dataset (Squirrel) and a simpler frequency response for a homophilic dataset (Citeseer). We observe that this trend follows for two other datasets Chameleon (heterophilic) and Computer (homophilic). See Figure 9.

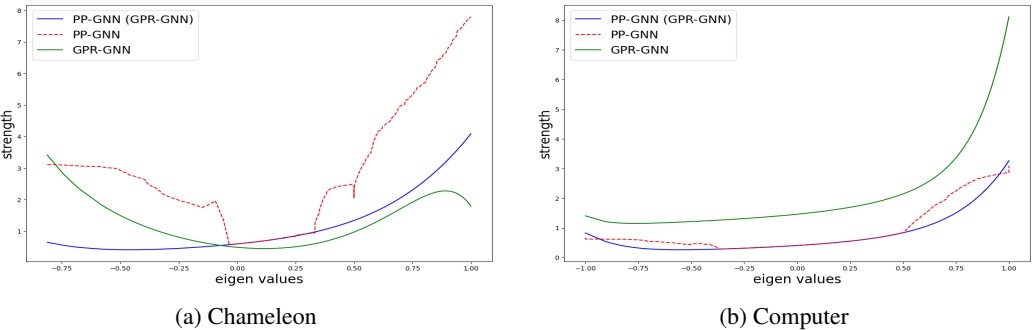

(a) Chameleon                           (b) Computer

Figure 9: Learnt Frequency Responses

### A.5.7 DOES THE POLYNOMIAL INITIALIZATION MATTER FOR GIVING GOOD PERFORMANCE?

In our experiments, we have varied the polynomial initialization schemes namely: [PPR, NPPR and Random], as described in GPRGNN paper Chien et al. (2021). To understand whether this polynomial initialization is important, we perform an ablative study to compare the various initialization schemes. We observe that our model (PP-GNN) retains similar performance across all initialization schemes. For brevity, we report results on six datasets (Cora, Citeseer, Squirrel, Chameleon, Texas and Wisconsin). The results can be found in Table 9. Our experiment results suggest that tuning polynomial initialization for each partition can indeed be avoided without compromising on the performance.

| Test Acc | Chameleon | Citeseer | Cora | Squirrel | Texas | Wisconsin |
|---|---|---|---|---|---|---|
| **PP-GNN (NPPR)** | 66.73 (1.86) | 78.41 (1.54) | 89.52 (0.85) | 57.93 (1.46) | 88.38 (3.61) | 87.25 (2.96) |
| **PP-GNN (Random)** | 68.71 (2.47) | 78.28 (1.70) | 89.42 (0.98) | 57.59 (1.56) | 88.38 (3.48) | 88.63 (2.58) |
| **PP-GNN (PPR)** | 67.74 (2.31) | 78.25 (1.76) | 89.52 (0.85) | 56.86 (1.20) | 89.73 (4.90) | 88.24 (3.33) |
| **Best Performing Baselines** | 65.45 (1.17) | 78.15 (0.74) | 88.26 (1.32)* | 53.50 (0.96) | 84.86 (6.77)* | 86.67 (4.69)* |
| | AdaGNN | ARMA | TDGNN | AdaGNN | H2GCN | H2GCN |

Table 9: Results for PP-GNN using different polynomial (gamma) initializations versus the best performing baseline models. The names of the best performing baseline models are given right below the test accuracy, in the 'Best Performing Baselines' row.

### A.5.8 DOES THE MLP EVEN MATTER?

In PP-GNN there is a two layered MLP (that transforms the input node features) followed by a single graph filtering layer similar to GPR-GNN.

To understand MLP's significance, we ran an additional experiment, where we have used a single linear layer, instead of the two layered MLP. We continue to observe competitive (with respect to our original PP-GNN model) performance, across most datasets. The results can be found in Table 10. Also, the two layer MLP is not the significant contributor towards performance. This can also be seen by comparing GPR-GNN's performance with that of LGC's. LGC can be interpreted as a linear version of GPR-GNN, and achieves comparable performance as GPR-GNN.

Note that PP-GNN (Linear) was not thoroughly trained as the PP-GNN (Original) due to time constraints. The hyperparameter ranges are explained right below the next paragraph.

| | Computers | Chameleon | Citeseer | Cora | Squirrel | Texas | Wisconsin |
|---|---|---|---|---|---|---|---|
| GPR-GNN | 82.38 (1.60) | 62.59 (2.04) | 76.84 (1.69) | 87.77 (1.31) | 46.31 (2.46) | 81.35 (5.32) | 82.55 (6.23) |
| LGC | 83.44 (1.77) | 61.14 (2.07) | 76.96 (1.73) | 88.02 (1.44) | 44.26 (1.49) | 80.20 (4.28) | 81.89 (5.98) |
| PP-GNN (Original) | 85.23 (1.36) | 67.74 (2.31) | 78.25 (1.76) | 89.52 (0.85) | 56.86 (1.20) | 89.73 (4.90) | 88.24 (3.33) |
| PP-GNN (Linear) | 84.27 (1.19) | 67.88 (1.62) | 77.86 (1.74) | 88.43 (0.69) | 55.11 (1.72) | 85.58 (4.70) | 86.24 (3.23) |

Table 10: Comparision of Linear GPR-GNN and Linear PP-GNN with respect to other pertinent baselines.

Table 10 above seems to suggest that PP-GNN (Linear) is competitive to PP-GNN (Original). We believe the difference in performance is an artifact of inadequate tuning. We can infer that adding MLP may give marginal improvements over the linear version. This phenomenon is also observed in GPR-GNN. To illustrate this, we can compare GPR-GNN with LGC (linear version of GPR-GNN). We can observe in the above table that GPR-GNN and LGC are comparable in performance.

For this additional set of experiments (Linear PP-GNN), due to time constraints, we have reduced the ranges for several hyper-parameters of our model, while also reducing the number of Optuna Trials to 50. These reduced ranges are as follows:

- Learning Rate: [0.003, 0.005]
- Weight Decay : [0.0001, 0.001]
- Dropout: [0.3, 0.5]
- Hidden Dims: [32, 64]
- Order of the Polynomial: [2, 4]
- No. of Partitions (Buckets): [2, 4]
- Number of Eigen Value: [256, 1024]
- $\eta$: Sampled uniformly between [0, 1] using Optuna

Despite this smaller hyper-parameters search space, we are only shy by a few percentage points on the Texas dataset. We believe this difference can be recovered with thorough tuning.

A.6    TRAINING TIME ANALYSIS

In the following subsections, we provide comprehensive timing analysis.

**Computational Complexity:**    Listed below is the computational complexity for each piece in our model for a single forward pass. Notation $n$: number of nodes, $|E|$: the number of edges, $A$: symmetric normalized adjacency matrix, $F$: features dimensions, $d$: hidden layer dimension, $C$: number of classes, $e^*$ denotes the cost of EVD, $K$: polynomial order/hop order, $l$: number of eigenvalues/vectors in a single partition of spectrum (for implementation, we keep $l$ same for all such intervals), $m$: number of partitions of a spectrum.

- MLP: $O(nFd + ndC)$
- GPR-term: $O(K|E|C) + O(nKC)$. The first term is the cost for computing $A^K f(X)$ for sparse $A$. The second term is the cost of summation $\sum_k A^k f(X)$.
- Excess terms for PP-GNN: $O(mnlC)$. This is obtained by the optimal matrix multiplication present in Equation 3 of the main paper ($\mathbf{U}_i$ is $n \times l$, $H_i(\gamma_i)$ is $l \times l$, $\mathbf{Z}_0()$ is $n \times C$). The additional factor $m$ is because we have $m$ different contiguous intervals/different polynomials. Note that typically $n$ is much larger than $l$.
- EVD-term: $e^*$, the complexity for obtaining the eigenvalues/vectors of the adjacency matrix, which is usually very sparse for the observed graphs. Most publicly available solvers for this task utilize Lanczos' algorithm (which is a specific case of a more general Arnoldi iteration). However, the convergence bound of this iterative procedure depends upon the starting vectors and the underlying spectrum (particularly the ratio of the absolute difference of two largest eigenvalues to the diameter of the spectrum) [Saad (1980), LI

(2010), Cullum & Willoughby (2002)]. Lanczos' algorithm is shown to be a practically efficient way for obtaining extreme eigenpairs for a similar and even very large systems. We use ARPACK's built-in implementation to precompute the eigenvalues/vectors for all datasets before training, thus amortizing this cost across training with different hyper-parameters configuration.

**Per Component Timing Breakup:** In Table 11, we provide a breakdown of cost incurred in seconds for different components of our model. Since the eigenpairs' computation is a one time cost, we amortize this cost over the total hyper-parameters configurations and report the effective training time in the last column on of Table 11.

**Average Training Time:** In Table 12, we report the training time averaged over 100 hyper-parameter configurations for several models. To understand the relative performance of our model with respect to GCN, we compute the relative time taken and report it in Table 13. We can observe in Table 13 that PP-GNN is $\sim 4x$ slower than GCN, $\sim 2X$ slower than GPR-GNN and BernNET, and $\sim 2X$ faster than AdaGNN. However, it is important to note that in our average training time, the time taken to compute $K$ top and bottom eigenvalues/vectors is amortized across the number of trials

**Can we cut down the total training cost for PP-GNN?** To study this, we analyzed the chosen hyper-parameters for our 100 configurations experiments and prescribe below the new ranges for hyperparameters. Reduced hyperparameters space:

- Learning Rate: [0.003, 0.005]
- Weight Decay : [0.0001, 0.001]
- Dropout: [0.3, 0.5]
- Hidden Dims: [32, 64]
- Order of the Polynomial: [2, 4]
- No. of Partitions (Buckets): [2, 4]
- Number of Eigen Value: [256, 1024]
- $\eta$: Sampled uniformly between [0, 1] using Optuna

We further restrict the total number of Optuna trials to 20. Note that as shown in Section A.5.7, our model is insensitive to the polynomial initialization scheme. Hence we use 'random' initialization, thereby reducing the number of hyperparameters even further. The total time taken by PP-GNN (including the one-time cost for obtaining eigenpairs) for training (and performing hyperparameter optimization) over 20 hyper-parameter configurations is reported in Table 14. For comparison, we also show the total training time for the 100 configurations experiments in Table 15 which too includes the time taken to obtain $\min(n, 1024)$ eigenpairs. From Tables [14,15] we can observe that the total training time has reduced by $\sim 5x$. In Table 16, we can observe that even with the reduced hyper-parameter range, our model performs competitively or better.

A.7 DISCREPANCIES IN PERFORMANCE BETWEEN DIFFERENT GCN IMPLEMENTATIONS

We tested three different implementations of GCN(s) (Kipf & Welling, 2017) provided by 1) Thomas Kipf and Max Welling [12] (In TensorFlow), 2) The implementation[13] of the authors of the GPR-GNN paper (Chien et al., 2021) (in PyTorch), 3) The code base [14] publicly available at the OGBN-ArXiv leader-board [15] (in PyTorch), on the OGBN-ArXiv dataset.

---

[12] https://github.com/tkipf/gcn
[13] https://github.com/jianhao2016/GPRGNN
[14] https://github.com/snap-stanford/ogb/tree/master/examples/nodeproppred/arxiv
[15] https://ogb.stanford.edu/docs/leader_nodeprop/#ogbn-arxiv

| PP-GNN | Training Time | EVD Cost | Number of EV's obtained | Effective Training Time |
|---|---|---|---|---|
| **Texas** | 11.89 | 0.00747 | 183 (All EVs) | 11.89 |
| **Cornell** | 11.63 | 0.03271 | 183 (All EVs) | 11.63 |
| **Wisconsin** | 12.08 | 0.01225 | 251 (All EVs) | 12.08 |
| **Chameleon** | 21.44 | 3.71883 | 2048 | 21.48 |
| **Squirrel** | 31.38 | 15.8152 | 2048 | 31.54 |
| **Cora** | 22.46 | 54.3684 | 2048 | 23.01 |
| **Citeseer** | 20.51 | 56.9744 | 2048 | 21.08 |
| **Cora-Full** | 63.98 | 155.304 | 2048 | 65.53 |
| **Pubmed** | 52.54 | 256.71 | 2048 | 55.11 |
| **Computers** | 28.63 | 76.2738 | 2048 | 29.39 |
| **Photo** | 19.3 | 48.3683 | 2048 | 19.78 |
| **Flickr** | 161.16 | 304.114 | 2048 | 164.20 |
| **ArXiv** | 189.94 | 412.504 | 1024 | 194.06 |
| **WikiCS** | 27.92 | 65.4376 | 2048 | 28.57 |

Table 11: **PP-GNN's per component timing cost**. Training Time refers to the end to end training time (without eigen decomposition) averaged across 100 trials. EVD cost refers to the time taken to obtain **x** top and bottom eigenvalues. This **x** can be found in the 'Number of EV's obtained' column. Since EVD is a one time cost, we average this cost over the total number of trials and add it to the training time. We refer to this cost as the Effective Training Time.

| Dataset | GPR-GNN | PP-GNN | MLP | GCN | BernNet | ARMA | AdaGNN |
|---|---|---|---|---|---|---|---|
| **Texas** | 9.27 | 11.89 | 1.08 | 3.46 | 5.59 | 6 | 13.97 |
| **Cornell** | 9.41 | 11.63 | 1.06 | 3.69 | 5.37 | 5.51 | 12.56 |
| **Wisconsin** | 9.67 | 12.08 | 1.07 | 3.42 | 5.69 | 5.36 | 13.57 |
| **Chameleon** | 14.69 | 21.48 | 2.6 | 6.42 | 12.46 | 7.84 | 28.77 |
| **Squirrel** | 18.94 | 31.54 | 5.04 | 7.52 | 17.82 | 28.87 | 90.36 |
| **Cora** | 12.9 | 23.01 | 1.95 | 5.94 | 12.25 | 10.67 | 22.15 |
| **Citeseer** | 10.62 | 21.08 | 3.72 | 4.56 | 9.52 | 19.5 | 35.34 |
| **Cora-Full** | 24.98 | 65.53 | 7.77 | 8.01 | 31.26 | 40.21 | 175.58 |
| **Pubmed** | 14 | 55.11 | 6.21 | 11.73 | 12.64 | 27.76 | 162.01 |
| **Computers** | 7.67 | 29.39 | 2.24 | 6.68 | 7.48 | 27.76 | 118.43 |
| **Photo** | 8.58 | 19.78 | 1.68 | 5.1 | 7.95 | 14.34 | 45.46 |
| **Flickr** | 42.64 | 164.20 | 21 | 30.4 | 62.11 | 119.3 | 178.7371 |
| **ArXiv** | 118.35 | 194.06 | 78.9 | 102.88 | 693.92 | 771.59 | 307.84 |
| **WikiCS** | 14.37 | 28.57 | 3.34 | 10.8 | 11.43 | 30.79 | 73.63 |

Table 12: Training Time (in seconds) across Models

Note that the authors of the leader-board code report a test accuracy of $71.74\%$ on the OGBN-ArXiV dataset. Extensive tuning was done on the Kipf and Welling implementation in order to match the results of the leaderboard code. However, even after extensive hyper-parameter tuning, the test accuracy increased to only $65.53\%$, which is still much lesser.

Two other implementations that we tested were: a) The GCN implementation from the authors of the GPR-GNN paper (Chien et al., 2021) (in PyTorch) b) The leader-board implementation (in PyTorch). The results for all the three implementations on the OGBN-ArXiv Dataset can be found in Table 17.

| Dataset | GPR-GNN | PP-GNN | MLP | GCN | BernNet | ARMA | AdaGNN |
|---|---|---|---|---|---|---|---|
| **Texas** | 2.68 | 3.44 | 0.31 | 1.00 | 1.62 | 1.73 | 4.04 |
| **Cornell** | 2.55 | 3.15 | 0.29 | 1.00 | 1.46 | 1.49 | 3.40 |
| **Wisconsin** | 2.83 | 3.53 | 0.31 | 1.00 | 1.66 | 1.57 | 3.97 |
| **Chameleon** | 2.29 | 3.35 | 0.40 | 1.00 | 1.94 | 1.22 | 4.48 |
| **Squirrel** | 2.52 | 4.19 | 0.67 | 1.00 | 2.37 | 3.84 | 12.02 |
| **Cora** | 2.17 | 3.87 | 0.33 | 1.00 | 2.06 | 1.80 | 3.73 |
| **Citeseer** | 2.33 | 4.62 | 0.82 | 1.00 | 2.09 | 4.28 | 7.75 |
| **Cora-Full** | 3.12 | 8.18 | 0.97 | 1.00 | 3.90 | 5.02 | 21.92 |
| **Pubmed** | 1.19 | 4.70 | 0.53 | 1.00 | 1.08 | 2.37 | 13.81 |
| **Computers** | 1.15 | 4.40 | 0.34 | 1.00 | 1.12 | 4.16 | 17.73 |
| **Photo** | 1.68 | 3.88 | 0.33 | 1.00 | 1.56 | 2.81 | 8.91 |
| **Flickr** | 1.40 | 5.40 | 0.69 | 1.00 | 2.04 | 3.92 | 5.88 |
| **ArXiv** | 1.15 | 1.89 | 0.77 | 1.00 | 6.74 | 7.50 | 2.99 |
| **WikiCS** | 1.33 | 2.65 | 0.31 | 1.00 | 1.06 | 2.85 | 6.82 |
| **Average** | **2.03** | **4.09** | **0.50** | **1.00** | **2.19** | **3.18** | **8.39** |

Table 13: Training Time of models relative to the training time of GCN

| Dataset | Chameleon | Citeseer | Computers | Cora | Cora-Full | Photo | Pubmed | Squirrel | Texas | Wisconsin | OGBN-ArXiv |
|---|---|---|---|---|---|---|---|---|---|---|---|
| Time | 00:03:46 | 00:10:17 | 00:34:37 | 00:05:24 | 00:59:29 | 00:10:31 | 00:57:40 | 00:10:38 | 00:02:27 | 00:02:33 | 01:03:20 |

Table 14: End to end training time (HH:MM:SS) for optimizing over 20 hyperparameter configurations

Besides different implementations of GCN(s), we find that batch normalization is done in the implementation used in the GCN leader-board code. We see that (in Table 17) the performance numbers vary depending on the implementation. Also, batch normalization helps to get nearly 2% improvement for this dataset. Since we did not perform batch normalization in all our experiments we find a performance gap of $\sim 2\%$ for this dataset in results reported in our paper (against the leader-board code).

We have also compared the GCN implementation from Kipf and Welling as well as the GCN implementation from the authors of the GPR-GNN across various datasets in Table 18.

We can see that both the TensorFlow and the PyTorch implementations lead to a similar performance for most of the datasets. However, there are some differences in performance for some datasets (Wisconsin, Cora-Full (PCA), OGBN-ArXiv, Computer and Photos). We have ensured that hyperparameters, preprocessing steps etc., are same across both these implementations. We believe that these differences can be attributed to the different internal workings of TensorFlow and PyTorch.

A further study is required to understand where these gaps are coming from.

Note: In the main paper (in Tables [1, 2]), we report the results obtained on the GCN implementation by the authors of the GPR-GNN paper.

| Dataset | Chameleon | Citeseer | Computers | Cora | Cora-Full | Cornell | Flickr | Photo | Pubmed | Squirrel | Texas | Wikics | Wisconsin | OGBN-ArXiv |
|---------|-----------|----------|-----------|------|-----------|---------|--------|-------|--------|----------|-------|--------|-----------|------------|
| Time | 01:23:24 | 01:21:26 | 01:51:41 | 01:30:31 | 04:03:35 | 00:44:00 | 03:39:57 | 01:15:25 | 03:24:29 | 01:59:20 | 00:44:03 | 00:41:04 | 00:46:58 | 05:06:5 |

Table 15: End to end training time (HH:MM:SS) for optimizing over 100 hyperparameter configurations

| | Texas | Squirrel | Chameleon | Cora-Full (PCA) | OGBN-ArXiv | Citeseer | Pubmed | Cora | Photos (Amazon) |
|---|-------|----------|-----------|-----------------|------------|----------|--------|------|-----------------|
| PP-GNN | 85.14 (2.30) | 59.15 (1.91) | 69.10 (1.37) | 60.93 (0.83) | 69.1 | 77.87 (1.93) | 89.43 (0.47) | 88.87 (0.90) | 92.18 (0.58) |
| Best Performing Baseline | 84.86 (6.77)* H2GCN | 53.50 (0.96) AdaGNN | 65.45 (1.17) AdaGNN | 61.84 (0.90) LGC | 69.37 GCN | 77.07 (1.64)* H2GCN | 89.59 (0.33)* H2GCN | 88.26 (1.32) TDGNN | 92.54 (0.28) TDGNN |

Table 16: Comparison of PP-GNN trained on a reduced hyper-parameter space against the best performing baseline models. Note that the best performing baseline models are trained on the full (original hyper-parameter set) and also for 100 Optuna Trials. The names of the best performing baseline models are given below the value of the test accuracy, in the 'Best Performing Baseline' row.

| OGBN-ArXiv | Framework | Batch Norm | Test Accuracy |
|------------|-----------|------------|---------------|
| GCN (Kipf and Welling) | TensorFlow | True | 65.84 |
| GCN (Kipf and Welling) | TensorFlow | False | 65.53 |
| GCN (Leaderboard Code) | PyTorch | True | 71.88 |
| GCN (Leaderboard Code) | PyTorch | False | 70.23 |
| GCN (GPR-GNN) | PyTorch | True | 71.05 |
| GCN (GPR-GNN) | PyTorch | False | 69.37 |

Table 17: GCN results with different implementations.

| | Texas | Wisconsin | Squirrel | Chameleon | Cornell | Cora-Full (PCA) | Ogbn-arxiv | Citeseer | Pubmed | Cora | Computer | Photos |
|---|-------|-----------|----------|-----------|---------|-----------------|------------|----------|--------|------|----------|--------|
| GCN (GPR-GNN, PyTorch) | 59.73 (4.89) | 58.82 (4.89) | 47.78 (2.13) | 62.83 (1.52) | 60.00 (4.90) | 59.63 (0.86) | 69.37 | 76.47 (1.34) | 88.41 (0.46) | 87.36 (0.91) | 82.50 (1.23) | 90.67 (0.68) |
| GCN (Kipf and Welling, TensorFlow) | 61.62 (6.14) | 53.53 (4.73) | 46.04 (1.61) | 61.43 (2.70) | 62.97 (5.41) | 45.44 (1.01) | 63.48 | 76.47 (1.33) | 87.86 (0.47) | 86.27 (1.34) | 78.16 (1.85) | 86.38 (1.71) |

Table 18: Comparing different GCN implementations. (GPR-GNN implementation vs Kipf and Welling Implementation)

