# OpenReview forum: "Effective Polynomial Filter Adaptation for Graph Neural Networks"
_ICLR.cc/2022/Conference — ICLR 2022 Submitted_

### Official Review · Reviewer_eu6Q · 2021-10-30

**Correctness:** 3
**Technical Novelty And Significance:** 3
**Empirical Novelty And Significance:** 2
**Recommendation:** 5
**Confidence:** 5

**Main Review:**

This paper focuses on the task of node classification (semi-supervised learning) in heterophilic graphs. The authors identify that low-pass filter-based GNNs perform poorly on heterophilic graphs, precisely because they tend to smooth out differences in neighboring node features. Thus, they propose to look into filters capable of learning high-frequency content that can thus learn labels even if neighboring nodes do not share a label.

In particular, the authors propose to learn different low-order polynomials for different parts of the spectrum. I find this idea very interesting, and worth investigating.

However, I find the premises of the paper not to be entirely adequate. In particular, there is quite a few references missing that should be more relevant for the discussion at hand. This leads the experimental results, albeit certainly very extensive, quite uninformative.

In what follows, I will introduce some well-known definitions to ground the discussion and to illustrate my point. Most of the basic graph signal processing (GSP) definitions can be found in

[R1] A. Sandryhaila and J. M. F. Moura, "Discrete signal processing on graphs," IEEE Trans. Signal Process., vol. 61, no. 7, pp. 1644-1656, 1 Apr. 2013.

[R2] A. Sandyhaila and J. M. F. Moura, "Discrete signal processing on graphs: Frequency analysis," IEEE Trans. Signal Process., vol. 62, no. 12, pp. 3042-3054, Jun. 2014.

Given a graph G = (V,E), a graph signal is defined as a mapping between the node set and the real numbers, x: V \to R. This can be conveniently collected in a vector x \in R^n, where n is the number of nodes of the graph.

An FIR graph filter is a mapping H(S) between graph signals given by

H(S) = \sum_{k=0}^{K} h_k S^k

where S is the so-called graph shift operator (i.e. a n x n matrix description of the graph, for example, the adjacency matrix, the Laplacian matrix, the Markov matrix, or any of their normalized counterparts) and where h_k are the filter taps. The output y = H(S)x is another graph signal.

At this point, the authors correctly identify the shortcomings of GCN (Kipf and Welling 2017) --and many derived works-- in terms of the frequency response of the learned filters: GCN forces K = 1, h_0 = 0 and only learns h_1. By choosing S = normalization(I+A), then it is evident that the frequency response of the learned filters is going to be h_1*\lambda, and since S is derived from the adjacency matrix, higher eigenvalues denote lower frequencies [R2]. Thus, the only filters learned are low-pass filters, leading to good classification in homophily graphs (where neighbors share a label). This is, of course, also true of SGCs (Wu et al, 2019) where the h_k = 0 for all k < K and only h_K is learned, thus the frequency response is h_K * \lambda^k, which is also a low pass filter when choosing S = I+A.

It's worth noting, that even if K = 1, it is possible to create a high-pass filter if only h_0 can be learned as well. As a matter of fact, by setting h_0 = 2 and h_1 = -1, the resulting frequency response is 2 - \lambda, which is a high-pass filter (high-frequency response for small lambda, lower frequency response for higher lambda).

Now, that one needs to learn all filter taps, and that the value of K is a design choice akin to the kernel size in CNNs, is duly noted in Defferrard et al, 2021. Albeit, ChebNets restrict themselves to normalized Laplacians and filter taps computed by means of a Chebyshev polynomial.

At this point, the author's assessment is correct, in that, in order to learn classification in heterophilic graphs, high-pass filters are necessary (to compute differences with neighborhoods), and thus GCNs (and their variants like SGC) are always going to fall short.

My concerns arise when handling node features, which are typically d-dimensional vectors. In this case, the concept of graph signal is extended to encompass node features: x is now a mapping from the node set to R^d. This is now conveniently collected in a matrix X in R^{n \times d}, as the authors correctly observe.

When handling node features, the dimension of the node features at the output, say e, may be different from the dimension of the node features at the input, say d. Thus, the FIR graph filter is extended to be a bank of D x E graph filters, which can be compactly written as

H(X;S) = \sum_{k=0}^{K} S^k X H_k

where now the filter taps H_k are D x E matrices. This was duly noted in Defferrard et al 2016, albeit with a very particular value of S, and with a Chebyshev recursion on the H_k. This was properly extended to arbitrary filters with arbitrary values of S back in 2018:

[R3] F. Gama, A. G. Marques, G. Leus, and A. Ribeiro, "Convolutional neural network architectures for signals supported on graphs," IEEE Trans. Signal Process., vol. 67, no. 4, pp. 1034–1049, 17 Dec. 2018.

Once we have the FIR graph description for node features

H(X;S) = \sum_{k=0}^{K} S^k X H_k

It is immediate to see many of the other architectures as imposing different constraints on this general definition, i.e.

GCN: K = 1, S = normalization(I+A), H_0 = 0, H_1 = learnable

SGC: K chosen, S = normalization(I+A), H_k = 0 for k < K, H_K = learnable

GPR-GNN: K chosen, S = normalization(I+A), H_k = \gamma_k I for k > 0 (In the GPR-GNN, H_0 is learned by means of the same MLP applied to the features of each node, if such MLP has length L, this can be written as X_l = sigma(X_{l-1} H_l), so it can be seen as adding L layers with K=0 at the beginning of the GNN)

As we can see, all these architectures, even if they can learn high-pass filters (as is the case of GPR-GNN), unnecessarily restrict the space of allowable filters, for no apparent reason. This will, undeniably, affect their performance.

Thus, we reach my first concern: I genuinely appreciate the exhaustiveness of the numerical experiments, but I believe them to be misguided. Comparisons with GCNs and SGCs are pointless in the context of heterophilly, and comparison with GPR-GNN is misleading due to the unnecessary constraints. I would expect, at the very least, a comparison with a GNN built with arbitrary FIR graph filters with no constraints, i.e., where the linear part of the layer is given by the following equation

H(X;S) = \sum_{k=0}^{K} S^k X H_k

ChebNets are a good implementation for this, albeit, for a fair comparison, the value of S should be the same, and thus, Chebyshev Polynomials may not be an option, so that the architecture in [R3] may make more sense.

The next argument that the authors make is the choice of K. The authors correctly identify that for K = N-1 all possible filters can be learned (in theory, albeit probably not in practice due to optimization problems in a space that grows with the size of the data) from the FIR graph filter due to the Cayley-Hamilton theorem (as long as the eigenvalues are distinct). The authors correctly argue that high values of K are impractical, because higher values of K involve a higher number of exchanges with neighbors, incurring a communication delay.

While this is true, the reasons to propose the spectrum partition are not entirely justified.

On the one hand, it has been noted that even K=1 is enough to learn a high-pass filter.

On the other hand, higher values of K allow for sharper frequency transitions, which may or may not be useful.

The authors thus propose to partition the frequency space (at the expense of an eigendecomposition) and use low-order polynomials over different parts of the spectrum.

However, it is known that the use of ARMA graph filters lead to sharper transition with a fewer number of exchanges

[R4] R. Levie, F. Monti, X. Bresson, and M. M. Bronstein, "CayleyNets:
Graph convolutional neural networks with complex rational spectral filters," IEEE Trans. Signal Process., vol. 67, no. 1, pp. 97–109, Jan. 2019.

[R5] F. M. Bianchi, D. Grattarola, L. Livi, and C. Alippi, "Graph Neural Networks with Convolutional ARMA Filters", IEEE Trans. Pattern Anal. Mach. Intell., 26 Jan. 2021, early access. [Online]. Available: https://doi.org/10.1109/TPAMI.2021.3054830

This gives rise to my second concern: I believe that comparisons with GNNs using ARMA graph filters is absolutely necessary to see if the sharper transitions obtained by smaller frequency bands are worth their use over other techniques to design sharp transitions with small K.

The third concern comes from the fact that, when partitioning the spectrum, the authors are learning a collection of filters, one for each part of the spectrum. However, the expression for processing node features already includes a bank of graph filters. How does the choice of node features in the hidden layers (i.e. number of hidden units) interplay with the number of filters learned in the spectrum transition?

To be more precise on my third concern, let's assume that we partition the spectrum on three different frequency bands, and we learn a different low-order polynomial for each of them. Due to superposition, these three bands can be thought of as three different filters. How do I know that this is not equivalent to learning those three different filters in the filter bank used to map d-dimensional node features into e-dimensional node features? What's the relationship between the number of features in the filter bank and the partitioning of the spectrum?

The fourth concern starts from the use of GNNs such as SuperGAT which involve non-convolutional graph filters. The frequency responses of these filters are ill-defined, thus comparing with non-convolutional GNNs, while great to show SotA is an unfair comparison in general, but in particular in an architecture that proposes a frequency interpretation of the results. A general framework to understand the differences between convolutional and non-convolutional GNNs can be found in

[R6] E. Isufi, F. Gama, and A. Ribeiro, "EdgeNets: Edge varying graph neural networks," IEEE Trans. Pattern Anal. Mach. Intell., 13 Sep. 2021, early access. [Online]. Available: https://doi.org/10.1109/TPAMI.2021.3111054

In this respect, I would like to note that ARMA graph filters are convolutional, as well as FIR filters (and IIR filters). Thus, a comparison with ARMA filters is fair in the context of graph convolutional neural networks.

For the fifth concern, I would like to note that the notion of partitioning the frequency spectrum is evocative of wavelet filter banks. A very nice tutorial on graph wavelets can be found in

[R7] D. I. Shuman, "Localized spectral graph filter frames: A unifying framework, survey of design considerations, and numerical comparison", IEEE Signal Process. Mag., vol. 37, no. 6, pp. 43-63, Nov. 2020.

In particular, graph wavelets have been used in the context of GNNs to create graph scattering transforms

[R8] D. Zou and G. Lerman, "Graph convolutional neural networks via scattering," Appl. Comput. Harmonic Anal., vol. 49, no. 3, pp. 1046-1074, Nov. 2020.

[R9] F. Gao, G. Wolf, and M. Hirn, "Geometric scattering for graph
data analysis," in 36th Int. Conf. Mach. Learning, Long Beach, CA, 15-9 June 2019, pp. 1–10.

[R10] F. Gama, J. Bruna, and A. Ribeiro, "Stability of graph scattering transforms," in 33rd Conf. Neural Inform. Process. Syst. Vancouver, BC: Neural Inform. Process. Syst. Foundation, 8-14 Dec. 2019, pp. 8038–8048.

I believe that the relationships with graph scattering transforms should be discussed, if not also included in the numerical simulations (in this respect, I would recommend the works by Wolf, who has studied the subject extensively, even proposing learnable variants of graph scattering transforms).

The sixth concern that I have is about computational complexity. It is stated that computing the k-lowest and k-highest eigenvalues are relatively inexpensive, reporting a cost of O(nkL) in addition to the GPR-GNN. I would suggest including the total cost of each of the main architectures to compare, and if possible, runtimes. As the authors are probably aware, the shift from conceptually interesting but impractical GNNs (Bruna et al, 2014) to computable ones (Defferrard et al, 2016) and the consequent boom of GNNs, is due to the avoidance of eigendecompositions. Going back to them, while interesting from a conceptual standpoint (and I do appreciate it), requires a much lengthier justification.

My seventh concern is that, unfortunately, I don't understand Theorem 1, and I would highly appreciate it if the authors can clarify it. My observation is that \hat{h} is a superset of h, (i.e. if we set h_f = 0, we end up with h). It should then be evident that the error achieved by \hat{h} should be lower or equal to the error achieved by h (i.e. h_f can only help... if it's useless, just set it to be equal to 0 yielding h). A clarification on what the authors tried to achieve with this theorem would be highly appreciated.

My eighth and last concern is the title. The inclusion of a polynomial filter is slightly misleading. There's nothing novel about it. However, the most interesting novelty, the spectrum partition, is completely omitted.

Summary of concerns:

The paper is certainly interesting in the idea of partitioning the spectrum and learning different filters for different bands. I think this idea is worth publishing in ICLR, albeit not in its current form.

A) The numerical experiments are extensive, but unfortunately, the choice of baselines is misguided. Comparisons with general FIR filters as well as ARMA filters are absolutely necessary. Comparisons with non-convolutional GNNs are optional, but I understand it from a SOTA perspective.

B) A discussion on the relationship between the proposed method and scattering transforms is necessary, due to the nature of the spectrum partitioning and wavelets. While including numerical simulations comparing with scattering transforms would be very much welcome, it is indeed not necessary and maybe only left for future work.

C) The interplay between the bank of FIR graph filters (i.e. the d x e bank of filters) and the partitioning of the spectrum should be explored: how do we know that these partitions cannot be learned in the bank of filters? choosing the intervals acts as a regularizer?

D) The increase in computational complexity due to the computation of eigenvalues should be thoroughly explained and total computational costs should be reported for all architectures.

E) I would appreciate it if the authors clarify my concern on Theorem 1.

F) I believe the title could be improved to better reflect the paper's contributions.

Minor comments:

(1) is incomplete. The term for j=0 is missing, and as it has been discussed extensively above, the ability to learn the filter tap for the term j=0 is key for obtaining high-pass filters (even if k is small).

Page 3, where it reads "nod features computed via A^j X becomes indistinguishable" should read "node features computed via A^j X become indistinguishable"

The i subindex in \gamma in eq. (3) is boldfaced and I believe it shouldn't be.

The acronym GPR-GNN is mentioned for the first time in point 2 on page 5, but it has never mentioned what it is. Please, explicitly say that this refers to the Generalized Page Rank architecture of Chien et al.

In theorem 4.2, where it says 'learn-able' it should say 'learnable'.

Please, in the paragraph titled 'baselines' of the simulation section, add the corresponding references (I know these are in the supplementary material, but at least the references should also be included here)

**Summary Of The Paper:**

This paper focuses on the task of node classification (semi-supervised learning) in heterophilic graphs. The authors identify that low-pass filter-based GNNs perform poorly on heterophilic graphs, precisely because they tend to smooth out differences in neighboring node features. Thus, they propose to look into filters capable of learning high-frequency content that can thus learn labels even if neighboring nodes do not share a label. In particular, the authors propose to learn different low-order polynomials for different parts of the spectrum.

**Summary Of The Review:**

I find the idea of partitioning the spectrum to be very interesting. However, the idea is not fully explored and a lot of space is devoted to uninformative comparisons.

I am currently assigning a score of 4, but I really see promise in this paper. So I'm willing to raise my score to 7 if my concerns are either addressed, or an explanation on why I have misinterpreted the paper is given.

The summary of the changes I expect is given in what follows:

A) The numerical experiments are extensive, but unfortunately, the choice of baselines is misguided. Comparisons with general FIR filters as well as ARMA filters are absolutely necessary. Comparisons with non-convolutional GNNs are optional, but I understand it from a SOTA perspective.

B) A discussion on the relationship between the proposed method and scattering transforms is necessary, due to the nature of the spectrum partitioning and wavelets. While including numerical simulations comparing with scattering transforms would be very much welcome, it is indeed not necessary and maybe only left for future work.

C) The interplay between the bank of FIR graph filters (i.e. the d x e bank of filters) and the partitioning of the spectrum should be explored: how do we know that these partitions cannot be learned in the bank of filters? choosing the intervals acts as a regularizer?

D) The increase in computational complexity due to the computation of eigenvalues should be thoroughly explained and total computational costs should be reported for all architectures.

E) I would appreciate it if the authors clarify my concern on Theorem 1.

F) I believe the title could be improved to better reflect the paper's contributions.

--- 0 ---

Update after the first round of reviews:

I consider D, B, and F to be appropriately addressed.

I have minor concerns on E.

I still have major concerns about A and C, mainly with respect to the omission of graph convolutional filters upon the objection of overfitting.

Thank you very much to the authors for their time and effort in addressing my concerns. I will increase my score to 5.

---

> ### Author Response · Authors · 2021-11-18
> **Response to Concern A: Additional Baselines [Reviewer eu6Q ]**
>
> **[A] The numerical experiments are extensive, but unfortunately, the choice of baselines is misguided. Comparisons with general FIR filters as well as ARMA filters are absolutely necessary. Comparisons with non-convolutional GNNs are optional, but I understand it from a SOTA perspective.**
>
> We thank the reviewer for pointing us to an important missing baseline ARMA[1]. We have also added comparisons against some very recent filtering approaches like ADAGNN [2] and BernNet [3] and report in TABLE 1.
>
> $\\\\$
> $\\\\$
> $\\\\$
> ---
> ### TABLE-1: Additional Baselines
> ---
>
> | | Texas | Wisconsin | Squirrel | Chameleon | Cornell | Flickr | Cora-Full | OGBN-ArXiv | Wiki-CS | Citeseer | Pubmed | Cora | Computer | Photos |
> |:-----------:|:----------------:|:---------------------:|:---------------------:|:---------------------:|:---------------------:|:---------------:|:----------------------:|:-----------------:|:---------------------:|:---------------------:|:----------------------:|:---------------------:|:---------------------:|:---------------------:|
> | **BernNET** | 83.24 (6.47) | 84.90 (4.53) | 52.56 (1.69) | 62.02 (2.28) | 80.27 (5.41) | 52.35 | 60.77 (0.92) | 67.32 | 79.75 (0.52) | 77.01 (1.43) | 89.03 (0.55) | 88.13 (1.41) | 83.69 (1.99) | 91.61 (0.51) |
> | **ARMA** | 79.46 (3.65) | 82.75 (3.56) | 47.37 (1.63) | 60.24 (2.19) | 80.27 (7.76) | 53.79 | 60.23 (1.21) | **69.49** | 78.94 (0.32) | 78.15 (0.74) | 88.73 (0.52) | 87.37 (1.14) | 78.55 (2.62) | 90.26 (0.48) |
> | **AdaGNN** | 71.08 (8.55) | 77.70 (4.91) | 53.50 (0.96) | 65.45 (1.17) | 71.08 (8.36) | 52.30 | 59.57 (1.18) | 69.44 | 77.87 (4.95) | 74.94 (0.91) | 89.33 (0..57) | 86.72 (1.29) | 81.27 (2.10) | 89.93 (1.22) |
> | **GPR-GNN [4]** | 81.35 (5.32) | 82.55 (6.23) | 46.31 (2.46) | 62.59 (2.04) | 78.11 (6.55) | 52.74 | 61.37 (0.96) | 68.44 | 79.68 (0.50) | 76.84 (1.69) | 89.08 (0.39) | 87.77 (1.31) | 82.38 (1.60) | 91.43 (0.89) |
> | **PP-GNN** (Our Model) | **89.73 (4.90)** | **88.24 (3.33)** | **56.86 (1.20)** | **67.74 (2.31)** | **82.43 (4.27)** | **54.44** | **61.42 (0.79)** | 69.28 | **80.04 (0.43)** | **78.25 (1.76)** | **89.71 (0.32)** | **89.52 (0.85)** | **85.23 (1.36)** | **92.89 (0.37)** |
> ---
> **Observation:** Our model's standing remains unaffected, across all datasets (other than OGBN-ArXiv). However, we would like to note that improvement gap over baselines has reduced from $\sim$10\% to $\sim$5\%. We will update our paper accordingly with results from new baselines and make appropriate changes in the paper to reflect the observed gap reduction.
>
> $\\\\$
> Regarding comparison against general (unconstrained) FIR filters, we are currently running experiments and will report/update the results once they are ready. We intend to complete this exercise before the end of rebuttal phase.
>
> **More Details:** For ADAGNN and BernNet, we use the authors code and tune it over the hyperparameters as provided in the paper. For ARMA, we use the official PyTorch implementation. As a sanity check, we also tested ARMA on the node classification datasets described in the paper and were able to reproduce similar numbers. We can make this code available.
>
> ---
> #### **References**
> ---
>
> [1] Filippo Maria Bianchi, Daniele Grattarola, Lorenzo Livi, and Cesare Alippi. Graph neural networks with convolutional arma filters. TPAMI, 2021.
>
> [2] Mingguo He, Zhewei Wei , Zengfeng, Hunang, and Hongteng, Xu. BernNet: Learning Arbitrary Graph Spectral Filters via Bernstein Approximation. In NeurIPS 2021
>
> [3] Yushun Dong, Kaize Ding, Brian Jalaian, Shuiwang Ji, and Jundong Li. Graph neural networks with adaptive frequency response filter. In CIKM, 2021.
>
> [4]: Eli Chien, Jianhao Peng, Pan Li, & Olgica Milenkovic. (2021). Adaptive Universal Generalized PageRank Graph Neural Network.

---

> > ### Comment · Reviewer_eu6Q · 2021-11-19
> > **Reviewer Response to Concern A**
> >
> > I thank the reviewers very much for including these new experiments. I still have several concerns.
> >
> > First, while I appreciate the inclusion of ARMA filters, I don't see AdaGNN or BernNET be proper replacements for FIR filters.
> >
> > AdaGNN allows for filters to have higher-order by using the ""layers"" with no activation function but restricts H_k (Phi_k in their paper) to be a diagonal matrix.
> >
> > BernNET is as complete as an arbitrary polynomial, but the paper does not specify how to handle the multi-feature case. Also, it would be interesting if the authors are imposing the "non-negative constraint" that imposes the BernNet paper (which, then again, would constraint the resulting polynomials).
> >
> > Second, it would be necessary to specify what are the hyperparameters used in the tables shown by the authors to be sure that the number of learnable parameters is the same in all cases.
> >
> > For instance, in the ARMA case, what's the number of output features (hidden units)? What's the order of the numerator polynomial? What's the order of the denominator polynomial? How many Jacobi iterations are carried out? How's the inverse computed? How the number of learnable parameters compares to the PP-GNN? It should be the same, with respect to overfitting.
> >
> > Lastly, and out of curiosity, the authors state that "For ARMA, we use the official PyTorch implementation". I have searched the pytorch documentation (https://pytorch.org/docs/stable/index.html) and I have not found an implementation of graph ARMA filters. I would very much appreciate it if the authors can clarify if I misunderstood or not.
> >
> > With respect to implementations of FIR filters, I would recommend ChebNets implementation which would give a full polynomial description. Although the original code has been written in TensorFlow, many adaptations to PyTorch have been carried out, for example:
> > https://github.com/dsgiitr/graph_nets/tree/master/GCN
> >
> > However, if the authors prefer an implementation that closely resembles eq. (1) and the general polynomial \sum_k S^k X H_k, they may be interested in trying:
> > https://github.com/alelab-upenn/graph-neural-networks/blob/master/alegnn/utils/graphML.py
> >
> > Finally, the current GCN implementation actually admits full polynomial descriptions as can be seen in line 174 of
> > https://github.com/tkipf/gcn/blob/master/gcn/layers.py

---

> > > ### Author Response · Authors · 2021-11-22
> > > **Follow-up Response to Concern A [Reviewer eu6Q]**
> > >
> > > We agree with the reviewer that AdaGNN and BernNet are not proper replacements for the general graph filter. However, the comparison was done to show how we fare against other recent spectral methods, which have shown promising results on the Heterophilic datasets.
> > >
> > >
> > > **BernNET is as complete as an arbitrary polynomial, but the paper does not specify how to handle the multi-feature case. Also, it would be interesting if the authors are imposing the "non-negative constraint" that imposes the BernNet paper (which, then again, would constraint the resulting polynomials).**
> > >
> > > BernNET performs a ReLU operation on the polynomial filter coefficients, thereby imposing the non-negative constraint. Our results for BernNET are reported with this setting (default). We observe that PP-GNN outperforms BernNET on several datasets. It seems that simply imposing the non-negativity constraint may not be sufficient. As stated in our follow-up to concern-C, this seems to suggest that the way the regularization is done plays a role in performance.
> > >
> > > $\\\\$
> > >
> > > **Second, it would be necessary to specify what are the hyperparameters used in the tables shown by the authors to be sure that the number of learnable parameters is the same in all cases**
> > >
> > > We use two sets of Hyper-parameters, a) General Hyper-parameters b) Model Specific Hyper-paramters.
> > >
> > > a) We use the same set of general hyper-parameters for all the models in comparison for fair evaluation. These ranges are as follows:
> > >
> > > - Learning Rate: [0.001, 0.002, 0.01, 0.05]
> > > - WeightDecay : [1e-4, 5e-4, 1e-3, 5e-3, 1e-2, 5e-2, 1e-1]
> > > - Dropout: [0.2, 0.3, 0.4, 0.5, 0.6, 0.7, 0.8]
> > >
> > > b) The model specific hyperparameters are described below:
> > >
> > > 1) **PP-GNN:** We use the following hyper-parameter ranges: Order of the Polynomial: [1, 10] in steps of 1,  Partitions (Buckets) [2,3,4,5,10,20], Number of Eigen Value: [32, 64, 128, 256, 512, 1024], $\eta$: [0, 1] sampled uniformly using Optuna, Hidden Dims: [16, 32, 64].
> > >
> > > 2) **BernNet:** We use the model specific hyper-parameters prescribed by the authors of the paper. We describe the hyper-parameters along with their ranges: Propagation Layer Learning Rate: [0.001, 0.002, 0.01, 0.05], Propagation Layer Dropout: [0.2, 0.3, 0.4, 0.5, 0.6, 0.7, 0.8]
> > >
> > >
> > > 3) **AdaGNN:** We use the model specific hyper-parameters prescribed by the authors of the paper. We describe the hyper-parameters along with their ranges: No. of Layers: [2, 4, 8, 16, 32, 128].
> > >
> > > 4) **ARMA:** The ArmaCONV paper does not specify a hyperparameter range in the paper. Hence these are the ranges we have followed: GCS stacks (S)', [1, 2, 3, 4, 5, 6, 7, 8, 9, 10], stacks’ depth(T): [1, 2, 3, 4, 5, 6, 7, 8, 9, 10]. However we only select configurations such that the number of learnable parameters are less than or equal to those in PP-GNN. The input to the ARMAConv layer are the node features and the ouput is the number of classes. This ouput is then passed through a softmax layer. We apologize for not explicitly stating that we obtained our implementation from the official **PyTorch Geometric Library**. We provide a link for the same here:  [LINK HERE](https://pytorch-geometric.readthedocs.io/en/latest/modules/nn.html#torch_geometric.nn.conv.ARMAConv)
> > >
> > > **Q: Are the learnable parameters comparable?**
> > >
> > > 1. **ARMA:** The number of parameters for ARMA are $(T-1)SC^2 + (T+1)SNC + STC$, where N is the number of node features and C is the number of classes, and S and T are Stacks and Depth as described earlier.
> > >
> > > Note: PPGNN has these learnable parameters $NH + HC + H + C + K + 1$, where H is the hidden dimension and K is polynomial order.
> > >
> > > For instance, in the Chameleon dataset, (N,C)=(500,5), following (T,S) pairs make the number of parameters less than that of PP-GNN:
> > > H=64: {1} x [10], {2} x [5], {3} x [3], {4} x [2], (5,1), (6,1)
> > > H=32: {1} x [5], {2} x [2], (3,1)
> > >
> > > The hyperparameter ranges we selected of ARMA are such that the number of parameters of ARMA are lesser than that of PPGNN. We obtained these by solving $(T-1)SC^2 + (T+1)SNC + STC$ <= $NH + HC$
> > >
> > > 2) **BernNet:** The number of learnable parameters are $NH + HC + H + C + K + 1$ which is of the same order as PP-GNN. K here is the polynomial order.
> > >
> > > 3) **AdaGNN:** The number of learnable parameters are $NH + HC + 2*(n_{layer}-1)*H + N + C$ which is almost always (depending on K) greater than PP-GNN. Hence for this case, it was difficult to reduce the number of parameters. [Note: $n_{layer}$ is the number of layers.]
> > >
> > > **Comparison with ChebNet**
> > > Both GPR-GNN and BernNet compare against ChebNets and show improvements in performance on several datasets. Hence, we only compared against GPR-GNN and BernNET. However, if required, we can update our main table to reflect this baseline.
> > >
> > > We thank the reviewer for pointing us to several code references, which helped us in our General FIR implementation (See follow up to concern-C). We hope that we have addressed your concerns regarding the baselines.

---

> > > > ### Comment · Reviewer_eu6Q · 2021-11-22
> > > > **Reviewer's Response to concerns A and C**
> > > >
> > > > I would like to thank the authors very much for their time and effort in responding to my concerns.
> > > >
> > > > Just to clarify:
> > > >
> > > > - ARMA and GFIR have a single layer, with no MLP to follow, while the PP-GNN has a single graph filtering layer followed by two-layer MLPs for classification?
> > > >
> > > > - It is still not clear to me the actual order of the polynomials used for ARMA and GFIR?
> > > >
> > > > Thanks!

---

> > > > > ### Author Response · Authors · 2021-11-23
> > > > > **Follow-up Response to "Reviewer's Response to concerns A and C" [Reviewer eu6Q] (Part-1)**
> > > > >
> > > > > **ARMA and GFIR have a single layer, with no MLP to follow, while the PP-GNN has a single graph filtering layer followed by two-layer MLPs for classification?**
> > > > >
> > > > > In PP-GNN there is a two layered MLP (that transforms the input node features) followed by a single graph filtering layer similar to GPR-GNN. In ARMA and GFIR, like you correctly pointed out, there is only a filtering layer without an MLP.
> > > > >
> > > > >
> > > > > In all these three models, a softmax layer (no weights) is added at the end.
> > > > >
> > > > > To understand MLP's significance, we ran an additional experiment, where we have used a single linear layer, instead of the two layered MLP. We continue to observe competetive (with respect to our original PP-GNN model) performance, across most datasets. The results can be found in Table - 1. Also, the two layer MLP is not the significant contributor towards performance. This can also be seen by comparing GPR-GNN's performance with that of LGC's. LGC can be interpreted as a linear version of GPR-GNN, and achieves comparable performance as GPR-GNN.
> > > > >
> > > > > $\\\\$
> > > > >
> > > > >
> > > > > ---
> > > > > ## Table - 1
> > > > > ---
> > > > >
> > > > > |                      | Computers      | Chameleon    | Citeseer     | Cora         | Squirrel     | Texas        | Wisconsin     |
> > > > > |----------------------|----------------|--------------|--------------|--------------|--------------|--------------|---------------|
> > > > > | GFIR (Unconstrained) | 78.39 (1.09)   | 51.71 (3.11) | 75.83 (1.94) | 87.93 (0.90) | 36.50 (1.12) | 73.24 (6.91) |  77.84 (3.21) |
> > > > > | GFIR (Constrained)   |   79.57 (2.12) | 61.27 (2.42) | 76.24 (1.43) | 87.46 (1.26) | 41.12 (1.17) | 74.59 (4.45) |  79.41 (3.10) |
> > > > > | GPR-GNN              | 82.38 (1.60)   | 62.59 (2.04) | 76.84 (1.69) | 87.77 (1.31) | 46.31 (2.46) | 81.35 (5.32) | 82.55 (6.23)  |
> > > > > | LGC                  | 83.44 (1.77)   | 61.14 (2.07) | 76.96 (1.73) | 88.02 (1.44) | 44.26 (1.49) | 80.20 (4.28) | 81.89 (5.98)  |
> > > > > | PP-GNN (Original)    | 85.23 (1.36)   | 67.74 (2.31) | 78.25 (1.76) | 89.52 (0.85) | 56.86 (1.20) | 89.73 (4.90) | 88.24 (3.33)  |
> > > > > | PP-GNN (Linear)      | 84.27 (1.19)   | 67.88 (1.62) | 77.86 (1.74) | 88.43 (0.69) | 55.11 (1.72) | 85.58 (4.70) | 86.24 (3.23)  |
> > > > >
> > > > > Please note that PP-GNN (Linear) was not thoroughly trained as the PP-GNN (Original) due to time constraint.
> > > > >
> > > > > **Observation:**  The Table above seems to suggest that PP-GNN (Linear) is competitive to PP-GNN (Original). We believe the difference in performance is an artifact of inadequate tuning. We can infer that adding MLP may give marginal improvements over the linear version. This phenomenon is also observed in GPR-GNN. To illustrate this, we can compare GPR-GNN with LGC (linear version of GPR-GNN). We can observe in the above table that GPR-GNN and LGC are comparable in performance.
> > > > >
> > > > > **Note regarding the comparision of GFIR, Linear GPR-GNN and Linear PP-GNN:**
> > > > >
> > > > > GFIR is defined as follows: $\sum_{i=0}^{K}S^kXH_k$
> > > > >
> > > > > The linear GPR-GNN (LGC) model is defined as follows: $\sum_{i=0}^{K}\gamma_k S^kXH$
> > > > >
> > > > > In both of these equations, the dimension of $H$ and $H_{k}$ is $m$ x $c$ where $m$ denotes the number of node features, and $c$ denotes the number of classes. Since PP-GNN is built on GPR-GNN, we get an expression similar to the GPR-GNN expression (with spectral partioning). Hence by making both GPR-GNN and PP-GNN have linear feature transformations, the node feature transformations of linear PP-GNN and linear GPR-GNN are similar to that of GFIR.
> > > > >
> > > > > **Note regarding lower hyperameter search:**
> > > > >
> > > > > For this additional set of experiments (Linear PP-GNN), due to time constraints, we have reduced the ranges for several hyper-parameters of our model, while also reducing the number of Optuna Trials to 50. These reduced ranges are as follows:
> > > > >
> > > > > - Learning Rate: {0.003, 0.005}
> > > > > - Weight_Decay : {0.0001, 0.001}
> > > > > - Dropout: {0.3, 0.5}
> > > > > - Hidden Dims: {32, 64}
> > > > > - Order of the Polynomial: {2, 4}
> > > > > - No. of Partitions (Buckets): {2, 4}
> > > > > - Number of Eigen Value: {256, 1024}
> > > > > - $\eta$: Sampled uniformly between [0, 1] using Optuna
> > > > >
> > > > > Despite this smaller hyper-parameters search space, we are only shy by a few percentage points on the Texas dataset. We believe this difference can be recovered with thorough tuning.

---

> > > > > > ### Comment · Reviewer_eu6Q · 2021-11-23
> > > > > > **Thank you**
> > > > > >
> > > > > > This is very interesting. Thank you.

---

> > > > > > ### Author Response · Authors · 2021-11-23
> > > > > > **Follow-up Response to "Reviewer's Response to concerns A and C" [Reviewer eu6Q] (Part-2)**
> > > > > >
> > > > > > **It is still not clear to me the actual order of the polynomials used for ARMA and GFIR?**
> > > > > >
> > > > > > In our ARMA experiments, we have used the PyTorch Geometric Library Code ([Link Here](https://pytorch-geometric.readthedocs.io/en/latest/modules/nn.html#torch_geometric.nn.conv.ARMAConv)), from the paper: *Filippo Maria Bianchi and Daniele Grattarola and Lorenzo Livi and Cesare Alippi (2019). Graph Neural Networks with convolutional ARMA filters. CoRR, abs/1901.01343.*
> > > > > >
> > > > > >
> > > > > > In the paper, the authors start with equation (6) where they have defined an $ARMA_K$ filter as follows:
> > > > > >
> > > > > > $h_{ARMA_{K}}(\lambda) = \frac{\sum_{k=0}^{K-1}p_{k}\lambda^k}{ 1+ \sum_{k=0}^{K}q_{k}\lambda^k}$
> > > > > >
> > > > > > Eventually, they stack $K$, $ARMA_1$ filters in parallel and then finally add them up as follows (Equation 12 in the paper):
> > > > > >
> > > > > > $\bar{X} = \sum_{k=1}^{K}\sum_{m=1}^{M}\frac{b_k}{1-a_k\mu_m}u_m u_m^TX$
> > > > > >
> > > > > > Writting this in a different way we get:
> > > > > >
> > > > > > $\bar{X} = \sum_{m=1}^{M}\sum_{k=1}^{K}\frac{b_k}{1-a_k\mu_m}u_m u_m^TX$
> > > > > >
> > > > > > Hence from the above equation we can see that for a particular eigenvalue $\mu_m$, we get the following filter response (Equation 13 in the original paper):
> > > > > >
> > > > > > $h_{ARMA_K}(\mu_m) = \sum_{k=1}^{k=K}\frac{b_k}{1-a_k\mu_m}$
> > > > > >
> > > > > >
> > > > > > When we do the above summation, we get:
> > > > > >
> > > > > > $h_{ARMA_K}(\mu_m) = \sum_{k=1}^{k=K}\frac{b_k\prod_{r=1, r\neq k}^{K}(1-a_r\mu_m)}{\prod_{r=1}^{K}(1-a_r\mu_m)}$
> > > > > >
> > > > > > If we look closely at this above expression, and multiply all the numerator terms the maximum numerator polynomial degree we get is $K-1$, and multiply all the denominator terms the maximum denominator polynomial degree we will get is $K$ and the maximum polynomial degree we will get is $K$. This is consistent with equation (6) of the paper. (Note: The authors have stated a particular line regarding the order of the polynomial filter, we copy it as it is: "By summing $K$ $ARMA_1$ filters, it is possible to recover
> > > > > > the analytical form of the $ARMA_{K}$ filter in Eq. (7)"). In the paper equation (7) is defined as:
> > > > > >
> > > > > > $\bar{X} = (I +\sum_{k=1}^{k=K}q_kL^k)^{-1} \sum_{k=1}^{k=K-1}p_kL^kX$
> > > > > >
> > > > > > Hence the maximum order of the numerator polynomial is $K-1$ and the maximum order of the denominator polynomial is $K$. $K$ here refers to the number of ARMA stacks used, which we earlier referred to as 'S'. We vary 'S' in the (1, 10) with step size of 1, while ensuring that the total learnable parameters are $\leq$ learnable parameters of PP-GNN.
> > > > > >
> > > > > > Similarly, for GFIR, we vary the order in range(1, 10) with step sizes of 1.
> > > > > >
> > > > > > $\\\\$
> > > > > > $\\\\$
> > > > > >
> > > > > > ---
> > > > > >
> > > > > > **How we ensured that the learnable parameters are comparable**
> > > > > >
> > > > > > Let N be the number of node features, C be the number of classes and H be the hidden dimension of the MLP of our PP-GNN model. Let S be the number of GCS Stacks and T be the number of layers (Note: we are using the same terms to describe S and T as descried in the paper). Since we wanted to ensure that the number of learnable parameters of PP-GNN to be comparable with that of ARMA (in order to ensure that the reason ARMA is not performing well is not because the number of learnable parameters of ARMA are greater than PP-GNN), we solve the inequality  $(T-1)SC^2 + (T+1)SNC + STC <= NH + HC$ as shown in our "follow-up response to concern A".
> > > > > > Note: N and C is dataset dependent, and we choose H to be the hidden dimension of the best PP-GNN model for that particular dataset. Since S and T, both are unknown, we get feasible set of values for S and T, and we report the numbers on the best possible configuration from that feasible set.
> > > > > >
> > > > > > The GFIR model is represented as: $\sum_{i=0}^{K}S^kXH_k$. The order of the polynomial (K) is determined in such a way that the numbers of learnable parameters of GFIR are comparable to PP-GNN (our model). This is to ensure that the models are comparable in nature (i.e. performance gains of PP-GNN is not because the number of learnable parameters are less than that of GFIR). Hence we solve the following inequality $K \leq \frac{NH + HC} {NC} - 1$ and obtain all the feasible points that satisfy this inequality. We then train multiple GFIR models on these configurations and report the best test accuracy across these configurations (for a particular dataset and its corresponding splits).
> > > > > >
> > > > > > ---
> > > > > >
> > > > > > We hope that our responses clarify the reviewer's confusion regarding the experimental setup of the ARMA and GFIR models.

---

> ### Author Response · Authors · 2021-11-18
> **Response to Concern B [Reviewer eu6Q ]**
>
> **[B] A discussion on the relationship between the proposed method and scattering transforms is necessary, due to the nature of the spectrum partitioning and wavelets. While including numerical simulations comparing with scattering transforms would be very much welcome, it is indeed not necessary and maybe only left for future work.**
>
> We thank the reviewer for pointing out interesting graph scattering transforms related works (example: [1]) that shows different ways to derive informative graph features using spectral information (via wavelets at different multi-resolution scales) facilitating downstream applications and analysis of graph data. We believe our approach is an alternate solution to shape/learn different parts of the spectrum and extract multi-dimensional spectral selective/informative features using labeled data. As suggested by the reviewer, we will include a detailed discussion (in the supplementary material) on how the learned/extracted graph features from these modeling approaches relate in different learning settings (semi-supervised/unsupervised).
>
>
> ---
> #### **References**
> ---
> [1]: F. Gao, G. Wolf, and M. Hirn, "Geometric scattering for graph data analysis," in 36th Int. Conf. Mach. Learning, Long Beach, CA, 15-9 June 2019, pp. 1–10.

---

> > ### Comment · Reviewer_eu6Q · 2021-11-19
> > **Reviewer Response to Concern B**
> >
> > Thank you very much for the response.

---

> ### Author Response · Authors · 2021-11-18
> **Response to Concern C: On Spectral Partitioning & Bank of Filters [Reviewer eu6Q ]**
>
> **[C] The interplay between the bank of FIR graph filters (i.e. the d x e bank of filters) and the partitioning of the spectrum should be explored: how do we know that these partitions cannot be learned in the bank of filters? choosing the intervals acts as a regularizer?**
>
> As the reviewer rightly points out, the multi-dimensional filter output (that is, n x e, where n is the number of nodes and 'e' is the number of output features, derived from multi-dimensional input node features, n x d) that feeds to the final linear classifier layer can be derived in several different ways. Models such as GCN [1], SGCN [2], GPR-GNN [3] (with MLP transformed input X), etc., have already been shown clearly as constrained models (in the filter space) by the reviewer using the general expression:
> $\[H(X;S) = \sum_{k=0}^{K} S^k X H_k\]$
>
> The totally unconstrained $H_k$ is more powerful. However, we expect that learnability challenges like over-fitting to emerge, in particular, when the number of labelled examples is limited (in a relative sense with respect to the number of parameters), as is the case in many practical node classification applications.
>
> Our model using piecewise polynomials can also be written as an instance of the general expression. It is: 1) more powerful/expressive than the GPR-GNN (as shown through our theoretical results) model, 2) offers flexibility to learn complex frequency responses (as needed to get significantly improved performance), and 3) control model complexity (to avoid any over-fitting) via implicit regularization using low-degree piece-wise polynomials over several partitions. From this viewpoint, our solution approach strikes a balance between learning totally unconstrained model and other severely constrained models, enabling to get improved performance.
>
> Finally, as rightly pointed out by the reviewer, the ARMA [4] modelling approach is definitely an important alternative (which we missed to include). We have now included convolutional ARMA model and a couple of other models (BernNet [5] and AdaGNN [6]) for a much more comprehensive comparison with our solution.
>
> We thank the reviewer for providing us with the general expression for filtered final layer output, which enabled us to place/explain our model as we move from overly constrained to unconstrained models. We will include these details in the supplementary section.
>
> $\\\\$
> ---
> #### **References**
> ---
> [1] Thomas N. Kipf, & Max Welling. (2017). Semi-Supervised Classification with Graph Convolutional Networks.
>
> [2] Felix Wu, Tianyi Zhang, Amauri Holanda de Souza Jr. au2, Christopher Fifty, Tao Yu, & Kilian Q. Weinberger. (2019). Simplifying Graph Convolutional Networks.
>
> [3] Eli Chien, Jianhao Peng, Pan Li, & Olgica Milenkovic. (2021). Adaptive Universal Generalized PageRank Graph Neural Network.
>
> [4] Filippo Maria Bianchi, Daniele Grattarola, Lorenzo Livi, and Cesare Alippi. Graph neural networks with convolutional arma filters. TPAMI, 2021.
>
> [5] Mingguo He, Zhewei Wei , Zengfeng, Hunang, and Hongteng, Xu. BernNet: Learning Arbitrary Graph Spectral Filters via Bernstein Approximation. In NeurIPS 2021
>
> [6] Yushun Dong, Kaize Ding, Brian Jalaian, Shuiwang Ji, and Jundong Li. Graph neural networks with adaptive frequency response filter. In CIKM, 2021.

---

> > ### Comment · Reviewer_eu6Q · 2021-11-19
> > **Reviewer Response to Concern C**
> >
> > Thank you very much for the answer. Besides the connection to Theorem 1 (see response to point E), the authors assert in this answer that "he totally unconstrained H_k is more powerful. However, we expect that learnability challenges like over-fitting to emerge, in particular, when the number of labeled examples is limited (in a relative sense with respect to the number of parameters), as is the case in many practical node classification applications."
> >
> > The number of learnable parameters in \sum_k S^k X H_k is Kde, still independent of the size of the graph n. In principle, depending on the choices of e (the number of "hidden units") I don't see that overfitting in this problem is given, and if so, it should be shown in the corresponding training vs. validation losses during the epochs. And if it exists, regularization could be used to address this.
> >
> > In other ways, the argument that "it leads to overfitting" without proving it, does not necessarily sustain itself when the number of learnable parameters can be chosen by the designer. To be more specific, if the number of learnable parameters for the general graph convolution \sum_k S^k X H_k is set to be equal to the number of learnable parameters in the proposed spectral filtering, why would we expect overfitting? I don't see that as a given.
> >
> > In this same vein, when the authors say "our solution approach strikes a balance between learning totally unconstrained model and other severely constrained models, enabling to get improved performance" it is not clear to me that learning the graph convolution in its generality necessarily leads to overfitting, and as such, it should be proved.
> >
> > Last but not least, it is observed that in papers like GCN and SGC, where the graph convolution is heavily constrained, techniques like dropout and L2 regularization are still used to avoid overfitting. So, put it differently, if we are going to see overfitting anyways (as we do in the most constrained cases), why would that preclude the use of the general convolution, especially when the intention is to learn frequency responses beyond low-pass filters?

---

> > > ### Author Response · Authors · 2021-11-22
> > > **Follow-up response to Concern C [Reviewer eu6Q ] (part-1)**
> > >
> > > We demonstrate the use of regularization for General FIR filters (GFIR). Towards this, we build two versions of GFIR: one with regularization (constrained), and the other without regularization (unconstrained). We also compare PP-GNN (our model) as well as GPR-GNN to the General FIR filter model (GFIR), which is described as follows: $\sum_{i=0}^{K}S^kXH_k$.
> > >
> > > We provide further details of the versions of the GFIR models below and report the results in TABLE-1.
> > >
> > > 1. `Unconstrained Setting:` Where we do not impose any regularization constraints such as dropout and L2 regularization.
> > > 2. `Constrained Setting:` Where we impose dropouts as well as L2 regularization.
> > >
> > > Note: The dimensions of $X$ is ($n$ x $m$), and $H_k$ is ($m$ x $c$) where:
> > > 1. $n$ denotes the number of nodes
> > > 2. $m$ denotes the number of features
> > > 3. $c$ denotes the number of classes
> > >
> > >
> > > ---
> > > ### TABLE-1: Comparison with General FIR Filters
> > > ---
> > >
> > > |      Train Acc /Test Acc    |        Computers       |        Chameleon      |        Citeseer       |          Cora         |        Squirrel       |          Texas        |        Wisconsin      |
> > > |:---------------------------:|:----------------------:|:---------------------:|:---------------------:|:---------------------:|:---------------------:|:---------------------:|:---------------------:|
> > > |     GFIR (Unconstrained)    |      78.39   (1.09)    |     51.71   (3.11)    |     75.83   (1.94)    |     87.93   (0.90)    |     36.50   (1.12)    |     73.24   (6.91)    |       77.84 (3.21)    |
> > > |      GFIR (Constrained)     |        79.57 (2.12)    |     61.27   (2.42)    |     76.24   (1.43)    |     87.46   (1.26)    |     41.12   (1.17)    |     74.59   (4.45)    |       79.41 (3.10)    |
> > > |            GPR-GNN          |     82.38   (1.60)     |     62.59   (2.04)    |     76.84   (1.69)    |     87.77   (1.31)    |     46.31   (2.46)    |     81.35   (5.32)    |     82.55   (6.23)    |
> > > |            PP-GNN           |       85.23 (1.36)     |      67.74 (2.31)     |      78.25 (1.76)     |      89.52 (0.85)     |      56.86 (1.20)     |      89.73 (4.90)     |      88.24 (3.33)     |
> > > ---
> > >
> > >
> > > **Observations:**
> > > 1.  Constrained GFIR performs better than the unconstrained version, with performance lifts of up to ~10%. This suggests that regularization is important for GFIR models.
> > > 2. GPR-GNN outperforms the constrained GFIR version. GPR-GNN further restricts the space of graphs explored as compared to GFIR. This suggests that regularization beyond simple L2/dropout kind of regularization (polynomial filter) is beneficial.
> > > 3. PP-GNN performs better than GPR-GNN. Our model slightly expands the space of graphs explored (as compared to GPR-GNN, but lesser than GFIR), while retaining good performance. This suggests that there is still room for improvement on how regularization is done.
> > >
> > > **A note on the number of learnable parameters is provided in the follow-up response (part-2).**
> > >
> > > From these experiments, we can infer that the performance gains that our model achieves over general FIR filters is not just because spectral partitioning induces regularization. The way that regularization is done also seems to matter (because different regularization schemes may lead to different optimal solutions). These observation raise a few open questions:
> > >
> > > 1. Is the regularization induced by spectral partitioning the same as that of L2 regularization, or even drop out, or is it different?
> > >
> > > 2. Are there other ways of regularizations that might give further improvements in performance?
> > >
> > > Both these questions would involve significant research work. While answering these questions are pertinent, it is beyond the scope of this paper. We plan to answer them in our future work.
> > >
> > >
> > > **Response continued in part-2**

---

> > > > ### Author Response · Authors · 2021-11-22
> > > > **Follow-up response to Concern C [Reviewer eu6Q ] (part-2)**
> > > >
> > > > **Note on the hyperparameter ranges/ number of learnable parameters:**
> > > >
> > > > We tune the General FIR filter model across the following hyperparameters:
> > > >
> > > > - Learning Rate: [0.001, 0.003, 0.005, 0.008, 0.01]
> > > > - Weight_Decay : [1e-4, 5e-4, 1e-3, 5e-3, 1e-2, 5e-2, 1e-1]
> > > > - Dropout:  [0.2, 0.3, 0.4, 0.5, 0.6, 0.7, 0.8]
> > > > - $k_{GFIR}$ (Polynomial Order): [1, $K_{constrained}$]  (in steps of 1)
> > > >
> > > > We make sure that $K_{constrained}$ is selected in such a way that the number of learnable parameters that GFIR learns is less than or equal to the number of parameters that our model learns. Below we explain how this is done.
> > > >
> > > > The number of learnable parameters for PP-GNN and GFIR is as follows:
> > > >
> > > > 1. PP-GNN (our model): $mh + hc + h + c + k_{PP-GNN}  + 1$
> > > > 2. GFIR: $(k_{GFIR} + 1)mc$
> > > >
> > > > Where:
> > > > 1. $k_{GFIR}$ and $k_{PP-GNN}$ denote the polynomial order.
> > > > 2. $h$ is the value of the hidden dimension of the first layer of the MLP of our PP-GNN model. (We use a two-layer MLP)
> > > > 3. $m$ is the number of node features
> > > > 4. $c$ is the number of classes
> > > >
> > > > Note that $k_{PP-GNN}$ ranges from [1 to 10, steps of 1], and hence we can approximate the number of learnable parameters of PP-GNN to $h(m  + c)$
> > > >
> > > > To ensure that the number of learnable parameters of GFIR is less than, equal to that of PP-GNN, we solve the following inequality:
> > > >
> > > > $(k_{GFIR} +1)mc \leq h(m + c)$
> > > >
> > > > $k_{GFIR} \leq \frac{h (m + c)}{mc} - 1$
> > > >
> > > > To get the maximum possible value of $k_{GFIR}$ and also to make sure that it is an integer, we set set $k_{GFIR_{max}} = floor(\frac{h (m + c)}{mc} - 1)$. We set the value of $K_{constrained}$ to $k_{GFIR_{max}}$
> > > >
> > > > Note that $m$ and $c$ are dataset dependent values. For all the datasets in consideration (in the above table), the maximum number of classes is 8. We select the value of $h$ (for each split, for each dataset) by observing the hyperparameters of the best performing PP-GNN model (on that dataset and on that split). Please note that the hyperparameter range for $h$ is as follows [16, 32, 64]. Hence even in the worst case (when $h=16$), $K_{constrained}$ would be 1. Note: From our experiments, we have observed that $h \geq 32$ across most datasets across most splits.
> > > >
> > > > In a nutshell, we have ensured that the number of learnable parameters in GFIR are comparable to those in PP-GNN.

---

> ### Author Response · Authors · 2021-11-18
> **Response to Concern D: Computational Complexity & Timing Analysis [Reviewer eu6Q ] [Part-1]**
>
> **[D] The increase in computational complexity due to the computation of eigenvalues should be thoroughly explained and total computational costs should be reported for all architectures.**
>
> We provide a response to this concern in two parts. The first part explains the breakdown of computational complexity for individual blocks (e.g. embedding computation) of our model. The second part provides timing comparison for different models.
>
>
> Listed below is the computational complexity for each piece in our model. Notation n: number of nodes, $|E|$: the number of edges, $A$: symmetric normalized adjacency matrix, $F$: features dimensions, $d$: hidden layer dimension, $C$: number of classes, e* denotes the cost of EVD, $K$: polynomial order/hop order, $l$: number of eigenvalues/vectors in a single partition of spectrum (for implementation, we keep $l$ same for all such intervals), $m$: number of partitions of a spectrum.
>
> - **MLP:** $O(nFd + ndC)$ (2 Layer MLP)
>
> - **GPR-term:** $O(K|E|C)$ + $O(nKC)$. The first term is the cost for computing $A^Kf(X)$ for sparse $A$. The second term is the cost of summation $\sum_kA^kf(X)$.
>
> - **Excess terms for PP-GNN:** $O(mnlC)$. This is obtained by the optimal matrix multiplication present in Equation 3 of the main paper ($\mathbf{U}_i$ is $n\times l$, $H_i(\gamma_i)$ is $l\times l$, $\mathbf{Z}_0()$ is $n\times C$). The additional factor $m$ is because we have $m$ different contiguous intervals/different polynomials. Note that typically 'n' is much larger than 'l'.
>
> - **EVD-term:** $e^{*}$, the complexity for obtaining the eigenvalues/vectors of the adjacency matrix, which is usually very sparse for the observed graphs. Most publicly available solvers for this task utilize Lanczos' algorithm (which is a specific case of a more general Arnoldi iteration). However, the convergence bound of this iterative procedure depends upon the starting vectors and the underlying spectrum (particularly the ratio of the absolute difference of two largest eigenvalues to the diameter of the spectrum) [1, 2, 3]. Lanczos' algorithm is shown to be a practically efficient way for obtaining extreme eigenpairs for a similar and even very large systems. We use ARPACK's built-in implementation to precompute the eigenvalues/vectors for all datasets before training, thus amortizing this cost across training with different hyper-parameters configuration. (See part-2 of our response to this concern)
>
> $\\\\$
> $\\\\$
>
> ---
> #### **References**
> ---
>
> [1] REN-CANG LI. Sharpness in rates of convergence for the symmetric lanczos method. Mathematics of Computation, 79(269):419–435, 2010. ISSN 00255718, 10886842. URL: http://www.jstor.org/stable/40590409
>
> [2] Y. Saad. On the rates of convergence of the lanczos and the block-lanczos methods. SIAM Journal on Numerical Analysis, 17(5):687–706, 1980. doi: 10.1137/0717059. URL: https://doi.org/10.1137/0717059
>
> [3] References of https://en.wikipedia.org/wiki/Lanczos_algorithm

---

> > ### Author Response · Authors · 2021-11-18
> > **Response to Concern D: Computational Complexity & Timing Analysis [Reviewer eu6Q ] [Part-2a]**
> >
> > We perform a comprehensive timing analysis, and report per-component timing for our model along with the average training time for several baselines in TABLES [1, 2]. In TABLE [2], we report the average end to end training time for several models in comparison. This average end to end training time of the model is averaged across 100 configurations, where each configuration is trained using a different set of hyper-parameters (as selected using Optuna framework[1] for hyper-parameter optimization). For PP-GNN, since the EVD computation (computing top and bottom eigen values/vectors) is done \textit{only once} (TABLE 1), we average the EVD time (cost) over 100 configurations (i.e., amortize the cost) and add it to get the effective training time per configuration. We use this measure  to compare time taken by our model with other baselines. (TABLES [2, 3]).
> >
> > To enable relative comparison, we use the time taken by GCN [2] as reference. In TABLE 3, we report scaling factors for the time taken by different models. We observe that PP-GNN is $\sim$4x slower than GCN, $\sim$2x slower than GPR-GNN [3] and  BernNet, $\sim$1.5x slower than ARMA, and $\sim$2x faster than AdaGNN. **[TABLE 3 is available in our follow up response]**
> >
> > $\\\\$
> >
> > ---
> > ### TABLE 1: PP-GNN’s per component timing cost
> > ---
> >
> > | PP-GNN | Avg. Training Time (sec) | EVD Cost (sec) [ONE TIME] | #EV's obtained | Effective Training Time (sec) |
> > |:----------:|:-------------:|:--------:|:---------------:|:--------------:|
> > | **Texas** | 11.89 | 0.007473 | 183 (All Evs) | 11.89 |
> > | **Cornell** | 11.63 | 0.032709 | 183 (All Evs) | 11.63 |
> > | **Wisconsin** | 12.08 | 0.012245 | 251 (All Evs) | 12.08 |
> > | **Chameleon** | 21.44 | 3.718832 | 2048 | 21.48 |
> > | **Squirrel** | 31.38 | 15.81523 | 2048 | 31.54 |
> > | **Cora** | 22.46 | 54.3684 | 2048 | 23.00 |
> > | **Citeseer** | 20.51 | 56.97437 | 2048 | 21.08 |
> > | **Cora-Full** | 63.98 | 155.3041 | 2048 | 65.53 |
> > | **Pubmed** | 52.54 | 256.7104 | 2048 | 55.11 |
> > | **Computers** | 28.63 | 76.27381 | 2048 | 29.39 |
> > | **Photo** | 19.3 | 48.36831 | 2048 | 19.78 |
> > | **Flickr** | 161.16 | 304.1138 | 2048 | 164.20 |
> > | **OGBN-ArXiv** | 189.94 | 412.5042 | 1024 | 194.07 |
> > | **WikiCS** | 27.92 | 65.43761 | 2048 | 28.57 |
> >
> >
> >
> > $\textbf{Table description:}$ Training Time refers to the end to end training time (without eigen decomposition) averaged across 100 trials. EVD cost refers to the time taken to obtain 'x' top and bottom eigenvalues. This 'x' can be found in the ‘Number of EV’s obtained’ column. Since EVD is a one time cost, we average this cost over the total number of trials and add it to the training time. We refer to this cost as the Effective Training Time.
> >
> > $\\\\$
> > $\\\\$
> >
> > ---
> > ### TABLE 2: Training Time (in seconds) across Models
> > ---
> >
> >
> >
> > | Dataset | GPR-GNN [1] | PP-GNN (our model) | MLP | GCN [2] | BernNet [3] | ARMA [4] | AdaGNN [5] |
> > |:---------:|:-------:|:------:|:----:|:------:|:-------:|:------:|:--------:|
> > | **Texas** | 9.27 | 11.89 | 1.08 | 3.46 | 5.59 | 6 | 13.97 |
> > | **Cornell** | 9.41 | 11.63 | 1.06 | 3.69 | 5.37 | 5.51 | 12.56 |
> > | **Wisconsin** | 9.67 | 12.08 | 1.07 | 3.42 | 5.69 | 5.36 | 13.57 |
> > | **Chameleon** | 14.69 | 21.48 | 2.6 | 6.42 | 12.46 | 7.84 | 28.77 |
> > | **Squirrel** | 18.94 | 31.54 | 5.04 | 7.52 | 17.82 | 28.87 | 90.36 |
> > | **Cora** | 12.9 | 23.00 | 1.95 | 5.94 | 12.25 | 10.67 | 22.15 |
> > | **Citeseer** | 10.62 | 21.08 | 3.72 | 4.56 | 9.52 | 19.5 | 35.34 |
> > | **Cora-Full** | 24.98 | 65.53 | 7.77 | 8.01 | 31.26 | 40.21 | 175.58 |
> > | **Pubmed** | 14 | 55.11 | 6.21 | 11.73 | 12.64 | 27.76 | 162.01 |
> > | **Computers** | 7.67 | 29.39 | 2.24 | 6.68 | 7.48 | 27.76 | 118.43 |
> > | **Photo** | 8.58 | 19.78 | 1.68 | 5.1 | 7.95 | 14.34 | 45.46 |
> > | **Flickr** | 42.64 | 164.20 | 21 | 30.4 | 62.11 | 119.3 | 178.7371 |
> > | **ArXiv** | 118.35 | 194.07 | 78.9 | 102.88 | 693.92 | 771.59 | 307.84 |
> > | **WikiCS** | 14.37 | 28.57 | 3.34 | 10.8 | 11.43 | 30.79 | 73.63 |
> >
> >
> >
> > $\textbf{Note:}$ For PP-GNN model we report the effective training time, as described in the description of TABLE 1.
> >
> > $\\\\$
> >
> > ---
> > #### **References**
> > ---
> > [1]: Yu Rong, Wenbing Huang, Tingyang Xu, & Junzhou Huang. (2020). DropEdge: Towards Deep Graph Convolutional Networks on Node Classification.
> >
> > **Note:** Other references are available in the follow up response.

---

> > > ### Author Response · Authors · 2021-11-18
> > > **Response to Concern D: Computational Complexity & Timing Analysis [Reviewer eu6Q ] [Part-2b]**
> > >
> > > ### TABLE 3: Training Time of models relative to the training time of GCN
> > > ---
> > > | Dataset | GPR-GNN [1] | PP-GNN (our model) | MLP | GCN [2] | BernNet [3] | ARMA [4] | AdaGNN [5] |
> > > |:---------:|:-------:|:------:|:----:|:----:|:-------:|:----:|:------:|
> > > | **Texas** | 2.68 | 3.44 | 0.31 | 1.00 | 1.62 | 1.73 | 4.04 |
> > > | **Cornell** | 2.55 | 3.15 | 0.29 | 1.00 | 1.46 | 1.49 | 3.40 |
> > > | **Wisconsin** | 2.83 | 3.53 | 0.31 | 1.00 | 1.66 | 1.57 | 3.97 |
> > > | **Chameleon** | 2.29 | 3.35 | 0.40 | 1.00 | 1.94 | 1.22 | 4.48 |
> > > | **Squirrel** | 2.52 | 4.19 | 0.67 | 1.00 | 2.37 | 3.84 | 12.02 |
> > > | **Cora** | 2.17 | 3.87 | 0.33 | 1.00 | 2.06 | 1.80 | 3.73 |
> > > | **Citeseer** | 2.33 | 4.62 | 0.82 | 1.00 | 2.09 | 4.28 | 7.75 |
> > > | **Cora-Full** | 3.12 | 8.18 | 0.97 | 1.00 | 3.90 | 5.02 | 21.92 |
> > > | **Pubmed** | 1.19 | 4.70 | 0.53 | 1.00 | 1.08 | 2.37 | 13.81 |
> > > | **Computers** | 1.15 | 4.40 | 0.34 | 1.00 | 1.12 | 4.16 | 17.73 |
> > > | **Photo** | 1.68 | 3.88 | 0.33 | 1.00 | 1.56 | 2.81 | 8.91 |
> > > | **Flickr** | 1.40 | 5.40 | 0.69 | 1.00 | 2.04 | 3.92 | 5.88 |
> > > | **ArXiv** | 1.15 | 1.89 | 0.77 | 1.00 | 6.74 | 7.50 | 2.99 |
> > > | **WikiCS** | 1.33 | 2.65 | 0.31 | 1.00 | 1.06 | 2.85 | 6.82 |
> > > | **Average** | 2.03 | 4.09 | 0.50 | 1.00 | 2.19 | 3.18 | 8.39 |
> > > ---
> > >
> > > $\textbf{Description:}$ Relative training time for models in comparison with respect to the GCN model obtained by computing the ratio of the average training time of each model to that of GCN.
> > >
> > > $\\\\$
> > > $\\\\$
> > >
> > > ---
> > > #### **References:**
> > > ---
> > >
> > > [1]: Eli Chien, Jianhao Peng, Pan Li, & Olgica Milenkovic. (2021). Adaptive Universal Generalized PageRank Graph Neural Network.
> > >
> > > [2] Mingguo He, Zhewei Wei , Zengfeng, Hunang, and Hongteng, Xu. BernNet: Learning Arbitrary Graph Spectral Filters via Bernstein Approximation. In NeurIPS 2021
> > >
> > > [3] Filippo Maria Bianchi, Daniele Grattarola, Lorenzo Livi, and Cesare Alippi. Graph neural networks with convolutional arma filters. TPAMI, 2021.
> > >
> > > [4] Yushun Dong, Kaize Ding, Brian Jalaian, Shuiwang Ji, and Jundong Li. Graph neural networks with adaptive frequency response filter. In CIKM, 2021.

---

> > ### Comment · Reviewer_eu6Q · 2021-11-19
> > **Reviewer Response to Concern D**
> >
> > Thank you very much for this detailed answer.
> >
> > Would it be possible to quantify e^* in terms of the size of the matrix/its sparsity? Even if it depends on some choices of the Lanczos algorithm?
> >
> > Also, thanks for clarifying that this can be precomputed before training. This clarification should be included.

---

> ### Author Response · Authors · 2021-11-18
> **Response to Concern E: Clarification in Theorem-1 [Reviewer eu6Q ]**
>
> **[E] I would appreciate it if the authors clarify my concern on Theorem 1.**
>
> In Theorem 4.1, as rightly understood by the reviewer, we attempted to formally state that using the sum of polynomials ($h + h_f$) will have lower error than using just h. We stated $4.1$ for the following purposes: (a) completion and (b) to lead into Theorem $4.2$ and its corollary (which talks about the dimensionality of the graph spaces establishing higher expressivity of the proposed model). In Theorem $4.2$, we are able to show that, using a sum of polynomials explores a larger space of graphs, explaining how the reduction in error showed in Theorem $4.1$ can come from.

---

> > ### Comment · Reviewer_eu6Q · 2021-11-19
> > **Reviewer Response to Concern E**
> >
> > Thank you. How would this be different, then, than from having the aggregation after filter bank, i.e.
> >
> > \sum_k S^k X H_k
> >
> > translates, for a single-feature graph signal, into
> >
> > \sum_{d'=1}^{d} H_{d'e}(S) x^d'
> >
> > where
> >
> > H_{d'e}(S) = \sum_k [H_k]_{d'e} S^k x^d
> >
> > is the regular filtering operation defined in eq. (1).
> >
> > In other words, a traditional convolution in a multi-feature graph signal case (i.e., \sum_k S^k X H_k) also includes a sum of filters. Would Theorem 1 also apply in this case? Would be the benefit, in terms of expressivity, of the proposed spectrum partitioning?

---

> > > ### Author Response · Authors · 2021-11-22
> > > **Follow-up Response to Concern E  [Reviewer eu6Q ]**
> > >
> > > For a general graph filter (whose output is represented as $\sum_k S^k X H_k$), Theorem-1 would still be applicable. i.e. in terms of expressivity, a general graph filter would be more expressive than any other polynomial filter (i.e. PP-GNN as well as GPR-GNN) of the same order. In our follow-up response to concern-C, we made a few observations: 1) Regularization is important, 2) The way the regularization is done is also important. Following our Theorem-2, we established that PP-GNN explores a larger graph space than GPR-GNN. However, this explored space is still less than that explored by GFIR. We believe that our model is able to expand the search space (as compared to GPR-GNN) in a way that doesn't hamper the performance of the model. While understanding exactly how this happens is a pertinent question, we believe that answering such a question is beyond the scope of this work. We would like to address this interesting question as a part of our future work.

---

> ### Author Response · Authors · 2021-11-18
> **Response to Concern F: Suggestion for Paper Title [Reviewer eu6Q ]**
>
> **[F] I believe the title could be improved to better reflect the paper's contributions.**
>
> We agree with the reviewer that the title does not fully reflect the contributions. One possible Title we thought of is: "Effective Polynomial Filter Adaptation for Graph Neural Networks Using Spectral Partitioning". Another possibility is: "A Piecewise Polynomial Filtering Approach for Graph Neural Networks".
>
>
> We sincerely hope that we have addressed all major concerns of the reviewer. We are happy to take care of any further suggestions/queries.

---

> > ### Comment · Reviewer_eu6Q · 2021-11-19
> > **Reviewer Response to Concern F**
> >
> > Thank you. I believe that the inclusion of spectral partitioning in the title would be a good hint. For example "Graph Neural Networks with Polynomial Filters and Spectral Partitioning". But, in any case, this is the authors' decision and, among the options provided, I believe "A Piecewise Polynomial Filtering Approach for Graph Neural Networks" is the best one.

---

### Official Review · Reviewer_SUfF · 2021-10-31

**Correctness:** 4
**Technical Novelty And Significance:** 2
**Empirical Novelty And Significance:** 2
**Recommendation:** 5
**Confidence:** 4

**Main Review:**


Strength
- The proposed architecture is well-motivated.
- The ablation study is extensive.

Weakness
- There are a lot of hyper-parameters to tune. ( The number of partitions, which can be interpreted as the number of filters, is swept in the range [2,3,4,5,10,20]. The polynomial filter order is swept in the range [1,10] in steps of size 1. The number of eigenvalues/vectors are swept in the range [32, 64, 128, 256, 512, 1024].)
- Computational time is unclear. I suggest the authors clearly write the training time for each dataset. Please clarify the eventual computational complexity of the model that's used in the experiments. How many times is your model more expensive than GCN or the polynomial filter?
- It is not clear why the reported performance of ogbn-arxiv is much worse than that in the leaderboard. For instance, the GCN performance reported in Table 2 is 63.48, while the official leaderboard reports (https://ogb.stanford.edu/docs/leader_nodeprop/#ogbn-arxiv) the test accuracy of 0.7174 ± 0.0029.


**Summary Of The Paper:**

In this paper, the authors aim to develop GNN that can better adapt to the given prediction task (both homophily and heterophiliy). Specifically, the authors extend the existing polynomial filter and propose to learn a filter function as a sum of polynomials over different subsets of the eigenvalues. The effectiveness of the proposed GNN architecture is demonstrated on diverse node classification tasks. The ablation studies were carried out to understand the proposed GNN architecture.

**Summary Of The Review:**

I suggest the weak reject to this paper in its current form. I can consider raising my score if all my concerns (raised in the weakness section) are addressed properly.

---

> ### Author Response · Authors · 2021-11-18
> **Response to Weakness-1 [Reviewer SUfF]**
>
> We are grateful to the reviewer for making several important observations in our experiments and making very useful suggestions to improve the quality of the paper. We present below both clarification and a detailed report from additional experiments conducted based on the suggestions.
> $\\\\$
>
> **Note:** We will append this new material as a separate section in our supplementary material.
>
> **[1] There are a lot of hyper-parameters to tune. ( The number of partitions, which can be interpreted as the number of filters, is swept in the range [2,3,4,5,10,20]. The polynomial filter order is swept in the range [1,10] in steps of size 1. The number of eigenvalues/vectors are swept in the range [32, 64, 128, 256, 512, 1024].)**
>
> We described the set of hyper-parameters along with their ranges for PP-GNN and other models in comparison in our paper. As rightly observed by the reviewer, performing a regular grid search on the prescribed ranges will combinatorially explode. Unfortunately, we overlooked to mention a critical detail on how these configurations are swept. We use the Optuna framework [1] for exploring the hyperparameter space and set the number of configurations to be tried to 100. This prevents the hyperparameter search space from blowing up and we found that this order of tuning is sufficient to get the performance reported in the paper. In Table 1, we report the effective training time (includes hyper-parameter optimization) for our model. We can observe that the hyper-parameter tuning cost is reasonable and not very high.
>
>
>
> $\\\\$
>
> ---
> ### TABLE-1: Overall Time Taken by PP-GNN (our model)
> ---
>
> | Dataset | Chameleon | Citeseer | Computers |   Cora   | Cora-Full |  Cornell |  Flickr  |   Photo  |  Pubmed  | Squirrel |   Texas  |  Wikics  | Wisconsin | OGBN-ArXiv |
> |:-------:|:---------:|:--------:|:---------:|:--------:|:---------:|:--------:|:--------:|:--------:|:--------:|:--------:|:--------:|:--------:|:---------:|:----------:|
> |   Time  |  01:23:24 | 01:21:26 |  01:51:41 | 01:30:31 |  04:03:35 | 00:44:00 | 03:39:57 | 01:15:25 | 03:24:29 | 01:59:20 | 00:44:03 | 00:41:04 |  00:46:58 |   05:06:5  |
>
> $\textbf{Description:}$ Effective end-to-end model training time for hyperparameter optimization over 100 configuration (including time taken for obtaining 1024 top and bottom eigenvalues/vectors). Time in HH:MM:SS.
>
> **Machine configuration:** Intel Xeon 2.60Ghz processor, 112GB Ram, Nvidia Tesla P-100 GPU with 16GB of memory
>
>
> $\\\\$
> $\\\\$
> $\\\\$
> ---
> ### References
> ---
> [1]: Takuya Akiba, Shotaro Sano, Toshihiko Yanase, Takeru Ohta, & Masanori Koyama. (2019). Optuna: A Next-generation Hyperparameter Optimization Framework.

---

> > ### Comment · Reviewer_SUfF · 2021-11-18
> > **Response**
> >
> > Thank you for your clarification. This additional set of experiments is super helpful.
> > Unfortunately, taking 1.5 hours on Cora, 5 hours on ogbn-arxiv sound really long to me, and this can be a critical drawback of the approach.

---

> > > ### Author Response · Authors · 2021-11-20
> > > **Response to Weakness-1 (follow-up) [Part-1] [Reviewer SUfF]**
> > >
> > > **Thanks for the extensive investigation of the training time. However, I found these tables misleading. While your
> > > method has a lot of hyper-parameters to tune (hence requires 100 configs), other models (e.g., GCN) do not have many
> > > such hyper-parameters. Hence, it is not fair to amortize the pre-processing time for your method.**
> > >
> > >
> > >
> > > We agree with the reviewer that with 100 trials, models like GCN would explore a larger fraction of the hyper-parameters search space than our model. This search space was prescribed in the paper [1], where the authors have shown that tuning GCNs thoroughly can outperform several existing SOTA models for **homophilic** datasets. Training all the models on these hyper-parameters with 100 Optuna Trials allowed us to convince ourselves (and the reviewer) that we are getting a good performance, not because of some statistical anomaly but the fact that our method holds merit.
> > >
> > >
> > >
> > > With the model's merit established, we can cut down on ranges for several hyper-parameters of our model, while retaining competitive performance across datasets. Below is the reduced hyper-params space:
> > >
> > >
> > >
> > > - Learning Rate: [0.003, 0.005]
> > > - Weight_Decay : [0.0001, 0.001]
> > > - Dropout: [0.3, 0.5]
> > > - Hidden Dims: [32, 64]
> > > - Order of the Polynomial: [2, 4]
> > > - No. of Partitions (Buckets): [2, 4]
> > > - Number of Eigen Value: [256, 1024]
> > > - $\eta$: Sampled uniformly between [0, 1] using Optuna
> > >
> > >
> > >
> > > Please note that we get rid of partition specific hyper-parameters (polynomial coefficients -- gammas). We later show through an ablative study that randomly initializing gammas suffices. We retrain our model using this new set of params described above on a subset of datasets and report the numbers in TABLE-1. Importantly, **we restrict to using 20 Optuna trials.**
> > >
> > >
> > >
> > > $\\\\$
> > >
> > >
> > >
> > > ---
> > > ### TABLE-1: PP-GNN with Reduced Hyper-parameters
> > > ---
> > >
> > >
> > >
> > > | | Texas | Squirrel | Chameleon | Cora-Full (PCA) | Ogbn-arxiv | Citeseer | Pubmed | Cora | Photos (Amazon) |
> > > |---------------------------------|:--------------------------------:|:------------------------------:|:------------------------------:|:---------------------------:|:---------------------:|:--------------------------------:|:------------------------------:|:-------------------------------:|:------------------------------:|
> > > | **PP-GNN** | 85.14 (2.30) | 59.15 (1.91) | 69.10 (1.37) | 60.93 (0.83) | 69.10 | 77.87 (1.93) | 89.43 (0.47) | 88.87 (0.90) | 92.18 (0.58) |
> > > | **Best Performing Baseline** | 84.86 (6.77)* **H2GCN** | 53.50 (0.96) **AdaGNN** | 65.45 (1.17) **AdaGNN** | 61.84 (0.90) **LGC** | 69.37 **GCN** | 77.07 (1.64)* **H2GCN** | 89.59 (0.33)* **H2GCN** | 88.26 (1.32) **TDGNN** | 92.54 (0.28) **TDGNN** |
> > >
> > >
> > >
> > >
> > > $Observation:$ Even with fewer hyper-params, our model continues to perform competitively or better.
> > >
> > >
> > >
> > > Next, we examine the end-to-end training time (including EVD) for this new setting on several datasets and report in TABLE-2.
> > >
> > >
> > > $\\\\$
> > >
> > > ---
> > > ### TABLE-2: End-to-End Training Time (20 Trials)
> > > ---
> > >
> > >
> > >
> > > | Dataset | chameleon | citeseer | computers | cora | corafull | photo | pubmed | squirrel | texas | wisconsin | OGBN-ArXiv |
> > > |:----------:|:---------:|:--------:|:---------:|:--------:|:--------:|:--------:|:--------:|:--------:|:--------:|:---------:|:----------:|
> > > | Time (sec) | 00:03:46 | 00:10:17 | 00:34:37 | 00:05:24 | 00:59:29 | 00:10:31 | 00:57:40 | 00:10:38 | 00:02:27 | 00:02:33 | 01:03:20 |
> > >
> > >
> > >
> > > **Note:** Time reported in HH:MM:SS format.
> > >
> > >
> > >
> > > We can infer from TABLE-2 that the total training time has significantly reduced across all the datasets. This reduction is mainly achieved by tuning only over lower polynomial orders and fixed number of polynomial partitions. The major bottleneck in terms of time in the prior experiment (100 trials) was due to the higher range of both polynomial order and number of polynomial partitions. The higher the polynomial order, and the higher the number of polynomial partitions, the more time it would take to complete a single trial.
> > >
> > >
> > >
> > > In our model, since Eigen Value Decomposition is done only once, it will be quite unfair to add the total EVD cost to every trial. However, we also agree with the reviewer that amortizing the EVD cost across 100 trials would be unfair. We believe that the above table exhibits a fair comparison as any model might require at least 20 trials for proper hyper-parameter optimization.
> > >
> > >
> > > $\\\\$
> > >
> > > **Continued in follow-up response**

---

> > > > ### Author Response · Authors · 2021-11-20
> > > > **Response to Weakness-1 (follow-up) [Part-2] [Reviewer SUfF]**
> > > >
> > > > **Following up from the previous comment.**
> > > >
> > > > As mentioned earlier, we perform an ablative study to show that our model is not sensitive to various initialization schemes.
> > > >
> > > >
> > > > $\textbf{Polynomial (gamma) initialization:}$ We perform an ablative study to understand the effect of polynomial initialization. We vary the initialization scheme in [PPR, NPPR and Random], as described in GPRGNN [2]. We observe that our model (PP-GNN) retains similar performance even with `random' initialization. For brevity, we report results on six datasets (Cora, Citeseer, Squirrel, Chameleon, Texas and Wisconsin). The results can be found in Table 3. Our experiment results suggest that tuning polynomial initialization for each partition can be avoided without compromising on the performance.
> > > >
> > > >
> > > >
> > > > $\\\\$
> > > > $\\\\$
> > > >
> > > >
> > > >
> > > > ---
> > > > ### TABLE-3: PP-GNN with Varying Initialization
> > > > ---
> > > >
> > > >
> > > >
> > > >
> > > > | | Chameleon | Citeseer | Cora | Squirrel | Texas | Wisconsin |
> > > > |---------------------------|----------------------------|--------------------------|----------------------------|----------------------------|----------------------------|---------------------------|
> > > > | **PP-GNN (NPPR)** | 66.73 (1.86) | 78.41 (1.54) | 89.52 (0.85) | 57.93 (1.46) | 88.38 (3.61) | 87.25 (2.96) |
> > > > | **PP-GNN (Random)** | 68.71 (2.47) | 78.28 (1.70) | 89.42 (0.98) | 57.59 (1.56) | 88.38 (3.48) | 88.63 (2.58) |
> > > > | **PP-GNN (PPR)** | 67.74 (2.31) | 78.25 (1.76) | 89.52 (0.85) | 56.86 (1.20) | 89.73 (4.90) | 88.24 (3.33) |
> > > > | **Best Performing Baselines** | 65.45 (1.17) **AdaGNN** | 78.15 (0.74) **ARMA** | 88.26 (1.32)* **TDGNN** | 53.50 (0.96) **AdaGNN** | 84.86 (6.77)* **H2GCN** | 86.67 (4.69)* **H2GCN** |
> > > >
> > > >
> > > >
> > > > $\textbf{Description:}$ Results for PP-GNN using different polynomial (gamma) initializations versus the best performing baseline models. The names of the best performing baseline models are given right below the test accuracy, in the `Best Performing Baselines' row. '*' indicates that the results were borrowed from the corresponding papers.
> > > >
> > > > We will add all the above experimentation details and results to the supplementary material with appropriate remarks about the timing analysis and hyper-parameter search space.
> > > >
> > > > $\\\\$
> > > >
> > > > ---
> > > > #### **References**
> > > > ---
> > > >
> > > >
> > > > [1] Oleksandr Shchur and Maximilian Mumme and Aleksandar Bojchevski and Stephan Günnemann (2018). Pitfalls of Graph Neural Network Evaluation. CoRR, abs/1811.05868
> > > >
> > > >
> > > >
> > > > [2] Eli Chien, Jianhao Peng, Pan Li, & Olgica Milenkovic. (2021). Adaptive Universal Generalized PageRank Graph Neural Network.

---

> ### Author Response · Authors · 2021-11-18
> **Response to Weakness-2 [Reviewer SUfF] (Part-1)**
>
> **[2] Computational time is unclear. I suggest the authors clearly write the training time for each dataset. Please clarify the eventual computational complexity of the model that's used in the experiments. How many times is your model more expensive than GCN or the polynomial filter?**
>
> We perform a comprehensive timing analysis, and report the per-component timing for our model along with average training time for several baselines in TABLES [1, 2]. In TABLE 2, we report the average end to end training time for several models in comparison. This average end to end training time of the model is averaged across 100 configurations, where each configuration is trained using a different set of hyper-parameters (as selected using Optuna framework for hyper-parameter optimization). For PP-GNN, since the EVD computation (computing top and bottom eigen values/vectors) is done $\textit{only once}$ (TABLE 1), we average the EVD time (cost) over 100 configurations (i.e., amortize the cost) and add it to get the effective training time per configuration. We use this measure  to compare time taken by our model with other baselines (TABLES [2, 3]). **[TABLE 3 is available in our follow up response]**
>
> To enable relative comparison, we use the time taken by GCN as reference. In TABLE 3, we report scaling factors for the time taken by different models. We observe that PP-GNN is $\sim$4x slower than GCN, $\sim$2x slower than GPR-GNN and  BernNet, $\sim$1.5x slower than ARMA, and $\sim$2x faster than AdaGNN.
>
> $\\\\$
> $\\\\$
> ---
> ### TABLE 1: PP-GNN’s per component timing cost
> ---
>
>
>
> | PP-GNN (our model) | Avg. Training Time (sec) | EVD Cost (sec) [ONE TIME] | #EV's obtained | Effective Training Time (sec) |
> |:----------:|:-------------:|:--------:|:---------------:|:--------------:|
> | **Texas** | 11.89 | 0.007473 | 183 (All EVs) | 11.89 |
> | **Cornell** | 11.63 | 0.032709 | 183 (All EVs) | 11.63 |
> | **Wisconsin** | 12.08 | 0.012245 | 251 (All EVs) | 12.08 |
> | **Chameleon** | 21.44 | 3.718832 | 2048 | 21.48 |
> | **Squirrel** | 31.38 | 15.81523 | 2048 | 31.54 |
> | **Cora** | 22.46 | 54.3684 | 2048 | 23.00 |
> | **Citeseer** | 20.51 | 56.97437 | 2048 | 21.08 |
> | **Cora-Full** | 63.98 | 155.3041 | 2048 | 65.53 |
> | **Pubmed** | 52.54 | 256.7104 | 2048 | 55.11 |
> | **Computers** | 28.63 | 76.27381 | 2048 | 29.39 |
> | **Photo** | 19.3 | 48.36831 | 2048 | 19.78 |
> | **Flickr** | 161.16 | 304.1138 | 2048 | 164.20 |
> | **OGBN-ArXiv** | 189.94 | 412.5042 | 1024 | 194.07 |
> | **WikiCS** | 27.92 | 65.43761 | 2048 | 28.57 |
>
>
>
> $\textbf{Table description:}$ Training Time refers to the end to end training time (without eigen decomposition) averaged across 100 trials. EVD cost refers to the time taken to obtain 'x' top and bottom eigenvalues. This 'x' can be found in the ‘Number of EV’s obtained’ column. Since EVD is a one time cost, we average this cost over the total number of trials and add it to the training time. We refer to this cost as the Effective Training Time.
>
> ---
>
> $\\\\$
> $\\\\$
> $\\\\$
> ---
> ### TABLE 2: Training Time (in seconds) across Models
> ----
>
>
>
> | Dataset | GPR-GNN[1] | PP-GNN (our model) | MLP | GCN [2] | BernNet [3] | ARMA [4] | AdaGNN [5] |
> |:---------:|:-------:|:------:|:----:|:------:|:-------:|:------:|:--------:|
> | **Texas** | 9.27 | 11.89 | 1.08 | 3.46 | 5.59 | 6 | 13.97 |
> | **Cornell** | 9.41 | 11.63 | 1.06 | 3.69 | 5.37 | 5.51 | 12.56 |
> | **Wisconsin** | 9.67 | 12.08 | 1.07 | 3.42 | 5.69 | 5.36 | 13.57 |
> | **Chameleon** | 14.69 | 21.48 | 2.6 | 6.42 | 12.46 | 7.84 | 28.77 |
> | **Squirrel** | 18.94 | 31.54 | 5.04 | 7.52 | 17.82 | 28.87 | 90.36 |
> | **Cora** | 12.9 | 23.00 | 1.95 | 5.94 | 12.25 | 10.67 | 22.15 |
> | **Citeseer** | 10.62 | 21.08 | 3.72 | 4.56 | 9.52 | 19.5 | 35.34 |
> | **Cora-Full** | 24.98 | 65.53 | 7.77 | 8.01 | 31.26 | 40.21 | 175.58 |
> | **Pubmed** | 14 | 55.11 | 6.21 | 11.73 | 12.64 | 27.76 | 162.01 |
> | **Computers** | 7.67 | 29.39 | 2.24 | 6.68 | 7.48 | 27.76 | 118.43 |
> | **Photo** | 8.58 | 19.78 | 1.68 | 5.1 | 7.95 | 14.34 | 45.46 |
> | **Flickr** | 42.64 | 164.20 | 21 | 30.4 | 62.11 | 119.3 | 178.7371 |
> | **ArXiv** | 118.35 | 194.07 | 78.9 | 102.88 | 693.92 | 771.59 | 307.84 |
> | **WikiCS** | 14.37 | 28.57 | 3.34 | 10.8 | 11.43 | 30.79 | 73.63 |
>
>
>
> $\textbf{Note:}$ For PP-GNN model we report the effective training time, as described in the description of TABLE 1.

---

> > ### Author Response · Authors · 2021-11-18
> > **Response to Weakness-2 [Reviewer SUfF] (Part-2)**
> >
> > ---
> > ### TABLE 3: Training Time of models relative to the training time of GCN
> > ---
> > | Dataset | GPR-GNN [1] | PP-GNN (our model) | MLP | GCN [2] | BernNet [3] | ARMA [4] | AdaGNN [5] |
> > |:---------:|:-------:|:------:|:----:|:----:|:-------:|:----:|:------:|
> > | **Texas** | 2.68 | 3.44 | 0.31 | 1.00 | 1.62 | 1.73 | 4.04 |
> > | **Cornell** | 2.55 | 3.15 | 0.29 | 1.00 | 1.46 | 1.49 | 3.40 |
> > | **Wisconsin** | 2.83 | 3.53 | 0.31 | 1.00 | 1.66 | 1.57 | 3.97 |
> > | **Chameleon** | 2.29 | 3.35 | 0.40 | 1.00 | 1.94 | 1.22 | 4.48 |
> > | **Squirrel** | 2.52 | 4.19 | 0.67 | 1.00 | 2.37 | 3.84 | 12.02 |
> > | **Cora** | 2.17 | 3.87 | 0.33 | 1.00 | 2.06 | 1.80 | 3.73 |
> > | **Citeseer** | 2.33 | 4.62 | 0.82 | 1.00 | 2.09 | 4.28 | 7.75 |
> > | **Cora-Full** | 3.12 | 8.18 | 0.97 | 1.00 | 3.90 | 5.02 | 21.92 |
> > | **Pubmed** | 1.19 | 4.70 | 0.53 | 1.00 | 1.08 | 2.37 | 13.81 |
> > | **Computers** | 1.15 | 4.40 | 0.34 | 1.00 | 1.12 | 4.16 | 17.73 |
> > | **Photo** | 1.68 | 3.88 | 0.33 | 1.00 | 1.56 | 2.81 | 8.91 |
> > | **Flickr** | 1.40 | 5.40 | 0.69 | 1.00 | 2.04 | 3.92 | 5.88 |
> > | **ArXiv** | 1.15 | 1.89 | 0.77 | 1.00 | 6.74 | 7.50 | 2.99 |
> > | **WikiCS** | 1.33 | 2.65 | 0.31 | 1.00 | 1.06 | 2.85 | 6.82 |
> > | **Average** | 2.03 | 4.09 | 0.50 | 1.00 | 2.19 | 3.18 | 8.39 |
> > ---
> >
> > $\textbf{Description:}$ Relative training time for models in comparison with respect to the GCN model obtained by computing the ratio of the average training time of each model to that of GCN.
> >
> > $\\\\$
> > $\\\\$
> > #### **References**
> >
> > [1] Eli Chien, Jianhao Peng, Pan Li, & Olgica Milenkovic. (2021). Adaptive Universal Generalized PageRank Graph Neural Network.
> >
> > [2] Thomas N. Kipf and Max Welling (2016). Semi-Supervised Classification with Graph Convolutional Networks. CoRR, abs/1609.02907.
> >
> > [3] Mingguo He, Zhewei Wei , Zengfeng, Hunang, and Hongteng, Xu. BernNet: Learning Arbitrary Graph Spectral Filters via Bernstein Approximation. In NeurIPS 2021
> >
> > [4] Filippo Maria Bianchi, Daniele Grattarola, Lorenzo Livi, and Cesare Alippi. Graph neural networks with convolutional arma filters. TPAMI, 2021.
> >
> > [5] Yushun Dong, Kaize Ding, Brian Jalaian, Shuiwang Ji, and Jundong Li. Graph neural networks with adaptive frequency response filter. In CIKM, 2021.

---

> > > ### Comment · Reviewer_SUfF · 2021-11-18
> > > **Response**
> > >
> > > Thanks for the extensive investigation of the training time. However, I found these tables misleading. While your method has a lot of hyper-parameters to tune (hence requires 100 configs), other models (e.g., GCN) do not have many such hyper-parameters. Hence, it is not fair to amortize the pre-processing time for your method.

---

> > > > ### Author Response · Authors · 2021-11-20
> > > > **Response to Weakness-2 (follow-up) [Reviewer SUfF]**
> > > >
> > > > Please refer to the comment titled **"Response to Weakness-1 (follow-up) [Part-1] [Reviewer SUfF]"**, where we have addressed comments made on both responses to Weakness 1 & 2.

---

> ### Author Response · Authors · 2021-11-18
> **Response to Weakness-3 [Reviewer SUfF]**
>
>
> **[3] It is not clear why the reported performance of ogbn-arxiv is much worse than that in the leaderboard. For instance, the GCN performance reported in Table 2 is 63.48, while the official leaderboard reports the test accuracy of $0.7174 \pm 0.0029$.**
>
> We thank the reviewer for pointing us to the discrepancy in GCN's test accuracy for the OGBN-ArXiv dataset with reference to numbers reported in the leader board [1]. We present a detailed explanation below as we investigated the problem thoroughly.
>
> First, We note that we used the original GCN code (in TensorFlow) by Thomas Kipf and Max Welling [2] while running our experiments. We did more extensive hyperparameter tuning as an attempt to match the number (i.e., $0.7174$) reported in the leader board [1], however, we were able to increase the test accuracy to only 65.53\%, which is still much lesser.
>
> Next, we tested with two other GCN implementations: (a) the implementation used in the leader board code and (b) GCN implementation from the authors of the GPR-GNN paper [3]. Our results can be found in the TABLE 1. Note that these implementations are in PyTorch.
>
> Besides different implementations of GCN, we find that batch normalization is done in the implementation used in the GCN-leader board code. Our experiment results on the OGBN-Arxiv data are given in TABLE 1. We see that the performance numbers vary depending on the implementation. Also, batch normalization helps to get nearly 2% improvement for the OGBN-ArXiv dataset.
>
> Please note that we did not perform batch normalization in all our experiments (for all models and datasets). For OGBN-ArXiv dataset, we find a performance gap of 2% in results reported in the main paper (Table 2, Section 5). For instance, we can observe from this table that GPR-GNN and PP-GNN (our model) achieves 68.44% and 69.28% respectively.
>
> Given these findings, if it is crucial we would be happy to rerun our GCN baseline for all datasets depending on reviewer's suggestion of any specific implementation.
>
> $\\\\$
> ---
> ### TABLE-1: GCN results with different implementations
> ---
> |              OGBN-ArXiv             |      Framework |     Batch Norm |     Test Acc |
> |:-----------------------------------:|:--------------:|:--------------:|:------------:|
> |          GCN (Kipf and Welling)     |     TensorFlow |        True    |       65.84  |
> |          GCN (Kipf and Welling)     |     TensorFlow |        False   |       65.53  |
> |          GCN (Leaderboard Code)     |       PyTorch  |        True    |       71.88  |
> |     GCN Stanford (Leaderboard Code) |       PyTorch  |        False   |       70.23  |
> |              GCN (GPR-GNN)          |       PyTorch  |        True    |       71.05  |
> |              GCN (GPR-GNN)          |       PyTorch  |        False   |       69.37  |
> ---
>
> $\\\\$
> We sincerely believe that we have taken reviewer's all suggestions into account, and we will include all the above details in the revised version.
> $\\\\$
>
> #### **References**
>
> [1] https://ogb.stanford.edu/docs/leader_nodeprop/#ogbn-arxiv
>
> [2] Thomas N. Kipf, & Max Welling. (2017). Semi-Supervised Classification with Graph Convolutional Networks.
>
> [3] Eli Chien, Jianhao Peng, Pan Li, & Olgica Milenkovic. (2021). Adaptive Universal Generalized PageRank Graph Neural Network.

---

> > ### Comment · Reviewer_SUfF · 2021-11-18
> > **Response**
> >
> > I thank the authors for the investigation. My guess of why the implementation Kipf and Welling did not work is possibly because they did not explicitly convert the directed graph into an undirected graph.
> >
> > Otherwise, it'd be good to build your method on top of one of the strong leaderboard submissions and show the improvement over that.

---

> > > ### Author Response · Authors · 2021-11-20
> > > **Response to Weakness-3 (follow-up)  [Reviewer SUfF]**
> > >
> > > [1] **I thank the authors for the investigation. My guess of why the implementation Kipf and Welling did not work is possibly
> > > because they did not explicitly convert the directed graph into an undirected graph**
> > >
> > >
> > >
> > > While the Kipf and Welling implementation does not explicitly convert the directed graph into an undirected one, we are
> > > doing this explicit conversion as a preprocessing step and feed this graph. We have also compared the
> > > GCN implementation from Kipf and Welling as well as the GCN implementation from the authors of the GPR-GNN [1]. This comparison can be found in TABLE-1.
> > >
> > >
> > >
> > > $\\\\$
> > >
> > >
> > >
> > > ---
> > > ### TABLE-1: Comparing different GCN implementations
> > > ---
> > >
> > >
> > >
> > >
> > > | | Texas | Wisconsin | Squirrel | Chameleon | Cornell | Cora-Full (PCA) | Ogbn-arxiv | Citeseer | Pubmed | Cora | Computer | Photos |
> > > |:-----------------------------------:|:---------------------:|:---------------------:|:---------------------:|:---------------------:|:---------------------:|:----------------------:|:-----------------:|:---------------------:|:---------------------:|:---------------------:|:---------------------:|:---------------------:|
> > > | **GCN (GPR-GNN, PyTorch)** | 59.73 (4.89) | 58.82 (4.89) | 47.78 (2.13) | 62.83 (1.52) | 60.00 (4.90) | 59.63 (0.86) | 69.37 | 76.47 (1.34) | 88.41 (0.46) | 87.36 (0.91) | 82.50 (1.23) | 90.67 (0.68) |
> > > | **GCN (Kipf and Welling, TensorFlow)** | 61.62 (6.14) | 53.53 (4.73) | 46.04 (1.61) | 61.43 (2.70) | 62.97 (5.41) | 45.44 (1.01) | 63.48 | 76.47 (1.33) | 87.86 (0.47) | 86.27 (1.34) | 78.16 (1.85) | 86.38 (1.71) |
> > >
> > >
> > >
> > >
> > > We can see that both the TensorFlow and the PyTorch implementations lead to a similar performance for most of the datasets. However, there are some differences in performance for some datasets (Wisconsin, Cora-Full (PCA), OGBN-ArXiv, Computer and Photos). We have ensured that hyperparameters, preprocessing steps etc., are same across both these implementations. We believe that these differences can be attributed to the different internal workings of TensorFlow and PyTorch. We would like the reviewer to note that while an investigation is required to understand the differences in underlying Deep Learning Frameworks, such a study is beyond the scope of this work.
> > >
> > > $\\\\$
> > > $\\\\$
> > >
> > > [2] **Otherwise, it'd be good to build your method on top of one of the strong leaderboard submissions and show the improvement over that.**
> > >
> > >
> > > From the results in our main paper (Section 5, Table 2) we can observe that ,models like GCNs perform well on the Homophilic Datasets. However they tend to perform poorly on the Heterophilic datasets.
> > >
> > >
> > >
> > > One of the research goals of our work is to develop a **robust** model, i.e. a model that works across datasets with varying levels of Homophily. In this regard, we built our model over GPR-GNN [1], which shows considerable improvements over different types of datasets. (Our implementation is built on top of GPR-GNN's codebase)
> > >
> > >
> > >
> > > Robustness of models (across datasets with varying levels of homophily) is still an ongoing research topic. Our results on several Homophilic datasets show that we perform competitively to GCN like models. Taking it beyond and improving over GCN like models on Homophilic datasets is the direction we wish to pursue in the future.
> > >
> > >
> > > $\\\\$
> > > $\\\\$
> > >
> > > ---
> > > #### **References**
> > > ---
> > >
> > >
> > >
> > > [1] Eli Chien, Jianhao Peng, Pan Li, & Olgica Milenkovic. (2021). Adaptive Universal Generalized PageRank Graph Neural Network.

---

### Official Review · Reviewer_kGGE · 2021-11-01

**Correctness:** 3
**Technical Novelty And Significance:** 4
**Empirical Novelty And Significance:** 3
**Recommendation:** 5
**Confidence:** 4

**Main Review:**

I found this paper enjoyable to read, and appreciated the authors' approach to the filter design problem.
Although the goal of the paper is to develop methods for node classification with graph neural networks, its core ideas are rooted in graph signal processing.
In that regard, this work's contributions go beyond those of interest to the graph machine learning community, also finding interest for those in the graph signal processing community.
However, I have significant concerns regarding the efficiency of the filtering approach, as well as the actual value of the theoretical analysis.

Although I can vaguely see how the efficient variant of your filtering program works, I am not fully convinced of its actual efficiency.
In particular, evaluating the filter on the low-pass and high-pass components is still only defined in the spectral domain.
Typically, although filters are easily defined in the spectral domain, it is more convenient to apply them in a spatial sense, via some low-order matrix polynomial.
Of course, the polynomial splines used here do not yield filter functions that are represented as low-order matrix polynomials: this is the whole point of the paper, that piecewise polynomials of low-order can do better than global polynomials of high-order in many cases!
In that case though, filtering can only be done in the spectral domain.
What I would like to see, then, is how these filters are actually implemented, in a way similar to Equation (1).
Although you remark on how this can be done using Lanczos iteration, I would like to see a more precise specification of the filtering operation.

I appreciated the statements of the analysis section, as it helped explain that piecewise polynomials are strictly more expressive than global polynomials in some sense.
However, I do not think that your claims of superiority over regular graph filters are necessarily fair.
I would like to draw attention to the claim following Corollary 4.2.1: "the dimension of the space [of filters] increases significantly by using just two adaptive filters."
While this is true, this downplays the complexity of implementing these filters.
The most notable property of normal graph filters of degree $k$ is that they are strictly dependent on the $k$-hop neighborhoods of each node (or $k+1$, depending on what shift operator is used).
This means that for low-order graph filters, it is easy to implement them on any given graph.
This is the reason why we like graph filters of low degree: because they are localized!
However, in the proposed approach, you claim to also be using low degree polynomials, while achieving significantly higher expressivity.
While it looks like this is the case on the surface, I am not convinced that this actually translates into anything useful.
A graph filter of degree $k$ applied to the low and high-pass partitions of the spectrum must only operate in the spectral domain: its spatial implementation will almost certainly not be localized to $k$-hop neighborhoods.
In this sense, it is almost obvious that your filters will be more expressive, since they are indirectly using much larger neighborhoods of the nodes, merely having the restriction of being implementable as low-degree polynomials in a particular subset of the spectrum.
This ties back to my concerns about the implementation of these filters: what does this look like in the spatial domain?
What are the localization properties?
The current explanation obscures these properties.

**Summary Of The Paper:**

This paper considers the problem of designing polynomial graph filters for use in graph neural networks.
The motivation for doing so stems from the notions of homophily and heterophily in labeling the nodes of a graph: the former requiring the use of low-pass filters, and the latter requiring the use of high-pass filters.
Here, the authors consider the design of spline polynomials for spectral-domain filtering, as opposed to the typical ``global'' polynomials used in graph filter design.
That is, they first partition the spectrum of the graph matrix into low-pass, band-pass, and high-pass components, learn polynomials over each partition, and then combine these polynomials, enforcing continuity at the boundary of each interval via a penalty function.

Of course, doing this requires a full eigendecomposition of the graph matrix, which has high complexity.
To ameliorate this issue, the authors propose an efficient variant of their spline spectral filter.
This approach only uses a coarse partition of the spectrum, rather than using the full eigendecomposition.

**Summary Of The Review:**

To summarize: I am currently under the impression that the apparent complexity-expressivity tradeoff is not as significant as the authors claim.
Although it is apparent that that filtering operations are implemented as low-order polynomials over a partition of the spectrum, their piecewise nature completely destroys the locality inherent to low-degree graph filters, which is why such filters are useful in the first place.
This is mere speculation on my part, but I suspect that this method would, for instance, fail to generalize well to graphs other than the one it is trained on, even if the structure is somewhat similar.
Based on these concerns, I do not recommend the acceptance of this paper.

---

> ### Author Response · Authors · 2021-11-18
> **Response to Comment-1 [Reviewer kGGE] (Part-1): Filtering operation and efficiency of proposed model**
>
> [1] **Although I can vaguely see how the efficient variant of your filtering program works, I am not fully convinced of its actual efficiency.** In particular, evaluating the filter on the low-pass and high-pass components is still only defined in the spectral domain. Typically, although filters are easily defined in the spectral domain, it is more convenient to apply them in a spatial sense, via some low-order matrix polynomial. Of course, the polynomial splines used here do not yield filter functions that are represented as low-order matrix polynomials: this is the whole point of the paper, that piecewise polynomials of low-order can do better than global polynomials of high-order in many cases! In that case though, filtering can only be done in the spectral domain. **What I would like to see, then, is how these filters are actually implemented, in a way similar to Equation (1). Although you remark on how this can be done using Lanczos iteration, I would like to see a more precise specification of the filtering operation.**
>
> $\\\\$
>
> We provide more details to explain the filtering operation. We first introduce some notation below.
>
> Let $\tilde{U_i} = U_{\sigma_i}$ denote the eigen submatrix for the partition $\sigma_i$. The corresponding low/high frequency adjacency matrices can be written as
>
> $\widetilde{A_i }^{(l)}= \tilde{U_i}^{(l)}\Lambda^{(l)}_{\sigma_i}\tilde{U_i}^{(l)T},   \sigma_i \in S^l $ and
>
> $\widetilde{A_i }^{(h)}= \tilde{U_i}^{(h)}\Lambda^{(h)}_{\sigma_i}\tilde{U_i}^{(h)T},   \sigma_i \in S^h $ respectively.
>
> An Equation similar to Equation (1) in our paper can be derived for our model by substituting Equation (4) from the paper into Equation (3). On substitution, we get:
>
> $\\\\$
> ---
> #### **Equation (1)**
> ---
>
> $Z = \eta_l\sum_{\sigma_i \in S^l}\sum_{j=1}^{k_1}\gamma_{ij}^{(l)} (\widetilde{A_i }^{(l)})^{j} Z_0(X;\theta)$ + $\eta_h\sum_{\sigma_i \in S^h}\sum_{j=1}^{k_2}\gamma_{ij}^{(h)} (\widetilde{A_i }^{(h)})^{j} Z_0(X;\theta)$ + $\eta_{gpr}\sum_{j=1}^{k_3}\gamma_{j}\widetilde{A}^j Z_0(X;\theta)$
>
> ---
> $\\\\$
> $U_{(\sigma_i)}^{(l)}$, $\Lambda_{(\sigma_i)}^{(l)}$ is the set of eigenvectors/values in the partition $\sigma_i$  of the low frequency components and $U_{(\sigma_i)}^{(h)}$, $\Lambda_{(\sigma_i)}^{(h)}$ is the set of eigenvectors/values in the partition $\sigma_i$ of the high frequency components and $\tilde{A}$ (See Equation 1 in the main paper).
>
> $\\\\$
> $\\\\$
>
> In our filter implementation, we pre-compute the top and bottom eigenvalues/vectors of $\tilde{A}$  and use them to compute partition specific node embeddings. Note that PP-GNN embedding (Equation 1 above) can be rewritten as a weighted combination of embeddings corresponding to low and high spectral partitions and GPR-GNN embedding as shown below.
>
> $\\\\$
> ---
> #### **Equation (2)**
> ---
>
> $Z = \eta_lZ_l + \eta_hZ_h + \eta_{gpr}Z_{gpr}$, where
>
> $Z_l = \sum_{\sigma_i \in S^l} \tilde{U_i}^{l}  H^{(l)}_{\sigma_i} \tilde{U_i}^{(l)T} Z_0(X;\theta)$
>
> $Z_h = \sum_{\sigma_i \in S^h} \tilde{U_i}^{h}  H^{(h)}_{\sigma_i} \tilde{U_i}^{(h)T} Z_0(X;\theta)$
>
> $Z_{gpr} = \sum_{j=1}^{k_3}\gamma_{j}\widetilde{A}^j Z_0(X;\theta)$
>
> ---
>
> where $H_{\sigma_i}^{(l)} = \sum_{j=1}^{k_1}\gamma_{ij}^{(l)} (\Lambda^{(l)}_{\sigma_i})^{j}$
>
> and $H_{\sigma_i}^{(h)} = \sum_{j=1}^{k_2}\gamma_{ij}^{(h)} (\Lambda^{(h)}_{\sigma_i})^{j}$  form the effective low and high frequency component filters.
>
> ---
> $\\\\$
> ## Computational Complexity
>
> **Notation:** n: number of nodes, $|E|$: the number of edges, $A$: symmetric normalized adjacency matrix, $F$: features dimensions, $d$: hidden layer dimension, $C$: number of classes, e* denotes the cost of EVD, $K$: polynomial order/hop order, $l$: number of eigenvalues/vectors in a single partition of spectrum (for implementation, we keep $l$ same for all such intervals), $m$: number of partitions of a spectrum.
>
> - **MLP:** $O(nFd + ndC)$ (2 Layer MLP)
>
> - **GPR-term:** $O(K|E|C)$ + $O(nKC)$. The first term is the cost for computing $A^Kf(X)$ for sparse $A$. The second term is the cost of summation $\sum_kA^kf(X)$.
>
> - **Excess terms for PP-GNN:** $O(mnlC)$. This is obtained by the optimal matrix multiplication present in Equation 3 of the main paper ($\mathbf{U}_i$ is $n\times l$, $H_i(\gamma_i)$ is $l\times l$, $\mathbf{Z}_0()$ is $n\times C$). The additional factor $m$ is because we have $m$ different contiguous intervals/different polynomials. Note that typically 'n' is much larger than 'l'.
>
> - **EVD-term:** $e^{*}$, the complexity for obtaining the eigenvalues/vectors of the adjacency matrix. Most publicly available solvers utilize Lanczos' algorithm. We use ARPACK's built-in implementation to precompute the eigenvalues/vectors before training, thus amortizing this cost across training with different hyper-parameters configuration.
>
> Beyond GPR-GNN's cost, we incur additional cost corresponding to computation of PP-GNN's excess term and EVD. More details on time complexity is provided in the follow up response (Part-2).

---

> > ### Author Response · Authors · 2021-11-18
> > **Response to Comment-1 [Reviewer kGGE] (Part-2)**
> >
> > We perform additional studies to understand PP-GNN's per component timing and then compare the effective training time against GPR-GNN's training time.
> >
> > Training Time refers to the end to end training time (without eigen decomposition) averaged across 100 trials. Each trial corresponds to an instance of hyper-parameter configuration. EVD cost refers to the time taken to obtain 'k' top and bottom eigenvalues. This 'k' can be found in the ‘Number of EV’s obtained’ column. Since EVD is a one time cost, we average this cost over the total number of trials and add it to the training time. We refer to this cost as the Effective Training Time. In Table-1, we provide a comprehensive per component timing cost for PP-GNN.
> >
> > ---
> > ### TABLE 1: PP-GNN’s per component timing cost
> > ---
> >
> > | PP-GNN | Avg. Training Time (sec) | EVD Cost (sec) [ONE TIME] | #EV's obtained | Effective Training Time (sec) |
> > |:----------:|:-------------:|:--------:|:---------------:|:--------------:|
> > | **Texas** | 11.89 | 0.007473 | 183 (All Evs) | 11.89 |
> > | **Cornell** | 11.63 | 0.032709 | 183 (All Evs) | 11.63 |
> > | **Wisconsin** | 12.08 | 0.012245 | 251 (All Evs) | 12.08 |
> > | **Chameleon** | 21.44 | 3.718832 | 2048 | 21.48 |
> > | **Squirrel** | 31.38 | 15.81523 | 2048 | 31.54 |
> > | **Cora** | 22.46 | 54.3684 | 2048 | 23.00 |
> > | **Citeseer** | 20.51 | 56.97437 | 2048 | 21.08 |
> > | **Cora-Full** | 63.98 | 155.3041 | 2048 | 65.53 |
> > | **Pubmed** | 52.54 | 256.7104 | 2048 | 55.11 |
> > | **Computers** | 28.63 | 76.27381 | 2048 | 29.39 |
> > | **Photo** | 19.3 | 48.36831 | 2048 | 19.78 |
> > | **Flickr** | 161.16 | 304.1138 | 2048 | 164.20 |
> > | **OGBN-ArXiv** | 189.94 | 412.5042 | 1024 | 194.07 |
> > | **WikiCS** | 27.92 | 65.43761 | 2048 | 28.57 |
> >
> > ---
> >
> > $\\\\$
> >
> > Next, we compare the average training time taken by PP-GNN and GPR-GNN, and report the absolute time and scaling factor (wrt to GPR-GNN) in Table-2.
> >
> > ---
> > ### TABLE 2: Average Training Time (in seconds) and Scaling Factor
> > ----
> >
> > | Dataset | GPR-GNN [1] | PP-GNN (our model) | Scaling Factor|
> > |:---------:|:-------:|:------:|:------:|
> > | **Texas** | 9.27 | 11.89 | 1.28 |
> > | **Cornell** | 9.41 | 11.63 | 1.24 |
> > | **Wisconsin** | 9.67 | 12.08 | 1.25 |
> > | **Chameleon** | 14.69 | 21.48 | 1.46 |
> > | **Squirrel** | 18.94 | 31.54 | 1.67 |
> > | **Cora** | 12.9 | 23.00 | 1.78 |
> > | **Citeseer** | 10.62 | 21.08 | 1.98 |
> > | **Cora-Full** | 24.98 | 65.53 | 2.62 |
> > | **Pubmed** | 14 | 55.11 | 3.94 |
> > | **Computers** | 7.67 | 29.39 | 3.83 |
> > | **Photo** | 8.58 | 19.78 | 2.31 |
> > | **Flickr** | 42.64 | 164.20 | 3.85 |
> > | **ArXiv** | 118.35 | 194.07 | 1.64 |
> > | **WikiCS** | 14.37 | 28.57 | 1.99 |
> >
> > $\textbf{Note:}$ For PP-GNN model we report the effective training time, as described in the description of TABLE 1. The Scaling Factor column is reported by computing the ratio of PP-GNN's training time to GPR-GNN's training time.
> >
> > **Observation:** We observe that PP-GNN is slower than GPR-GNN by a factor of 1.25x to 4x. On an average PP-GNN is slower than GPR-GNN by a factor of 2.2x with a standard deviation of 0.98x.
> >
> > We hope that our response on computational complexity and timing comparison with GPR-GNN addresses the concerns of the reviewer regarding our model's efficiency.

---

> > > ### Comment · Reviewer_kGGE · 2021-11-29
> > > **Thank you for the response**
> > >
> > > I appreciate the authors' response to my comments, and feel they did a good job addressing my concerns about efficiency. They have demonstrated that in practice, the one-time cost of the EVD is not debilitating. I think the final version of the paper should be very clear about the tradeoff of locality, making the purpose for using low-degree polynomials more clear.

---

> ### Author Response · Authors · 2021-11-18
> **Response to Comment-2 [Reviewer kGGE]**
>
> **[1] However, I do not think that your claims of superiority over regular graph filters are necessarily fair. I would like to draw attention to the claim following Corollary 4.2.1: "the dimension of the space [of filters] increases significantly by using just two adaptive filters."**
>
> We are afraid that there is some misunderstanding and we provide a clarification. Our intent with the statement "the dimension of the space [of filters] increases significantly by using just two adaptive filters." was not to claim superiority over regular graph filters. Our goal was to illustrate that modeling the filter learning problem as piece-wise polynomial learning gives us access to larger graph space than what one would get with just a single polynomial. This helps us to explain/achieve performance improvements that we observe with our model. Since there is a possibility of confusion as pointed out by the reviewer, we will explicitly add a subsection to address this in detail in the supplementary material.
>
> [2]**While this is true, this downplays the complexity of implementing these filters.**
>
> We address this concern in our response to comment-1.

---

> ### Author Response · Authors · 2021-11-18
> **Response to Comment-3 [Reviewer kGGE] (Part-1): Localization property, low-degree polynomials, efficiency**
>
> **[1] The most notable property of normal graph filters of degree  is that they are strictly dependent on the k-hop neighborhoods of each node (or (k+1), depending on what shift operator is used). This means that for low-order graph filters, it is easy to implement them on any given graph. This is the reason why we like graph filters of low degree: because they are localized! However, in the proposed approach, you claim to also be using low degree polynomials, while achieving significantly higher expressivity. While it looks like this is the case on the surface, I am not convinced that this actually translates into anything useful. A graph filter of degree  applied to the low and high-pass partitions of the spectrum must only operate in the spectral domain: its spatial implementation will almost certainly not be localized to k-hop neighborhoods. In this sense, it is almost obvious that your filters will be more expressive, since they are indirectly using much larger neighborhoods of the nodes, merely having the restriction of being implementable as low-degree polynomials in a particular subset of the spectrum. This ties back to my concerns about the implementation of these filters: what does this look like in the spatial domain? What are the localization properties? The current explanation obscures these properties.**
>
> We are afraid that there might be some misunderstanding on locality and use of low-degree polynomials with our model. We regret if this happened due to lack of proper explanation on our part. Here, we explain with more details to answer questions and clarify doubts raised by the reviewer.
>
> **On Locality:** We agree with the reviewer that we do not retain locality, but we would like to clarify that we do NOT claim any such property for our model and we will make this point more explicit.
>
> **Use of low-degree polynomials with spectral partitions:** Our intention to use/highlight the low degree polynomial is different. From filter function/response approximation viewpoint, we aim at approximating complex shaped function over the entire eigen spectrum using several low-degree polynomials with each polynomial dedicated to approximate a small contiguous interval (i.e., a partition) in the eigen spectrum, as used in piece-wise/spline function approximation methods (e.g. using cubic-spline models) and low-degree polynomials are sufficient to fit complex filter functions. Furthermore, since we need to learn the polynomial coefficients, it also helps to control the number of parameters by using low degree polynomials to control model complexity and avoid over-fitting in transductive setting with limited labeled data.
>
> The question that arises is: *what are the implications of spectral partitioning and fitting using piece-wise polynomials?*
>
> **Implication of spectral partitioning on locality:** First, since we work with eigenvalues/vectors at the low and high end spectrum, the "implicit" matrices (see the expressions for $\tilde{A} _i^{(l)}$  and $\tilde{A}_i^{(h)}$ in our response to the previous comment) are no longer local (as rightly observed by the reviewer, even when low degree polynomials are used for individual partitions). In other words, embedding of all nodes ($Z_0$)  are used to compute embedding of each node at the filter output.
>
> **Computational Cost and Timing Implications:** Next, our model incurs eigendecomposition cost and embedding computation cost for each spectral partition. Therefore, it is computationally more expensive than the GPR-GNN model. We did both computational complexity analysis (see response to the previous comment) and timing comparison of our model with several other competitive SoTA models including GPR-GNN. Two key observations are: (a) eigendecomposition cost is one-time and can be amortized over training across multiple hyperparameter configurations, as needed for hyperparameter optimization and (b) our model is slower by GPR-GNN only by a factor of $2.2X$ (refer to our response to comment-1, part-2). But, the benefit we get as accuracy improvement is quite significant, justifying the excess time taken when high accuracy is the key requirement and affordable.
>
> **Continued in a follow up response**

---

> > ### Author Response · Authors · 2021-11-18
> > **Response to Comment-3 [Reviewer kGGE] (Part-2): Localization property, low-degree polynomials, efficiency**
> >
> > **Continuation of Response to Comment-3 (Part-1)**
> >
> >
> > **Using only local neighborhood, Efficiency and Learning complex shaped filter response:** Finally, for heterophilic graphs, it helps to make use of *both* low and high frequency signals, and potentially, making use of signals from long range nodes (via eigenvectors computed from the entire graph) to achieve enhanced accuracy performance. But, this requires learning more complex shaped filter response as demanded by the task at hand.
> >
> > The GPR-GNN model [1] is efficient due to use of smaller local neighborhood with low-degree polynomials and sparse matrix multiplication (depending on sparsity of the adjacency matrix). GPR-GNN is able to incorporate high frequency components (as illustrated through analysing the polynomial coefficients in the GPR-GNN paper[1]) with low-degree polynomials also, but its  capability is still limited because it is a more constrained model (as shown in our analysis) than PP-GNN having better filter function approximation capability.
> >
> > High degree polynomials are required to model complex shaped filter responses. As we explained and demonstrated in the motivation section, models like GPR-GNN[1] are unable to learn higher degree polynomials, therefore, unable to learn/model complex shaped filter responses, even if we increase the polynomial degree ($K$). This also has an implication on inability to make use of signals from long range nodes. On the other hand, our model is able to learn effectively using spectral partitioning with low-degree polynomials and exploiting long range signals albeit at the expense of loss locality property and some increase in computational cost, a price to be paid to get improved accuracy performance.

---

> > ### Author Response · Authors · 2021-11-18
> > **Response to Comment-4 [Reviewer kGGE]**
> >
> > **[4] To summarize: I am currently under the impression that the apparent complexity-expressivity tradeoff is not as significant as the authors claim. Although it is apparent that that filtering operations are implemented as low-order polynomials over a partition of the spectrum, their piecewise nature completely destroys the locality inherent to low-degree graph filters, which is why such filters are useful in the first place. This is mere speculation on my part, but I suspect that this method would, for instance, fail to generalize well to graphs other than the one it is trained on, even if the structure is somewhat similar.**
> >
> > We have addressed the concerns related to locality properties, low-degree filters in our earlier responses. Furthermore, we would like to clarify that PP-GNN is proposed for transductive setting. Our goal with the paper was to learn a GNN model that is effective across homophilic and heterophilic graphs in the semi-supervised node classification setting (where we would like to label all unlabeled nodes in a given graph), as mentioned in the introduction section. We believe that the question/suspicion raised by the reviewer is relevant/applicable for the inductive setting and it requires filters to be transferrable. We agree that the study of transferable filters is an area of research on its own. To understand and analyze the transferability of the proposed piecewise polynomial filter is beyond the scope of this work.
> >
> > We sincerely hope that our responses provide sufficient explanations/clarifications and addresses all major concerns that the reviewer have. We will include all these details in the supplementary material. We are happy to answer any further queries.

---

> ### Author Response · Authors · 2021-11-20
> **Follow-up on response to Reviewer kGGE**
>
> Dear Reviewer kGGE,
>
> We would like to thank you for your insightful comments and suggestions that have significantly improved the paper. We hope that our responses and revision address your concerns satisfactorily and will lead to a more positive evaluation of our work. With the discussion window closing soon, we wanted to know if there are any further clarifications that we can provide from our end.

---

### Official Review · Reviewer_fpGz · 2021-11-02

**Correctness:** 4
**Technical Novelty And Significance:** 2
**Empirical Novelty And Significance:** 3
**Recommendation:** 5
**Confidence:** 4

**Details Of Ethics Concerns:**

There is no foreseeable ethics concern with this paper.

**Main Review:**

Strengths:
1) It is natural to utilize the piece-wise polynomial to learn a graph filter on different subsets of the spectrum.
2) The authors demonstrate that the sum of the polynomials approach is more accurate than a single polynomial at approximating a latent optimum filter and that the set of learnable filters is wider than that of GPR-GNN.
3) The experiments demonstrate that PP-GNN is capable of learning more complicated filters, such as capturing complex shapes on Squirrel and Chameleon, outperforming other advanced GNN models.

Weaknesses:
1) As the author discussed in the paper, PP-GNN cannot be scaled to big graphs due to the complexity of the eigendecomposition, and therefore must make a trade-off between performance and efficiency.
2) Once we know the eigen components at the top and bottom of the spectrum, we can alter the eigenvalues in a variety of ways. Additional research is needed to determine whether a polynomial base is the best choice for filter learning.
3) PP-GNN has a large number of hyperparameters (the number of partitions, the order of the polynomial filter, the number of eigenvalues, and so on), and the code demonstrates that the polynomial initialization for each partition has a separate set of hyperparameters as well. I'm concerned that a large number of hyperparameters will degrade the model's generalizability and result in considerable tuning work.
4) The implementation performs SVD on the node feature matrix, which is not discussed in the paper. In particular, the time cost for this operation should be discussed.
5) There are several related works that are missing, including ARMA[1], AdaGNN[2], and BernNet[3], all of which are capable of learning graph spectrum filters.
[1] Filippo Maria Bianchi, Daniele Grattarola, Lorenzo Livi, and Cesare Alippi. Graph neural networks with convolutional arma filters. TPAMI, 2021.
[2] Yushun Dong, Kaize Ding, Brian Jalaian, Shuiwang Ji, and Jundong Li. Graph neural networks with adaptive frequency response filter. In CIKM, 2021.
[3] Mingguo He, Zhewei Wei ,  Zengfeng, Hunang, and Hongteng, Xu. BernNet: Learning Arbitrary Graph Spectral Filters via Bernstein Approximation. In NeurIPS 2021.


**Summary Of The Paper:**

 This paper proposes PP-GNN, a novel graph neural network that learns multiple adaptive polynomial filters acting on different subsets of the eigenvalues. The authors combine GPR-GNN with existing efficient algorithms for generating top and bottom eigen components to reduce the expensive complexity of eigendecomposition. They show that the piece-wise polynomial method can approximate a latent optimal filter better than a single polynomial in theory.

**Summary Of The Review:**

The paper contains some interesting ideas, as evidenced by the experimental results. However, the theoretical depth and novelty may not be enough to meet ICLR's standards.

---

> ### Author Response · Authors · 2021-11-17
> **Response to Weakness 1 (Part-1)**
>
> We thank the reviewer for their valuable feedback and for making us aware of relevant baselines. We try out best to address the concerns raised by the reviewer.
>
> **[1] As the author discussed in the paper, PP-GNN cannot be scaled to big graphs due to the complexity of the eigendecomposition, and therefore must make a trade-off between performance and efficiency.**
>
> We agree with the reviewer that there could be a trade-off between performance and efficiency. In the submitted paper, the largest dataset, OGBN-ArXiv, had ~170k nodes. In the current version, to further evaluate our model's scalability, we perform an additional experiment on the Reddit dataset that has ~250k nodes. Our model achieves 96.81% test accuracy by adapting as small as 512 top and 512 bottom eigenpairs. This performance is very close to the state-of-the-art performance [1]: 97.02% reported in the literature. It took ~30 minutes (on a machine with Intel Xeon 2.60Ghz processor, 112GB Ram, Nvidia Tesla P-100 GPU with 16GB of memory) to compute the top and bottom 1024 eigenvalues/vectors for this dataset. We will include results from this dataset also in the revised version. We are yet to test our model on even larger graphs (a million nodes) and plan to take it up as future work.
>
> We would like to add that eigendecomposition is a one-time pre-compute cost. Its cost can be amortized over the training time taken to learn models across many different configurations for hyperparameter optimization (as needed with all models). The reviewer may be interested in seeing our results from a timing comparison experiment for an illustration of amortization effect (see TABLE [1, 2, 3]). Therefore, we believe that scalability is not a major problem for medium to large-sized graphs. **(TABLE 2 and 3 are available in the follow-up responses)**
>
> But, we agree that scalability could become a potential limitation for very large to extremely large graphs (both in terms of memory and speed).
>
> ---
> ### TABLE 1: PP-GNN’s per component timing cost
> ---
>
> |   PP-GNN   | Avg. Training Time (sec) | EVD Cost (sec) [ONE TIME] |  #EV's obtained | Effective Training Time (sec) |
> |:----------:|:-------------:|:--------:|:---------------:|:--------------:|
> | **Texas**      |     11.89     | 0.007473 |  183 (All Evs)  |      11.89     |
> | **Cornell**    |     11.63     | 0.032709 |  183 (All Evs)  |      11.63     |
> | **Wisconsin**  |     12.08     | 0.012245 |  251 (All Evs)  |      12.08     |
> | **Chameleon**  |     21.44     | 3.718832 |       2048      |      21.48     |
> | **Squirrel**   |     31.38     | 15.81523 |       2048      |      31.54     |
> | **Cora**       |     22.46     |  54.3684 |       2048      |      23.00     |
> | **Citeseer**   |     20.51     | 56.97437 |       2048      |      21.08     |
> | **Cora-Full**  |     63.98     | 155.3041 |       2048      |      65.53     |
> | **Pubmed**     |     52.54     | 256.7104 |       2048      |      55.11     |
> | **Computers**  |     28.63     | 76.27381 |       2048      |      29.39     |
> | **Photo**      |      19.3     | 48.36831 |       2048      |      19.78     |
> | **Flickr**     |     161.16    | 304.1138 |       2048      |     164.20     |
> | **OGBN-ArXiv** |     189.94    | 412.5042 |       1024      |     194.07     |
> | **WikiCS**     |     27.92     | 65.43761 |       2048      |      28.57     |
>
> $\textbf{Table description:}$ Training Time refers to the end to end training time (without eigen decomposition) averaged across 100 trials. EVD cost refers to the time taken to obtain 'x' top and bottom eigenvalues. This 'x' can be found in the ‘Number of EV’s obtained’ column.  Since EVD is a one time cost, we average this cost over the total number of trials and add it to the training time.  We refer to this cost as the Effective Training Time.
>
> ---
>
> **References:**
>
> [1]: Yu Rong, Wenbing Huang, Tingyang Xu, & Junzhou Huang. (2020). DropEdge: Towards Deep Graph Convolutional Networks on Node Classification.

---

> > ### Author Response · Authors · 2021-11-17
> > **Response to Weakness 1 (Part-2)**
> >
> > ### TABLE 2: Training Time (in seconds) across Models
> > ----
> >
> > |  Dataset  | GPR-GNN [1] | PP-GNN (our model) |  MLP |   GCN [2]  | BernNet [3] |  ARMA [4]  |  AdaGNN [5]  |
> > |:---------:|:-------:|:------:|:----:|:------:|:-------:|:------:|:--------:|
> > | **Texas**     |   9.27  |  11.89 | 1.08 |  3.46  |   5.59  |    6   |   13.97  |
> > | **Cornell**   |   9.41  |  11.63 | 1.06 |  3.69  |   5.37  |  5.51  |   12.56  |
> > | **Wisconsin** |   9.67  |  12.08 | 1.07 |  3.42  |   5.69  |  5.36  |   13.57  |
> > | **Chameleon** |  14.69  |  21.48 |  2.6 |  6.42  |  12.46  |  7.84  |   28.77  |
> > | **Squirrel**  |  18.94  |  31.54 | 5.04 |  7.52  |  17.82  |  28.87 |   90.36  |
> > | **Cora**      |   12.9  |  23.00 | 1.95 |  5.94  |  12.25  |  10.67 |   22.15  |
> > | **Citeseer**  |  10.62  |  21.08 | 3.72 |  4.56  |   9.52  |  19.5  |   35.34  |
> > | **Cora-Full** |  24.98  |  65.53 | 7.77 |  8.01  |  31.26  |  40.21 |  175.58  |
> > | **Pubmed**    |    14   |  55.11 | 6.21 |  11.73 |  12.64  |  27.76 |  162.01  |
> > | **Computers** |   7.67  |  29.39 | 2.24 |  6.68  |   7.48  |  27.76 |  118.43  |
> > | **Photo**     |   8.58  |  19.78 | 1.68 |   5.1  |   7.95  |  14.34 |   45.46  |
> > | **Flickr**    |  42.64  | 164.20 |  21  |  30.4  |  62.11  |  119.3 | 178.7371 |
> > | **ArXiv**     |  118.35 | 194.07 | 78.9 | 102.88 |  693.92 | 771.59 |  307.84  |
> > | **WikiCS**    |  14.37  |  28.57 | 3.34 |  10.8  |  11.43  |  30.79 |   73.63  |
> >
> > $\textbf{Note:}$ For PP-GNN model we report the effective training time, as described in the description of TABLE 1.
> >
> > ---
> >
> > $\\\\$
> > $\\\\$
> >
> > ### TABLE 3: Training Time of models relative to the training time of GCN
> > ---
> > |  Dataset  | GPR-GNN | PP-GNN |  MLP |  GCN | BernNet | ARMA | AdaGNN |
> > |:---------:|:-------:|:------:|:----:|:----:|:-------:|:----:|:------:|
> > | **Texas**     |   2.68  |  3.44  | 0.31 | 1.00 |   1.62  | 1.73 |  4.04  |
> > | **Cornell**   |   2.55  |  3.15  | 0.29 | 1.00 |   1.46  | 1.49 |  3.40  |
> > | **Wisconsin** |   2.83  |  3.53  | 0.31 | 1.00 |   1.66  | 1.57 |  3.97  |
> > | **Chameleon** |   2.29  |  3.35  | 0.40 | 1.00 |   1.94  | 1.22 |  4.48  |
> > | **Squirrel**  |   2.52  |  4.19  | 0.67 | 1.00 |   2.37  | 3.84 |  12.02 |
> > | **Cora**      |   2.17  |  3.87  | 0.33 | 1.00 |   2.06  | 1.80 |  3.73  |
> > | **Citeseer**  |   2.33  |  4.62  | 0.82 | 1.00 |   2.09  | 4.28 |  7.75  |
> > | **Cora-Full** |   3.12  |  8.18  | 0.97 | 1.00 |   3.90  | 5.02 |  21.92 |
> > | **Pubmed**    |   1.19  |  4.70  | 0.53 | 1.00 |   1.08  | 2.37 |  13.81 |
> > | **Computers** |   1.15  |  4.40  | 0.34 | 1.00 |   1.12  | 4.16 |  17.73 |
> > | **Photo**     |   1.68  |  3.88  | 0.33 | 1.00 |   1.56  | 2.81 |  8.91  |
> > | **Flickr**    |   1.40  |  5.40  | 0.69 | 1.00 |   2.04  | 3.92 |  5.88  |
> > | **ArXiv**     |   1.15  |  1.89  | 0.77 | 1.00 |   6.74  | 7.50 |  2.99  |
> > | **WikiCS**    |   1.33  |  2.65  | 0.31 | 1.00 |   1.06  | 2.85 |  6.82  |
> > | **Average**   |   2.03  |  4.09  | 0.50 | 1.00 |   2.19  | 3.18 |  8.39  |
> > ---
> >
> > $\textbf{Description:}$ Relative training time for models in comparison with respect to the GCN model obtained by computing the ratio of the average training time of each model to that of GCN.
> >
> > $\textbf{Observation:}$ We observe that PP-GNN (our model) is $\sim$4x slower than GCN, $\sim$2x slower than GPR-GNN and  BernNet, $\sim$1.5x slower than ARMA, and $\sim$2x faster than AdaGNN.
> >
> > $\\\\$
> > ---
> > **References:**
> > ---
> > [1] Eli Chien, Jianhao Peng, Pan Li, & Olgica Milenkovic. (2021). Adaptive Universal Generalized PageRank Graph Neural Network.
> >
> > [2] Thomas N. Kipf and Max Welling (2016). Semi-Supervised Classification with Graph Convolutional Networks. CoRR, abs/1609.02907.
> >
> > [3] Mingguo He, Zhewei Wei , Zengfeng, Hunang, and Hongteng, Xu. BernNet: Learning Arbitrary Graph Spectral Filters via Bernstein Approximation. In NeurIPS 2021
> >
> > [4] Filippo Maria Bianchi, Daniele Grattarola, Lorenzo Livi, and Cesare Alippi. Graph neural networks with convolutional arma filters. TPAMI, 2021.
> >
> > [5] Yushun Dong, Kaize Ding, Brian Jalaian, Shuiwang Ji, and Jundong Li. Graph neural networks with adaptive frequency response filter. In CIKM, 2021.

---

> ### Author Response · Authors · 2021-11-17
> **Response to Weakness 2**
>
> **[2] Once we know the eigen components at the top and bottom of the spectrum, we can alter the eigenvalues in a variety of ways. Additional research is needed to determine whether a polynomial base is the best choice for filter learning.**
>
> The  main  focus  of  this  work  is  to  address  the  problem  of  learning  complex-shaped  filter  responses  to  get  improved  performance  on  diverse  graphs  (both homophilic and heterophilic).  We proposed an effective piecewise polynomial filtering based method to address this problem along with some theoretical explanations and show how such a method helps.  We agree with the reviewer that more research is needed to find the best way (according to some criterion) to adapt the eigenvalues (not necessarily using polynomials).  But, this is beyond the scope of our current work, and we will include it as a potential future direction in a discussion section.

---

> ### Author Response · Authors · 2021-11-17
> **Response to Weakness 3 (part-1)**
>
>
> **[3] PP-GNN has a large number of hyperparameters (the number of partitions, the order of the polynomial filter, the number of eigenvalues, and so on), and the code demonstrates that the polynomial initialization for each partition has a separate set of hyperparameters as well. I'm concerned that a large number of hyperparameters will degrade the model's generalizability and result in considerable tuning work.**
>
> $\textbf{Hyperparameter Tuning}:$ As rightly observed by the reviewer, performing a regular grid search on the prescribed ranges for all the hyperparameters of PP-GNN is combinatorial and is expensive. Unfortunately, we missed mentioning one crucial point in this regard. Due to the impracticality of regular grid search, we use the Optuna framework [1] to explore the hyperparameter space. In all our experiments, we set the number of Optuna Trials to 100 and found that this order of search, is practically sufficient to achieve good performance.
>
> $\textbf{Polynomial (gamma) initialization:}$ We perform an ablative study to understand the effect of polynomial initialization. We vary the initialization scheme in [PPR, NPPR and Random], as described in GPRGNN [2]. We observe that our model (PP-GNN) retains similar performance even with `random' initialization. For brevity, we report results on six datasets (Cora, Citeseer, Squirrel, Chameleon, Texas and Wisconsin). The results can be found in Table 1. Our experiment results suggest that tuning polynomial initialization for each partition can indeed be avoided without compromising on the performance.
>
> $\\\\$
> $\\\\$
>
> ---
> ### TABLE-1: PP-GNN with Varying Initialization
> ---
>
>
> |                           | Chameleon                  | Citeseer                 | Cora                       | Squirrel                   | Texas                      | Wisconsin                 |
> |---------------------------|----------------------------|--------------------------|----------------------------|----------------------------|----------------------------|---------------------------|
> | **PP-GNN (NPPR)**            |     66.73 (1.86)           |     78.41 (1.54)         |     89.52 (0.85)           |     57.93 (1.46)           |     88.38 (3.61)           |     87.25 (2.96)          |
> | **PP-GNN (Random)**           |     68.71 (2.47)           |     78.28 (1.70)         |     89.42 (0.98)           |     57.59 (1.56)           |     88.38 (3.48)           |     88.63 (2.58)          |
> | **PP-GNN (PPR)**              |     67.74 (2.31)           |     78.25 (1.76)         |     89.52 (0.85)           |     56.86 (1.20)           |     89.73 (4.90)           |     88.24 (3.33)          |
> | **Best Performing Baselines** |     65.45 (1.17)  **AdaGNN**   |     78.15 (0.74)    **ARMA** |     88.26 (1.32)*    **TDGNN** |     53.50 (0.96)    **AdaGNN** |     84.86 (6.77)*    **H2GCN** |     86.67 (4.69)*    **H2GCN** |
>
> $\textbf{Description:}$ Results for PP-GNN using different polynomial (gamma) initializations versus the best performing baseline models. The names of the best performing baseline models are given right below the test accuracy, in the `Best Performing Baselines' row. '*' indicates that the results were borrowed from the corresponding papers.
>
>
> $\textbf{References}$
>
>
> [1] Takuya Akiba, Shotaro Sano, Toshihiko Yanase, Takeru Ohta, & Masanori Koyama. (2019). Optuna: A Next-generation Hyperparameter Optimization Framework.
>
> [2] Eli Chien, Jianhao Peng, Pan Li, & Olgica Milenkovic. (2021). Adaptive Universal Generalized PageRank Graph Neural Network.

---

> > ### Author Response · Authors · 2021-11-22
> > **Response to Weakness-3 (part-2)**
> >
> > We conducted additional experiments, where we have significantly reduced the hyper-parameters search space and ran our model with only 20 Optuna trials. Training all the models on these hyper-parameters with 100 Optuna Trials allowed us to convince ourselves (and the reviewer) that we are getting a good performance, not because of some statistical anomaly but the fact that our method holds merit.
> >
> > With the model's merit established, we can cut down on ranges for several hyper-parameters of our model, while retaining competitive performance across datasets. Below is the reduced hyper-params space:
> >
> > - Learning Rate: [0.003, 0.005]
> > - Weight_Decay : [0.0001, 0.001]
> > - Dropout: [0.3, 0.5]
> > - Hidden Dims: [32, 64]
> > - Order of the Polynomial: [2, 4]
> > - No. of Partitions (Buckets): [2, 4]
> > - Number of Eigen Value: [256, 1024]
> > - $\eta$: Sampled uniformly between [0, 1] using Optuna
> >
> >
> >
> > Please note that we get rid of partition specific hyper-parameters (polynomial coefficients -- gammas). As shown in ablative study (response to weakness-3 (part-1)) that randomly initializing gammas suffices. We retrain our model using this new set of params described above on a subset of datasets and report the numbers in TABLE-1. Importantly, **we restrict to using 20 Optuna trials.**
> >
> >
> >
> > $\\\\$
> >
> >
> >
> > ---
> > ### TABLE-1: PP-GNN with Reduced Hyper-parameters
> > ---
> >
> >
> >
> > | | Texas | Squirrel | Chameleon | Cora-Full (PCA) | Ogbn-arxiv | Citeseer | Pubmed | Cora | Photos (Amazon) |
> > |---------------------------------|:--------------------------------:|:------------------------------:|:------------------------------:|:---------------------------:|:---------------------:|:--------------------------------:|:------------------------------:|:-------------------------------:|:------------------------------:|
> > | **PP-GNN** | 85.14 (2.30) | 59.15 (1.91) | 69.10 (1.37) | 60.93 (0.83) | 69.10 | 77.87 (1.93) | 89.43 (0.47) | 88.87 (0.90) | 92.18 (0.58) |
> > | **Best Performing Baseline** | 84.86 (6.77)* **H2GCN** | 53.50 (0.96) **AdaGNN** | 65.45 (1.17) **AdaGNN** | 61.84 (0.90) **LGC** | 69.37 **GCN** | 77.07 (1.64)* **H2GCN** | 89.59 (0.33)* **H2GCN** | 88.26 (1.32) **TDGNN** | 92.54 (0.28) **TDGNN** |
> >
> >
> >
> >
> > $Observation:$ Even with fewer hyper-params, our model continues to perform competitively or better.
> >
> >
> >
> > Next, we examine the end-to-end training time (including EVD) for this new setting on several datasets and report in TABLE-2.
> >
> >
> > $\\\\$
> >
> > ---
> > ### TABLE-2: End-to-End Training Time (20 Trials)
> > ---
> >
> >
> >
> > | Dataset | chameleon | citeseer | computers | cora | corafull | photo | pubmed | squirrel | texas | wisconsin | OGBN-ArXiv |
> > |:----------:|:---------:|:--------:|:---------:|:--------:|:--------:|:--------:|:--------:|:--------:|:--------:|:---------:|:----------:|
> > | Time (sec) | 00:03:46 | 00:10:17 | 00:34:37 | 00:05:24 | 00:59:29 | 00:10:31 | 00:57:40 | 00:10:38 | 00:02:27 | 00:02:33 | 01:03:20 |
> >
> >
> >
> > **Note:** Time reported in HH:MM:SS format.
> >
> >
> >
> > We can infer from TABLE-2 that the total training time has significantly reduced across all the datasets. This reduction is mainly achieved by tuning only over lower polynomial orders and fixed number of polynomial partitions. The major bottleneck in terms of time in the prior experiment (100 trials) was due to the higher range of both polynomial order and number of polynomial partitions. The higher the polynomial order, and the higher the number of polynomial partitions, the more time it would take to complete a single trial.
> >
> > We hope these experiments address the reviewer's concerns.

---

> ### Author Response · Authors · 2021-11-17
> **Response to Weakness 4**
>
> **[4] The implementation performs SVD on the node feature matrix, which is not discussed in the paper. In particular, the time cost for this operation should be discussed.**
>
>  The SVD of the node feature matrix is **NOT** necessary for the proposed method. It is just a remnant of a trial experiment we had done earlier and forgot to remove it. This is a miss on our part. We are grateful to the reviewer for helping us to discover this miss. The reviewer may note that the current implementation does perfect reconstruction. Therefore, all our experiment results correspond to using the original feature matrix only.

---

> ### Author Response · Authors · 2021-11-17
> **Response to Weakness 5**
>
> **[5] There are several related works that are missing, including ARMA[1], AdaGNN[2], and BernNet[3], all of which are capable of learning graph spectrum filters. [1] Filippo Maria Bianchi, Daniele Grattarola, Lorenzo Livi, and Cesare Alippi. Graph neural networks with convolutional arma filters. TPAMI, 2021. [2] Yushun Dong, Kaize Ding, Brian Jalaian, Shuiwang Ji, and Jundong Li. Graph neural networks with adaptive frequency response filter. In CIKM, 2021. [3] Mingguo He, Zhewei Wei , Zengfeng, Hunang, and Hongteng, Xu. BernNet: Learning Arbitrary Graph Spectral Filters via Bernstein Approximation. In NeurIPS 2021**
>
> Based on the suggestion from the reviewer, we conducted more experiments with additional baselines: ARMA [1] (which we missed) and very recent works like ADAGNN [2] and BernNet [3]. We report these results in TABLE-1. The main observation is that our model's standing remains unaffected across all datasets (other than OGBN-ArXiv). However, the improvement gap has reduced from $\sim$10\% to $\sim$5\%. We will include these results and modify our claims accordingly in the revised version.
>
> $\textbf{Additional Experiment Details:}$ For ADAGNN and BernNet, we use the authors code and tune it over the hyperparameters as provided in the paper. For ARMA, we use the official PyTorch implementation. As a sanity check, we also tested ARMA on the node classification datasets described in the paper and reproduced similar numbers. We will update our paper accordingly to accommodate these new baselines.
>
> Overall, we are thankful to the reviewer for helping us to improve the paper quality by finding several misses (including an important baseline ARMA) and giving us an opportunity to clarify and fix them.
>
> $\\\\$
> $\\\\$
> ---
> ### TABLE-1: Additional Baselines
> ---
>
> |  | Texas | Wisconsin | Squirrel | Chameleon | Cornell | Flickr | Cora-Full | OGBN-ArXiv | Wiki-CS | Citeseer | Pubmed | Cora | Computer | Photos |
> |:-----------:|:----------------:|:---------------------:|:---------------------:|:---------------------:|:---------------------:|:---------------:|:----------------------:|:-----------------:|:---------------------:|:---------------------:|:----------------------:|:---------------------:|:---------------------:|:---------------------:|
> | **BernNET** | 83.24 (6.47) | 84.90 (4.53) | 52.56 (1.69) | 62.02 (2.28) | 80.27 (5.41) | 52.35 | 60.77 (0.92) | 67.32 | 79.75 (0.52) | 77.01 (1.43) | 89.03 (0.55) | 88.13 (1.41) | 83.69 (1.99) | 91.61 (0.51) |
> | **ARMA** | 79.46 (3.65) | 82.75 (3.56) | 47.37 (1.63) | 60.24 (2.19) | 80.27 (7.76) | 53.79 | 60.23 (1.21) | **69.49** | 78.94 (0.32) | 78.15 (0.74) | 88.73 (0.52) | 87.37 (1.14) | 78.55 (2.62) | 90.26 (0.48) |
> | **AdaGNN** | 71.08 (8.55) | 77.70 (4.91) | 53.50 (0.96) | 65.45 (1.17) | 71.08 (8.36) | 52.30 | 59.57 (1.18) | 69.44 | 77.87 (4.95) | 74.94 (0.91) | 89.33 (0..57) | 86.72 (1.29) | 81.27 (2.10) | 89.93 (1.22) |
> | **GPR-GNN [4]** | 81.35 (5.32) | 82.55 (6.23) | 46.31 (2.46) | 62.59 (2.04) | 78.11 (6.55) | 52.74 | 61.37 (0.96) | 68.44 | 79.68 (0.50) | 76.84 (1.69) | 89.08 (0.39) | 87.77 (1.31) | 82.38 (1.60) | 91.43 (0.89) |
> | **PP-GNN** (Our Model) | **89.73 (4.90)** | **88.24 (3.33)** | **56.86 (1.20)** | **67.74 (2.31)** | **82.43 (4.27)** | **54.44** | **61.42 (0.79)** | 69.28 | **80.04 (0.43)** | **78.25 (1.76)** | **89.71 (0.32)** | **89.52 (0.85)** | **85.23 (1.36)** | **92.89 (0.37)** |
>
> ---
> #### **References**
> ---
> [4]: Eli Chien, Jianhao Peng, Pan Li, & Olgica Milenkovic. (2021). Adaptive Universal Generalized PageRank Graph Neural Network.

---

> ### Author Response · Authors · 2021-11-20
> **Follow-up on response to Reviewer fpGz**
>
>  Dear Reviewer fpGz ,
>
> We would like to thank you for your insightful comments and suggestions that have significantly improved the paper.
> We hope that our responses and revision address your concerns satisfactorily and will lead to a more positive evaluation of our work. With the discussion window closing soon, we wanted to know if there are any further clarifications that we can provide from our end.

---

### Author Response · Authors · 2021-11-25
**Updated Paper (To all the Reviewers)**

We thank all the reviewers for their valuable feedback. We hope that we have addressed all the concerns raised.

We have updated our paper to reflect these changes. We point below to specific sections/subsections for the concerns raised by each reviewer.

---

**Reviewer fpGz:**

- Computational Complexity and Efficiency: $A.6$
- Note on Hyperparameter Tuning: $A.5.7$, $A.6$
- New Relevant Baselines and Clarifications: $A.5.2$, $A.5.3$, $A.5.4$ and $A.5.8$

---

**Reviewer kGGE:**

- Computational Complexity and Efficiency: $A.6$

---

**Reviewer SUfF:**

- Note on Hyperparameter Tuning: $A.5.7$, $A.6$
- Comprehensive Computational Time Analysis: $A.6$
- Discrepancies between different GCN implementations (OGBN-ArXiv): $A.7$


---

**Reviewer eu6Q:**

- New Relevant Baselines and Clarifications (including GFIR and the significance of the MLP): $A.5.2$, $A.5.3$, $A.5.4$ and $A.5.8$
- Comprehensive Computational Time Analysis: $A.6$
- We have updated the paper title.
- We missed adding a discussion between the proposed method and scattering transforms in the paper because of time constraints. We will include this discussion in the final version of the paper.

---

### Author Response · Authors · 2021-11-30
**Revised Evaluation [Reviewers eu6Q, SUfF, kGGE, fpGz]**

Dear Reviewers,

We thank you for reviewing our work. Additionally, we are thankful to **Reviewers eu6Q, SUfF, and kGGE** for participating in rebuttal phase and providing feedback. We believe we have addressed all the raised concerns thoroughly and have updated the appendix section to reflect them. We promise to make any remaining changes (very few) in the final revision.

We request you to peruse through the responses and revise the score appropriately, or provide us with justifications for the scores post rebuttal.

Thank you.

---

### Decision · Program_Chairs · 2022-01-20

**Decision:**

Reject

**Comment:**

This paper proposes to apply a piece-wise polynomial filter on the spectral corresponding to the graph convolution to enhance the model expressivity of graph neural networks. The effectiveness of the proposed model is investigated through numerical experiments and it was shown that the method achieves fairly nice performances.

This paper gives a natural extension to the usual adaptive Generalized PageRank approaches to more expressive piece-wise polynomial filters. However, the reviewers are not enthusiastic on this paper. This is mainly because of the following concerns: (1) Since it requires diagonalization of the aggregation operator, it requires much more computational burden than the usual polynomial filters, which prevents the method from being applied to data with much more large size. (2) The choice of the filter could be more investigated, in particular, the complexity-expressivity trade-off (in other words, bias-variance trade-off) could be discussed more, for example, by theoretical work.

In summary, the paper seems not to be well matured for being published in ICLR conference.